# Balancing Plasticity and Stability with Fast and Slow Successor Features

**Raymond Chua** [1 2]   **Doina Precup** [† 1 2 3 4]   **Blake Richards** [† 1 2 3 5 6]

## Abstract

A hallmark of intelligence is the ability to adapt in non-stationary environments, yet deep Reinforcement Learning (RL) agents often struggle in such settings. Prior studies introduce non-stationarity through abrupt shifts in features or dynamics, whereas real-world environments often evolve gradually through continual drift. This distinction has important implications for the "stability-plasticity dilemma" in RL, as abrupt task changes may demand more plasticity than naturalistic settings. To address this, we modify existing 3D Miniworld and MuJoCo environments to incorporate naturalistic, continual non-stationarity, and use them to examine how stability and adaptation affect performance under continuous environmental change. We find that methods favoring stability, such as synaptic consolidation, outperform approaches focused on plasticity, such as parameters resetting. Motivated by this result, and prior evidence that Successor Features (SFs) reduce interference, we investigate whether SFs are better consolidation targets than Q-values. Across both environments, applying neuro-inspired synaptic consolidation to SFs yields superior performance on continually changing settings. Moreover, consolidation is most effective when SFs are stabilized across multiple timescales, which capture complementary aspects of gradual environmental change. Together, these results suggest that stability is more critical in continual learning when changes are gradual, and that multi-timescale consolidation of predictive representations is an effective approach.

## 1. Introduction

Events in the real world are often constantly evolving. Humans and animals must therefore adapt in environments where the underlying dynamics shift naturally and continually. In contrast, many continual learning studies in Artificial Intelligence (AI) focus on abrupt, task-boundary changes, where the features or dynamics across tasks differ substantially. Standard RL techniques, such as Q-learning, struggle under such conditions and often suffer from catastrophic forgetting (McCloskey & Cohen, 1989; French, 1999). Developing methods that enable deep RL agents to learn effectively in naturalistic, continually changing environments remain a major goal in AI research (Khetarpal et al., 2022; Abel et al., 2023; Silver & Sutton, 2025).

While early work in supervised continual learning emphasized stability (i.e., the ability to retain previously acquired knowledge and prevent catastrophic forgetting) (Kirkpatrick et al., 2017; Zenke et al., 2017), RL poses unique challenges, since changing policies alter the samples an agent encounters, compounding environmental non-stationarity. In RL, Atari (Bellemare et al., 2013) has emerged as one of the standard testbeds for sequential task learning, where stability-focused methods such as Elastic Weight Consolidation (EWC) (Kirkpatrick et al., 2017) and replay (Rolnick et al., 2019) became dominant strategies. More recently, studies have shifted towards the complementary issue of plasticity (i.e., the capacity to rapidly adapt to new experiences), using either sequential Atari tasks (Abbas et al., 2023) or artificial tasks in MuJoCo created by randomly sampling friction coefficients per task (Dohare et al., 2024). Stability has mostly been studied in sequential multi-task settings and plasticity in single-task dynamics, but real-world environments rarely fit either case, as naturalistic, continual non-stationarity appears as a single task while still creating a stream of shifting sub-tasks.

Despite these advances, it remains unclear how stability and plasticity trade off in more real-world-like environments that undergo naturalistic, continual non-stationarity, where agents must adapt to naturalistic and continual changes without explicit task boundaries. A natural way to study this problem is to develop environments with naturalistic, continually evolving dynamics, and compare algorithms that are task agnostic, and either enhance plasticity (e.g., param-

---

[†]Joint senior authorship. [1]School of Computer Science, McGill University, Montréal, Canada [2]Mila - Quebec Artificial Intelligence Institute, Montréal, QC, Canada [3]CIFAR Learning in Machines and Brains [4]Google Deepmind [5]Dept of Neurology & Neurosurgery, Montréal, Canada [6]Montreal Neurological Institute of McGill University, Montréal, Canada. Correspondence to: Raymond Chua <ray.r.chua@gmail.com>.

*Proceedings of the 43rd International Conference on Machine Learning*, Seoul, South Korea. PMLR 306, 2026. Copyright 2026 by the author(s).

eter resets (Nikishin et al., 2022; 2023; Sokar et al., 2023; Dohare et al., 2024; Lee et al., 2024)) or preserve stability (e.g., consolidation that either protects important parameters (Kirkpatrick et al., 2017) or allow learning across multiple timescales (Kaplanis et al., 2018; 2019; Anand & Precup, 2023)). While other approaches exist, such as replay-based methods (Riemer et al., 2018; Rolnick et al., 2019; Caccia et al., 2023), they are less suited to our setting since their benefits rely on storing and mixing past and recent samples, which is problematic when no clear task boundaries exist. More broadly, prior approaches tackle the stability-plasticity trade-off at the level of Q-values or policies, leaving the role of representations largely understudied.

In this paper, we investigate whether predictive representations can offer a principled solution to the plasticity–stability dilemma under naturalistic, continual non-stationarity. We focus on Successor Features (SFs) (Barreto et al., 2017; Borsa et al., 2018; Chua et al., 2024), which capture predictive structure and enable transfer across tasks with shared dynamics, and study whether they can simultaneously support rapid adaptation and resistance to interference. We evaluate this question in two environments with continuously evolving dynamics: a slippery four rooms environment, where actions are occasionally replaced, and MuJoCo control tasks, where the embodiment's mass varies over time. These sources of non-stationarity reflect realistic changes in action outcomes (e.g., wet or icy ground) and body dynamics. Non-stationarity is induced via continuous stochastic drift processes, including noisy sinusoidal dynamics (Xie et al., 2020), as well as its non-periodic variant and Ornstein–Uhlenbeck (OU) drift.

In summary, our main contributions in this paper are:

1. **A naturalistic continual non-stationarity evaluation protocol.** We introduce a continual RL setup with smooth, continuous non-stationarity and no explicit task boundaries, instantiated using periodic, or non-periodic stochastic sine functions or OU dynamics, in both navigation and continuous control domains.

2. **A controlled diagnosis of the plasticity–stability trade-off.** By systematically comparing mechanisms that inject plasticity with those that preserve stability, we provide evidence that performance degradation under continuous non-stationarity is primarily driven by instability rather than insufficient plasticity.

3. **A novel integration of SFs with multi-timescale synaptic consolidation.** We propose a principled framework that combines predictive representations (SFs) with synaptic consolidation across multiple timescales, enabling stable learning under continuous non-stationarity while preserving adaptability.

4. **Interpretability of predictive representations across timescales.** We use cross-attention over SFs learned at different consolidation timescales as a diagnostic tool to quantify their relative contributions, providing new insights into how stability and plasticity are distributed over temporal dimensions.

## 2. Related work

Our work builds upon prior studies of stability or plasticity in RL. Early work to mitigate forgetting emphasized stability, introducing methods that use importance measures such as Fisher information to protect parameters critical for previous tasks (Kirkpatrick et al., 2017; Schwarz et al., 2018), manipulating replay mechanisms (Rolnick et al., 2019; Riemer et al., 2018; Kaplanis et al., 2020; Caccia et al., 2023), augmenting architectures (Powers et al., 2022), or employing consolidation systems that maintain multiple sets of parameters updated at different timescales (Kaplanis et al., 2018; 2019; Anand & Precup, 2023). Several approaches explicitly rely on task information, for example by using task boundaries to trigger consolidation (Kirkpatrick et al., 2017) or distillation phases (Schwarz et al., 2018). Other approaches that are described as task-agnostic, but nevertheless rely on auxiliary mechanisms such as recency tracking by separating "new" or "replay" samples (Rolnick et al., 2019), or the use of drift detection mechanisms that trigger architecture adaptations (Powers et al., 2022). These assumptions are problematic in environments that evolve naturally and continually, where there are no discrete task boundaries to detect and the very notion of "new" versus "old" experiences becomes ill-defined.

More recently, studies in continual RL have shifted attention to the problem of loss of plasticity. By analyzing neural activities, effective rank of the representations, and gradient dynamics during training, proposed mitigation strategies have focused on modifying the activation functions or optimizers (Ben-Iwhiwhu et al., 2022; Abbas et al., 2023), regularizing the parameters using weight decay or normalization (Lyle et al., 2024), and, more commonly, injecting plasticity by resetting subsets of network parameters such as the last few layers (Nikishin et al., 2022; 2023) or the ones that are least active (Sokar et al., 2023; Dohare et al., 2024). However, most of these approaches have been evaluated only in discrete or single-task settings, or under non-stationarity that is abrupt rather than naturally and continually.

Among these prior approaches, our study is most closely related to consolidation-based approaches (Kirkpatrick et al., 2017; Schwarz et al., 2018; Kaplanis et al., 2018) and to recent efforts examining loss of plasticity in deep RL (Nikishin et al., 2023; Dohare et al., 2024) as they do not require explicit or implicit task statistics. However, they have yet to be evaluated under naturalistic, continually evolving set-

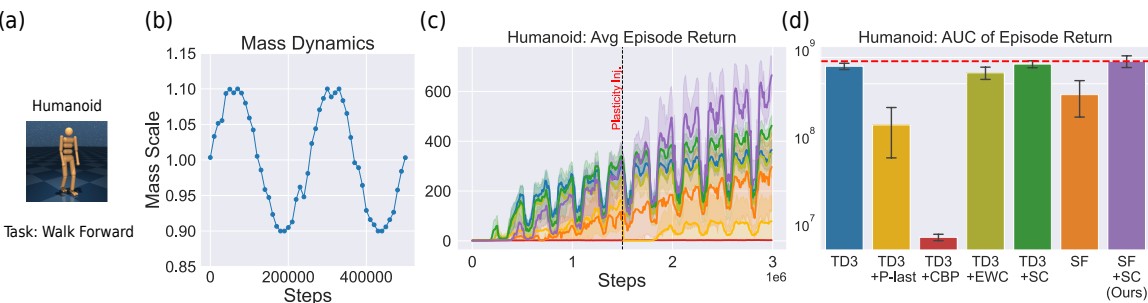

*Figure 1.* Motivating stability-plasticity tradeoffs in naturalistic, continually non-stationary RL where the environment evolves gradually, rather than abruptly. To illustrate, we show **(a)** the Humanoid walking forward task and **(b)** an example of the noisy sine function used to generate smooth changes in its mass. **(c)** Average episode return plot and **(d)** Area under the curve (AUC) show that stability-preserving methods (EWC, SC) outperform purely plastic ones (CBP, P-last), with further gains from consolidating SFs (SF+SC, purple) rather than Q-values (TD3+SC, green). Plasticity injection for TD3+P-last (yellow) was performed halfway through the training.

tings, and remain limited to discrete tasks or single-task settings. Moreover, whether such approaches are effective when applied to learned representations, rather than Q-values or policies, remain unclear. In this work, we address these limitations by analyzing stability and plasticity under naturalistic, continual changes, and by proposing a synaptic consolidation system with SFs, that consolidates representations across multiple timescales.

## 3. Preliminaries

### 3.1. Reinforcement Learning under Continuous Non-Stationarity

A Markov Decision Process (MDP) defined by the tuple $(\mathcal{S}, \mathcal{A}, p, r, \gamma)$, where $\mathcal{S}$ and $\mathcal{A}$ denote the state and action spaces, $p(s' \mid s, a)$ is the transition function, $r : \mathcal{S} \to \mathbb{R}$ is the reward function, and $\gamma \in [0, 1)$ is the discount factor (Sutton & Barto, 2018).

At each time step $t$, the agent observes $S_t \in \mathcal{S}$, selects an action $A_t \sim \pi(\cdot \mid S_t)$, transitions to $S_{t+1} \sim p(\cdot \mid S_t, A_t)$, and receives reward $R_{t+1}$.

Standard MDPs assume stationary dynamics, i.e., a fixed transition function $p(s' \mid s, a)$. However, real-world environments are often non-stationary. Prior work in continual reinforcement learning typically models such non-stationarity as a sequence of discrete tasks with abrupt changes.

In contrast, we consider *continuous non-stationarity*, where the environment evolves gradually over time. We introduce a time-varying latent parameter $\omega_t \in \Omega$ that modulates the transition dynamics, yielding a sequence of MDPs:

$$\mathcal{M}_t = (\mathcal{S}, \mathcal{A}, p_{\omega_t}, r, \gamma), \tag{1}$$

where $p_{\omega_t}(s' \mid s, a) \equiv p(s' \mid s, a; \omega_t)$ varies smoothly with $\omega_t$.

We assume $\omega_t$ evolves according to a continuous stochas-

tic process (e.g., a noisy sine wave), resulting in gradual changes in dynamics rather than abrupt task switches. This setting captures *naturalistic non-stationarity*, where the agent must continually adapt to drifting conditions while retaining prior knowledge. The latent variable $\omega_t$ is not observed by the agent, and its evolution may revisit similar values over time, leading to recurring dynamics (Figure 1b).

### 3.2. Successor Features

Successor Features (SFs) provide a decomposition of the state-action value function into reward parameters and predictive representations of future feature occupancy:

$$Q(S_t, A_t, \boldsymbol{w}) = \psi(S_t, A_t, \boldsymbol{w})^\top \boldsymbol{w}, \tag{2}$$

where $\psi \in \mathbb{R}^n$ captures the expected discounted occupancy of features, and $\boldsymbol{w} \in \mathbb{R}^n$ parameterizes the reward function (Borsa et al., 2018).

Canonically, SFs for a state-action pair $(s, a)$ under a policy $\pi$ are defined as:

$$\psi^\pi(s, a) = \mathbb{E}^\pi \left[ \sum_{i=t}^{\infty} \gamma^{i-t} \phi(S_{i+1}) \mid S_t = s, A_t = a \right], \tag{3}$$

where $\phi \in \mathbb{R}^n$ denotes basis features (Barreto et al., 2017). The reward can be expressed as a linear function of these features:

$$R_{t+1} = \phi(S_{t+1})^\top \boldsymbol{w}. \tag{4}$$

Prior work has primarily leveraged SFs $\psi$ for *transfer learning under stationary dynamics*, where the transition function, $p(s' \mid s, a)$, remains fixed and only the reward parameters $\boldsymbol{w}$ change across tasks. In such settings, SFs enable efficient generalization by reusing learned predictive representations across different reward functions (Barreto et al., 2017; Borsa et al., 2018).

In contrast, this work considers environments with *non-stationary transition dynamics*, where the underlying dy-

namics evolve continuously over time, induced by the time-varying latent parameter $\omega_t$. In our settings, both basis features $\phi$ and SFs $\psi$ must adapt to changing dynamics, $p(s' \mid s, a; \omega_t)$, potentially leading to instability and interference. This raises the key question of whether SFs alone can remain effective under naturalistic, continuously changing dynamics, whether they suffer from limitations in stability or plasticity, and, if so, how these limitations can be mitigated.

To study this, we build on Simple SFs (Chua et al., 2024), which learn SFs directly during interaction without auxiliary losses or pre-training. Both $\psi$ and $\boldsymbol{w}$ are learned jointly via:

$$L_w = \frac{1}{2} \left\| R_{t+1} - \overline{\phi}(S_{t+1})^\top \boldsymbol{w} \right\|^2, \tag{5}$$

$$L_\psi = \frac{1}{2} \left\| \hat{y} - \psi(S_t, A_t, \boldsymbol{w})^\top \boldsymbol{w} \right\|^2, \tag{6}$$

where $\overline{\phi}(S_{t+1})$ is the L2-normalized feature representation treated as constant via a stop-gradient operator. The bootstrapped target is:

$$\hat{y} = R_{t+1} + \gamma \, \max_{a'} \, \psi(S_{t+1}, a', \boldsymbol{w})^\top \boldsymbol{w}. \tag{7}$$

### 3.2.1. ROLE OF THE REWARD PARAMETERS

In our setting, the latent variable $\omega_t$ induces time-varying transition dynamics, but does not define a sequence of discrete tasks with distinct reward functions. Instead, the reward remains a function of the state $S_t$ through the basis features $\phi$ (Eq. 4).

Accordingly, $\boldsymbol{w}$ should be interpreted as a vector of reward parameters that linearly combines basis features $\phi$ to predict rewards, rather than as a task identifier. While $\boldsymbol{w}$ is learned online, it does not track the non-stationarity induced by $\omega_t$. Instead, adaptation to changing dynamics is primarily captured by the basis features $\phi$ and the SFs $\psi$. Following Borsa et al. (2018), we condition SFs on the reward parameter $\boldsymbol{w}$, i.e., $\psi(s, a, \boldsymbol{w})$, ensuring consistency with the decomposition $Q(s, a, \boldsymbol{w}) = \psi(s, a, \boldsymbol{w})^\top \boldsymbol{w}$.

### 3.3. Neuro-inspired Synaptic Consolidation Mechanism

In this work, we revisit the Synaptic Consolidation mechanism (SC) (Benna & Fusi, 2016) which has been previously adapted to deep RL (Kaplanis et al., 2018; 2019). Despite these adaptations, SC remains far less studied than Systems Consolidation (McClelland et al., 1995) in AI.

We revisit SC because (i) it can be learned without explicit or implicit task statistics, (ii) it generalizes beyond dual fast/slow schemes (Anand & Precup, 2023; Lee et al., 2024) by supporting multiple timescales, (iii) its linear-chain formulation provides a principled balance of stability and plasticity without ad-hoc mechanisms, and (iv) it has already

been shown to be effective in deep RL (Kaplanis et al., 2018; 2019). In summary, the multi-timescale mechanism promotes both rapid adaptation and stability. Fast timescale components enable behaviour consistent with *functional plasticity*, defined as the observable ability of an agent to adapt under changing dynamics.

To make this concrete, we now outline the SC mechanism — originally proposed to model how synaptic strength stabilizes over time (Benna & Fusi, 2016) — which can be understood as a chain of $K$ interacting variables, $u_1, u_2, \ldots, u_K$, each associated with a capacity $C_k \in \mathbb{Z}+$. The first variable $u_1$, corresponds to visible synaptic efficacy $v$, i.e., the strength of the connection between two neurons, and is the most plastic component (see Figure 2a. for a schematic of this system). Its dynamics are:

$$C_1 \frac{d_{u_1}}{dt} = \frac{dv}{dt} + g_{1,2}(u_2 - u_1), \quad \text{for k = 1} \tag{8}$$

where $g_{1,2} \in \mathbb{R}$ determines the flow strength between $u_1$ and $u_2$.

Interior variables $u_k$ (for k = 2,3, ..., K-1), interact bidirectionally with their two neighbors:

$$C_k \frac{d_{u_k}}{dt} = g_{k-1,k}(u_{k-1} - u_k) + g_{k,k+1}(u_{k+1} - u_k) \tag{9}$$

with $g_{k-1,k}, g_{k,k+1} \in \mathbb{R}$. Finally, the last variable $u_K$, has no downstream neighbor. Thus, setting $u_{K+1} \leftarrow 0$ produces a natural leak term that induces decay:

$$C_K \frac{d_{u_K}}{dt} = g_{K-1,K}(u_{K-1} - u_K) + g_{K,K+1}(-u_K) \tag{10}$$

Together, the capacity $C_k$ and the flow strength $g_{k,k+1}$ define the continuous timescales of plasticity and stability of each variable $u_k$. To implement these dynamics in RL, which operates in discrete steps, we discretize them with Euler's method.

## 4. Learning Successor Features with Synaptic Consolidation

In line with prior work which discretize synaptic consolidation by replacing synaptic efficacy $v$ with the parameters of a Q-value function (Kaplanis et al., 2018) or a policy (Kaplanis et al., 2019), *we instead apply synaptic consolidation to the parameters of the SFs.* Specifically, each variable $u_k$ is mapped to the corresponding SF parameters $\theta_k \in \mathbb{R}^n$, yielding $\psi_{u_k} = \psi_{\theta_{u_k}} \in \mathbb{R}^n$, where $\psi$ denotes the SFs. For brevity, we will write $\psi_{u_k}$ to denote $\psi_{\theta_{u_k}}$. We next derive the learning rules using Euler's method.

Let $\eta_k = \Delta t / C_k$. There will be two learning phases ($t + \frac{1}{2}$ and $t + 1$) for the most plastic variable, SF $\psi_{u_1}$. At phase

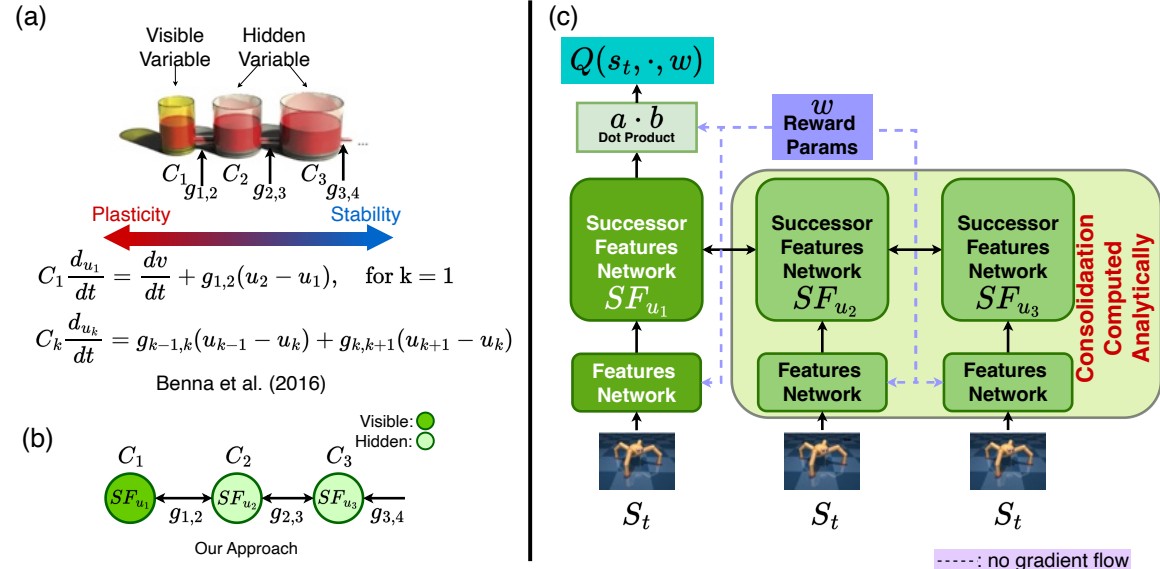

*Figure 2.* **a:** Neuro-inspired synaptic consolidation model adapted from (Benna & Fusi, 2016). The visible variable, $u_1$, represents the synaptic efficacy $v$, while downstream hidden variables $u_2, u_3, ...$ interact bidirectionally across timescales, with beaker capacities $C_1 < C_2 < ..., < C_K$ and tube widths representing flow strength $g_{1,2} > g_{2,3} > ..., > g_{K,K+1}$ controlling the rate of interaction between the variables. Together, the beaker sizes ($C_k$) and flow strength ($g_{k,k+1}$) govern the effective timescales of plasticity and stability. **b:** The synaptic efficacy $v$ is replaced by the parameters of SFs, thus allowing SFs to be learned across different timescales. **c:** Our architectural design. See section 4 for more details on training the system.

$t + \frac{1}{2}$, SF $\psi_{u_1}$ is learned via optimizing Q-SF-TD loss $L_\psi$ (Eq. 6):

$$\psi_{u_1}^{t+\frac{1}{2}} = \psi_{u_1}^t - \alpha \nabla_{\psi_{u_1}} L_{\psi_{u_1}} \qquad (11)$$

where $\alpha \in \mathbb{R}$ is the learning rate. At the second phase, $t + 1$, we update $\psi_{u_1}$ and the rest of the SFs variable $(\psi_{u_2}, \psi_{u_3}, \ldots, \psi_{u_K})$ using the Euler update. For the first variable $\psi_{u_1}$:

$$\psi_{u_1}^{t+1} = \psi_{u_1}^{t+\frac{1}{2}} + \eta_1 \big[ g_{1,2} \big( \psi_{u_2}^t - \psi_{u_1}^{t+\frac{1}{2}} \big) \big] \qquad (12)$$

For the interior variables $k = 2, \ldots, K - 1$:

$$\psi_{u_k}^{t+1} = \psi_{u_k}^t + \eta_k \big[ g_{k-1,k} (\psi_{u_{k-1}}^t - \psi_{u_k}^t) + g_{k,k+1} (\psi_{u_{k+1}}^t - \psi_{u_k}^t) \big] \qquad (13)$$

For the last variable $K$:

$$\psi_{u_K}^{t+1} = \psi_{u_K}^t + \eta_K \big[ g_{K-1,K} (\psi_{u_{K-1}}^t - \psi_{u_K}^t) - g_{K,K+1} (\psi_{u_K}^t) \big] \qquad (14)$$

We provide a pseudocode of the algorithm in Appendix C. It is important that these updates (Eqs. 12, 13 and 14) are performed using Stochastic Gradient Descent (SGD) rather than adaptive approaches like Adaptive Moment Estimation (Adam, (Kingma & Ba, 2014)), which do not preserve the timescales information. We provide a proof sketch in Appendix B supporting this claim.

## 5. Experimental results

In this study, we consider two environments. The first is a slippery variant of the 3D Four Rooms environments, adapted from Chua et al. (2024), which mimics conditions such as walking on wet or icy surfaces. In this environment, the "slippery" event refers to the agent's action being randomly replaced by an alternative, based on a probability value sampled from a noisy sine function to simulate continual dynamics shifts (Figure 10 in Appendix E).

The agent alternates between two tasks, where in the first task it receives a reward of +1 for reaching the green box and -1 for reaching the yellow box, and in the second task the rewards are reversed. The agent cycles through this two-task sequence twice and only receives egocentric pixel observations (Figure 4).

The second environment is the MuJoCo suite (Todorov et al., 2012), using the DeepMind Control Suite (DMC) (Tunyasuvunakool et al., 2020), which provides an accessible framework for modifying dynamics of the embodiments. We focus on four embodiments, Half-cheetah, Walker, Quadruped and Humanoid, ordered by increasing complexity in terms of their observation and action spaces. The agents are rewarded for walking or running forward. To simulate the continual dynamics shifts, at every ten episodes, we perturbed the embodiment's mass by sampling from a noisy sine function. See Appendix D for code availability.

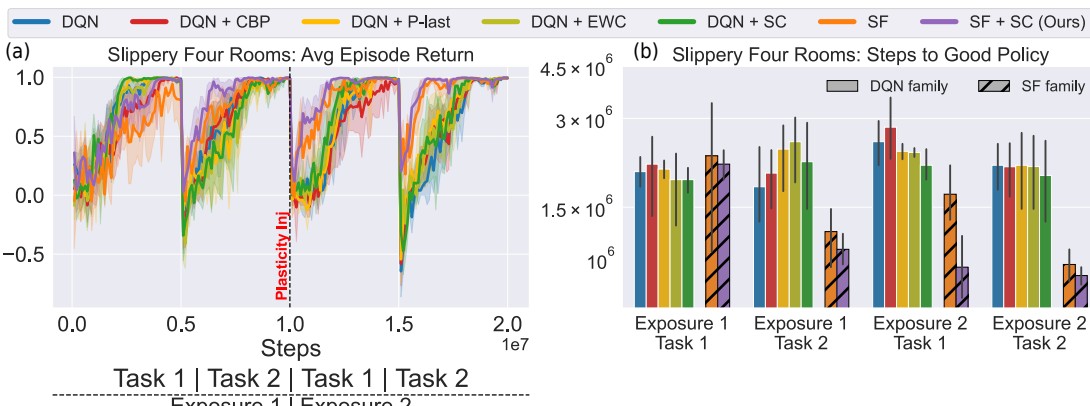

*Figure 3.* Results from Slippery Four Rooms with naturalistic, continual evolving slip dynamics that randomly replace actions. **(a):** Average return across two sequential tasks (Task 1 and 2), each repeated twice (Exposure 1 and 2). In DQN+P-last (yellow), plasticity injection is applied midway through training by randomly re-initializing the last layer's parameters. **(b):** Steps to reach a predefined performance threshold (fewer is better). Overall, stability-preserving methods (EWC, SC) outperform plastic ones (P-last), with further gains from combining SFs with SC (SF+SC).

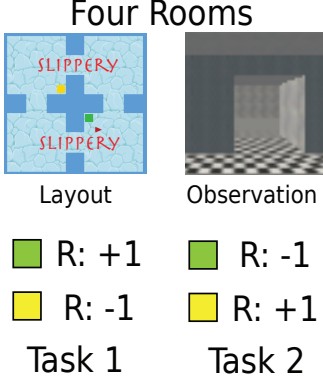

*Figure 4.* Slippery Four Rooms

For base models, we use Double Deep Q-Network (DQN) (Van Hasselt et al., 2016) for Slippery Four Rooms environment and the Deterministic Policy Gradient algorithm (Silver et al., 2014) with twin critics for MuJoCo (TD3) (Fujimoto et al., 2018). For SFs, we use Simple SFs (Chua et al., 2024) which can be learned without auxiliary losses. We selected these models due to their flexibility, which makes extensions with plasticity injections or synaptic consolidation mechanisms rather straightforward.

We compared against baselines that do not require task statistics. For plasticity, we reset subsets of parameters—last layer (DQN+P-last, TD3+P-last) (Nikishin et al., 2023) or least used via continual backprop (TD3+CBP) (Dohare et al., 2024), the latter more effective in MuJoCo. For stability, we used online Elastic Weight Consolidation (EWC) (Schwarz et al., 2018) for Q-values and SFs (DQN+EWC, TD3+EWC, SF+EWC). We also included synaptic consolidation (SC, Figure 2) with Q-value (DQN+SC, TD3+SC) and SF vari-

ants (SF+SC). Results are averaged over 5 seeds.

Our experiments are designed to address the following questions:

1. When agents undergo naturalistic, continual non-stationary shifts, is the primary bottleneck one of plasticity or stability? If stability is the limiting factor, which consolidation mechanism is more effective? EWC or SC?

2. Second, is it more effective to consolidate parameters of Q-value or Successor Features?

3. Third, will SC remain effective under continual non-periodic or stochastic drifts?

## 5.1. Stability as the Bottleneck: Synaptic Consolidation outperforms EWC

To study the question of plasticity versus stability, we first evaluated the performance in the slippery Four Rooms environment using DQN, along with its plasticity injection variants (P-last) and stability-preserving variants (EWC and SC). The results in Figure 3 show that stability-preserving models (DQN + EWC and DQN + SC) consistently outperformed the plasticity-injection model (DQN + P-last), with synaptic consolidation (DQN + SC) achieving higher learning efficiency.

We next evaluated performance in the MuJoCo suite using TD3, together with its two plasticity-injection variants (P-last and CBP) and stability-preserving variants (EWC and SC). The results in Figure 5 show that stability-preserving models (TD3 + EWC and TD3 + SC) consistently outperformed the plasticity-injection model (TD3 + P-last and

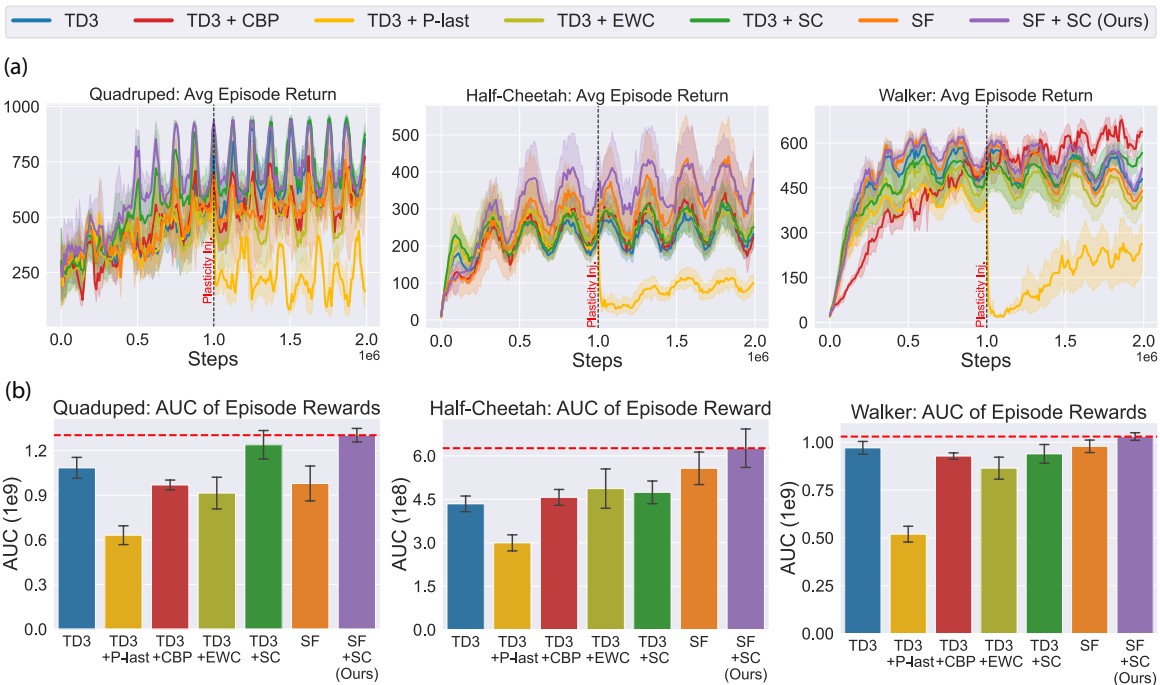

*Figure 5.* Results from MuJoCo, where agent embodiments undergo continuous mass changes during training. **(a):** Average episode return. Plasticity injection for TD3+P-last (yellow) was performed halfway through the training. **(b):** Area under the Curve (AUC) of returns in (a). Together, the plots show that stability-preserving methods (EWC and SC) outperform plastic ones (CBP and P-last), with further gains achieved by combining synaptic consolidation with SFs (SF+SC). Results for Humanoid can be seen in Figure 1(c-d).

TD3 + CBP). Once again, synaptic consolidation (TD3 + SC) achieved higher learning efficiency, particularly in the more complex embodiments, Humanoid (Figure 1) and Quadruped (Figure 5a and b).

Both sets of results indicate that *stability is the primary bottleneck* as agents lacking stability fail to learn effectively in the naturalistic, continual non-stationary environments. The results further showed that the synaptic consolidation (SC) mechanism is more effective than EWC.

Next, we asked, what exactly should be stabilized: the Q-value parameters themselves, or the parameters of the underlying representations such as SFs?

### 5.2. What should be stabilized: Q-values or Successor Features?

As many stability preserving methods focus on stabilizing parameters of Q-value functions, we investigate if SFs could be a better target for consolidation. To address this, we evaluated a SFs variant combined with synaptic consolidation (SF + SC) in both slippery Four Rooms and the MuJoCo suite. The results for both slippery Four Rooms (Figure 3) and the MuJoCo suite (Figures 1 and 5) showed that SF + SC consistently improves performance compared to Q-value based consolidation. We also compared with EWC (Appendix K), which revealed that while SC is more effec-

tive than EWC overall, only the combination of SFs with SC yields consistently strong performance.

### 5.3. Quantifying Non-Stationarity via Mass Perturbations

To systematically study how the magnitude of environmental change affects learning dynamics, we parameterize non-stationarity through controlled perturbations of body mass. Specifically, we consider three regimes of variation—mild (25%), moderate (50%), and severe (100%)—defined relative to the maximum perturbation before the MuJoCo simulation becomes unstable.[1]

Across Humanoid (Figure 6) and other embodiments, we observe a clear dependence on the degree of non-stationarity. Under mild variation, where the transition dynamics remain close to stationary, applying synaptic consolidation (SC) to Q-values is most effective. In contrast, as the magnitude of mass perturbation increases, SC applied to Successor Features (SFs) becomes increasingly advantageous, particularly under moderate and severe regimes where the environment undergoes substantial and continuous changes.

---

[1]Unless otherwise stated, the main results presented correspond to the severe (100%) setting.

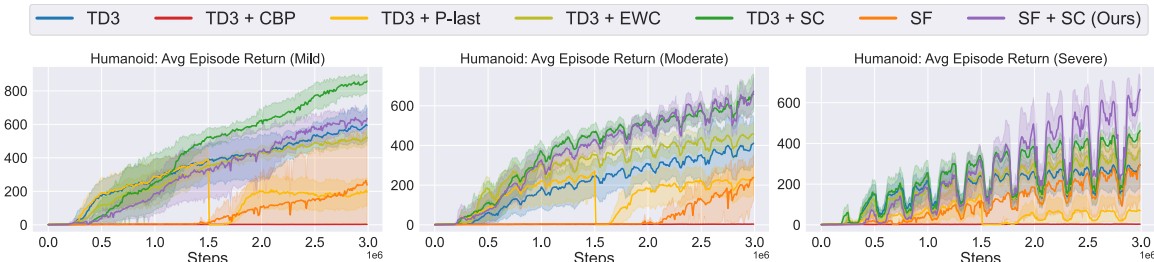

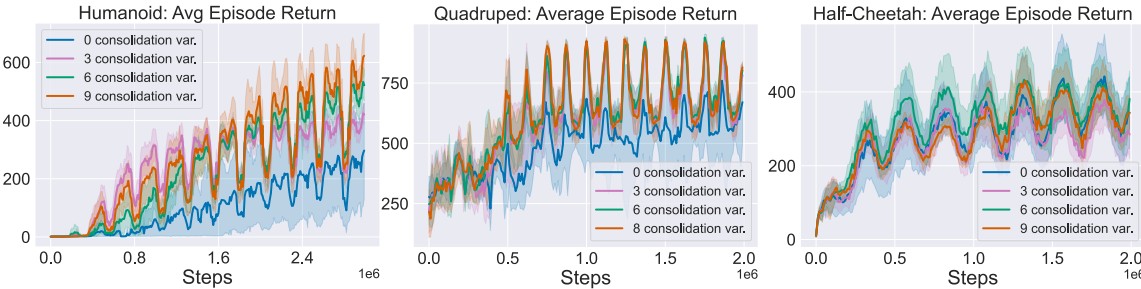

*Figure 6.* Quantification of mass changes for the Humanoid embodiment. We consider three levels of mass dynamics variation: mild (25%), moderate (50%), and severe (100%), corresponding to the maximum change allowed before the physical simulation becomes unstable. Across these settings, plasticity-preserving methods (CBP, P-last) are less effective than approaches incorporating synaptic consolidation (SC). Under moderate—and especially severe—conditions, where the agent experiences substantial changes in dynamics, applying SC to Successor Features (SF + SC) yields the best performance. In contrast, under mild changes, the benefits of SC are limited, as the environment remains close to stationary. For other results, please refer to Appendix T.

*Figure 7.* Analysis of timescales in MuJoCo using our model, SF + SC. More consolidation variables (6–9) improve learning efficiency, highlighting the benefit of slower timescales. Zero variables correspond to the Simple SF agent. See Appendix M for results in the Slippery Four Rooms environment.

### 5.4. Robustness to Non-Periodic and Stochastic Drift

To test robustness beyond periodic drift, we replace the noisy sine modulation with either a non-periodic sine function (Appendix E.4) or Ornstein–Uhlenbeck (OU) process (Appendix E.6). Results from mass changes induced by a non-periodic sine function (Appendix H) and OU process (Appendix I) show that applying SC to SFs continues to improve learning performance.

## 6. Analyzing Multi-Timescale Contributions

To gain insights into why combining SFs and SC yields an effective model, we perform ablation studies by varying the number of consolidation variables, and we complement this with a cross-attention (Dosovitskiy et al., 2020) analysis to analyze the relative contributions of individual variables.

### 6.1. Do fast or slow timescale variables matter more for learning?

In this analysis, we varied over the number of consolidation variables (3, 6, or 9), with fewer variables yielding greater plasticity, and more variables yielding greater stability. The aim was to assess whether the inclusion of slower

timescale variables improve policy learning. Figure 7 illustrates the results using the embodiments from MuJoCo (see Appendix M for more results). We found that using six or more timescale variables leads to better learning performance. Zero variables correspond to the Simple SF agent. For the slippery four rooms environment (Figure 29 in Appendix 6.1), we observed a similar trend: using six or nine consolidation variables leads to better performance. Together, these results demonstrated that preserving stability through the use of slow timescales is crucial in our settings.

### 6.2. What does cross-attention reveal about the contributions of consolidation variables?

While varying over the number of consolidation variables provides a coarse measure of their utility, it does not reveal which specific variables contribute most. At the same time, using a cross-attention mechanism prevents the need for information to propagate gradually via the flow strength ($g_{k,k+1}$), instead providing an instant readout from all the consolidation variables (see Figure 8a).

We adapt the cross-attention mechanism by letting the reward weight vector $w$ serves as the query, while the SF consolidation variables (excluding the most plastic $SF_{u_1}$)

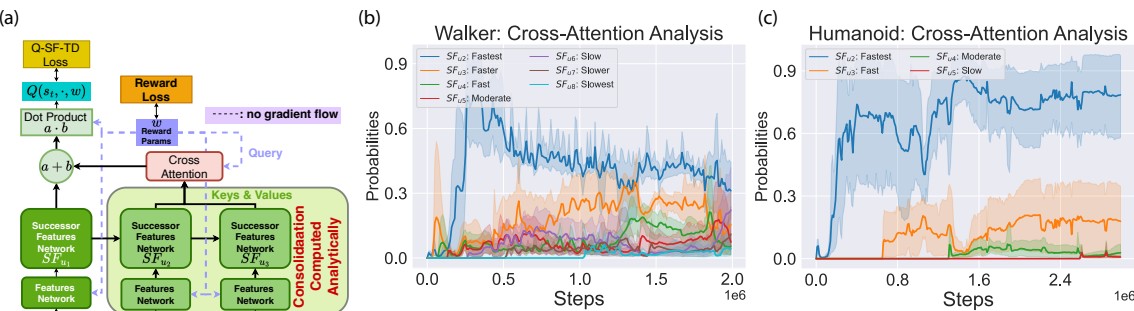

*Figure 8.* Cross-Attention analysis of individual consolidations, replacing memory recall via backflow. **(a):** Implementation design. **(b-c):** Attention probabilities over consolidation variables, where higher probability indicates greater contribution to learning. Full results are provided in Appendix O.

serve as keys and values. The softmax probabilities from the cross-attention mechanism (Figure 8b and c) showed that the faster timescale variables, particularly $SF_{u_2}$ and $SF_{u_3}$, received the highest attention. However, slower variables also contributed. Together, these findings suggest that fast timescales drive most learning, while slower ones provide complementary stability.

### 6.3. Can Larger Networks Replace Multi-Timescale Consolidation?

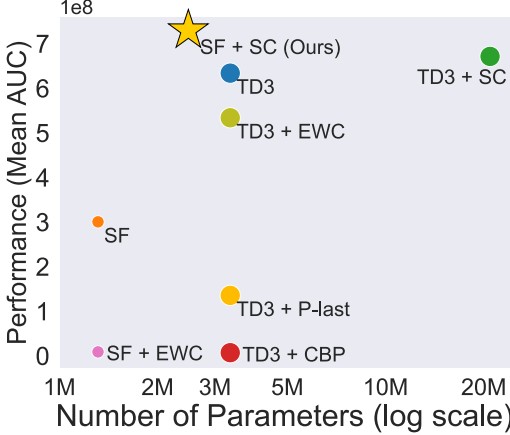

*Figure 9.* Capacity analysis using Humanoid. X-axis shows the number of parameters, while the y-axis shows performance measured by area under the curve (AUC). Increasing the parameter count of TD3 and its variants did not consistently improve performance compared to SF + SC (star), suggesting the contribution of consolidating SFs beyond network capacity scaling alone.

Since each consolidation variable introduces an additional set of parameters, improvements from synaptic consolidation could potentially be attributed to increased model capacity rather than the consolidation dynamics themselves. To control for this possibility, we compare against enlarged baseline networks (TD3, TD3 + CBP, TD3 + EWC, TD3 + P-last) with parameter counts exceeding those of the SF-based models (SF, SF + EWC, SF + SC).

The results in Figure 9 show that, despite substantially increasing their capacity, TD3 and its variants fail to match the performance of SF + SC, which achieves the strongest performance while using significantly fewer parameters. This suggests that the gains do not arise solely from increased model size, but rather from the combination of predictive representations and multi-timescale consolidation dynamics. From a scalability perspective, these findings further indicate that our approach is parameter-efficient and capable of achieving strong continual learning performance.

## 7. Discussion

In this work, we introduced naturalistic continual non-stationarity benchmarks to study the stability–plasticity trade-off in deep RL. All our experimental results suggest that stability is often the primary bottleneck, and that maintaining plasticity (e.g., reset-based methods) is insufficient. It is important to note that we do not directly measure plasticity, and evaluate adaptation through learning performance.

We find that combining SC with SFs yields a multi-timescale learning system in which slower components preserve stable predictive structure while faster components enable rapid adaptation. Cross-attention analysis indicates that different timescales contribute differently to behavior. Importantly, this stability–plasticity balance does not emerge from policy constraints alone. Compared with Proximal Policy Optimization (PPO) (Schulman et al., 2017), which relies on trust-region updates to stabilize learning, SFs combined with SC exhibited substantially greater robustness under continuous non-stationarity (Appendix S).

However, several limitations remain. First, the method is restricted to SGD-based updates, as combining it with adaptive optimizers like Adam can lead to instability. Second, it introduces computational overhead due to non-parallelizable analytical updates, which grows with the number of timescales. More extreme settings such as multi-agents remain interesting directions for future research.

## Impact statement

Our studies primarily revolve around navigation tasks and embodiments simulations, with the techniques developed being most pertinent to the field of robotics. Given this specific focus, the broader societal implications of our work are likely to be quite limited.

## Acknowledgments

We would like to express our deepest gratitude to Christos Kaplanis for his tremendous support throughout this project. From its earliest stages, Christos generously shared his expertise, insights, and experience on applying synaptic consolidation mechanisms to value and policy networks, building on his prior work (Kaplanis et al., 2018; 2019). His guidance and discussions played an important role in shaping many of the ideas explored in this paper, and we are sincerely grateful for his encouragement and collaboration throughout this journey.

We would also like to thank Guillaume Lajoie, Razvan Pascanu, Claudia Clopath, Rui Ponte Costa, Marcus Benna, and Stefano Fusi for insightful discussions on continual reinforcement learning and the role synaptic consolidation can play in supporting continual adaptation and stability.

We would also like to thank Paul Masset, Isabeau Prémont-schwarz, Nanda Harishankar Krishna, Roy Henha Eyono, Hafez Ghaemi, and Maren Wehrheim for their thoughtful feedback and for reviewing earlier drafts of the manuscript.

We are also grateful for the wonderful research community at Mila[2], McGill University, and in Montréal for fostering an inspiring and collaborative environment that motivated us to pursue ambitious and interdisciplinary research at the intersection of Neuroscience and Artificial Intelligence (NeuroAI).

Raymond Chua was supported by the DeepMind Graduate Award and UNIQUE Excellence Scholarship (PhD). We extend our gratitude to the FRQNT Strategic Clusters Program (2020-RS4-265502 - Centre UNIQUE - Quebec Neuro-AI Research Center).

Blake A. Richards was supported by NSERC (Discovery Grant RGPIN-2020- 05105, RGPIN-2018-04821; Discovery Accelerator Supplement: RGPAS-2020-00031), CIFAR (Canada AI Chair; Learning in Machine and Brains Fellowship), and by funds provided by the National Science Foundation and DoD OUSD (R&E) under Cooperative Agreement DBI-2229929 (The NSF AI Institute for Artificial and Natural Intelligence).

This research was also enabled by computational resources provided by Calcul Québec [3] and the Digital Research Alliance of Canada [4]. The authors acknowledge the material support of NVIDIA in the form of computational resources.

Last but not least, we are also grateful to the anonymous reviewers whose insightful comments and suggestions significantly enhanced the quality of this manuscript.

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

# A. Appendix

This supplementary section provides detailed insights and additional information that supports the findings and methodology discussed in the main paper. Below is a brief overview of what each section contains:

| Appendix Section |
| --- |
| Appendix B: Proof Sketch on preserving timescales with SGD |
| Appendix C: Pseudocode Implementation |
| Appendix D: Code Availability |
| Appendix E: Environments |
| Appendix F: Experimental Details |
| Appendix G: Plasticity-Stability Analysis in non-stationary conditions induced by a periodic noisy sinusoidal function |
| Appendix H: Plasticity-Stability Analysis in non-stationary conditions induced by a non-periodic noisy sinusoidal function |
| Appendix I: Plasticity-Stability Analysis in non-stationary conditions induced by Ornstein–Uhlenbeck processes |
| Appendix J: Schematic of Synaptic Consolidation for Q-values vs. Successor Features |
| Appendix K: Q-values vs. SFs, Elastic Weight Consolidation vs Synaptic Consolidation Comparison |
| Appendix L: Q-values vs. SFs, With and Without Synaptic Consolidation Comparison |
| Appendix M: Analysis of Fast and Slow Timescale Variables |
| Appendix N: Architecture for recalling consolidated Successor Features using Cross-Attention |
| Appendix O: Cross-Attention Analysis of Fast and Slow Timescale Variables |
| Appendix P: Agents |
| Appendix Q: Implementation details |
| Appendix R: Computational Complexity |
| Appendix S: Comparison with Trust-Region Methods (PPO) |
| Appendix T: Quantifying Continuous Non-Stationarity |

# B. Proof Sketch on preserving timescales with SGD

In this section, using mathematical analysis, we provide the intuition why the learning of the multiple-timescales Successor Features (SFs) must be done using Stochastic Gradient Descent (SGD) and not the commonly used Adaptive moment estimation (Adam) (Kingma & Ba, 2014).

For the sake of brevity, we consider a tabular Reinforcement Learning setting where the SFs $\psi(s, a)$ are not parameterised and depends only on a state-action pair $(s, a)$:

$$\psi(s, a) = \mathbb{E}_\pi \left[ \sum_{t=0}^{\infty} \gamma^t \phi(s_t, a_t) \mid s_0 = s, a_0 = a \right] \tag{15}$$

where $\gamma \in [0, 1]$ is the discount factor and $\phi$ is the basis features. Without loss of generality, we consider an arbitrary consolidation system with $K \in \mathbb{Z}$ possible consolidation variables operating at $K$ possible timescales, where $K > 1$. Let $u_k$ be a variable and $\psi_{u_k} \in \mathbb{R}^n$ be the SFs operating at timescale $k \in (1, 2, ..., K)$. The terms $g_{k-1,k}/C_k, g_{k,k+1}/C_k$, where $C_k = 2^{k-1}$ and $g_{k,k+1} \propto 2^{-k-2}$ determines the overall timescales of learning the SFs $\psi_{u_k}$.

Recall that after applying the Euler's method to the continuous dynamics (section 3.3), we get the following update rules for the consolidation variables $(\psi_{u_1}, \psi_{u_2}, \ldots, \psi_{u_K})$. Let $\eta_k = \Delta t / C_k$ and ignoring the first learning phase $t + 1/2$, at step $t + 1$, for the first variable $\psi_{u_1}$, we get:

$$\psi_{u_1}^{t+1} = \psi_{u_1}^{t+\frac{1}{2}} + \eta_1 [g_{1,2} (\psi_{u_2}^t - \psi u_1^{t+\frac{1}{2}})] \tag{16}$$

For the interior variables $k = 2, \ldots, K - 1$:

$$\psi_{u_k}^{t+1} = \psi_{u_k}^t + \eta_k [g_{k-1,k}(\psi_{u_{k-1}}^t - \psi_{u_k}^t) + g_{k,k+1}(\psi_{u_{k+1}}^t - \psi_{u_k}^t)] \tag{17}$$

For the last variable $K$:

$$\psi_{u_K}^{t+1} = \psi_{u_K}^t + \eta_K \left[ g_{K-1,K}(\psi_{u_{K-1}}^t - \psi_{u_K}^t) - g_{K,K+1}(\psi_{u_K}^t) \right] \tag{18}$$

Without loss of generality, we consider the case of updating $\psi_{u_1}$ in Eq. 16, where $u_1$ and $u_2$ represent the first and second consolidation variables. Let $\Delta t = 1$ and $\kappa_{1,2}$ be the timescale ratio $\frac{g_{1,2}}{C_1}$, the update term for $\psi_{u_1}^{t+1}$, corresponding to:

$$\eta_1 \left[ g_{1,2}\left( \psi_{u_2}^t - \psi_{u_1}^{t+\frac{1}{2}} \right) \right] = \frac{g_{1,2}}{C_1} \big( \underbrace{\psi_{u_2}^t - \psi_{u_1}^{t+\frac{1}{2}}}_{:=g_t} \big) \tag{19}$$

$$= \kappa_{1,2} \odot g_t \tag{20}$$

$$= \tilde{g}_t \tag{21}$$

In optimization, we usually use a learning rate, such as $\alpha \in \mathbb{R}$ to update the variables.

As we shall see in the proof sketch below, both $\kappa_{1,2} \in \mathbb{R}$ and $\alpha \in \mathbb{R}$ contribute to the effective learning rate. We will first present our analysis using SGD, followed by Adam.

## B.1. Stochastic Gradient Descent (SGD)

Recall that the update rule for SGD at step $t$ is defined as:

$$\psi_{t+1}(s,a) \leftarrow \psi_t(s,a) - \alpha \odot \tilde{g}_t \tag{22}$$

**Proposition B.1.** *Optimizing Eq. 21 using Stochastic Gradient Descent (SGD) ensures that the gradient updates explicitly scales with $\alpha$, thus preserving the relative timescale information.*

*Proof.* Without loss of generality, let $\kappa \in \{\kappa_{1,2}, \kappa_{2,3}, ..., \kappa_{K,K+1}\}$:

$$\psi_{t+1}(s,a) \leftarrow \psi_t(s,a) - \alpha \tilde{g}_t \tag{23}$$

$$= \psi_t(s,a) - \alpha(\kappa \cdot g_t) \qquad \text{Sub } \tilde{g}_t = \kappa \cdot g_t \text{ from Eq.20} \tag{24}$$

$$= \psi_t(s,a) - (\alpha \cdot \kappa \cdot g_t) \tag{25}$$

It is then trivial to conclude that when the learning rate $\alpha = 1$, the effective learning rate $\alpha \cdot \kappa \cdot g_t = \kappa \cdot g_t$, thus preserving the relative scale of the updates, even when the timescale ratio $\kappa$ decreases as we move down the chain of dynamic variables due to the fact that $\kappa_{1,2} >> \kappa_{2,3} >> ..., \kappa_{K,K+1}$. $\square$

## B.2. Adaptive moment estimation (Adam)

Recall that the update rule for Adam (Kingma & Ba, 2014) at step $t$ is defined as:

$$m_t \leftarrow \beta_1 \cdot m_{t-1} + (1 - \beta_1) \cdot \tilde{g}_t \qquad \text{First moment} \tag{26}$$

$$v_t \leftarrow \beta_2 \cdot v_{t-1} + (1 - \beta_2) \cdot \tilde{g}_t^{\,2} \qquad \text{Second moment} \tag{27}$$

$$\hat{m}_t \leftarrow \frac{m_t}{(1 - \beta_1^t)} \qquad \text{Bias correction for first moment} \tag{28}$$

$$\hat{v}_t \leftarrow \frac{v_t}{(1 - \beta_2^t)} \qquad \text{Bias correction for second moment} \tag{29}$$

$$\psi_{t+1}(s,a) \leftarrow \psi_t(s,a) - \frac{\alpha}{\sqrt{\hat{v}_t} + \epsilon} \cdot \hat{m}_t \tag{30}$$

where $\frac{\alpha}{\sqrt{\hat{v}_t} + \epsilon}$ is the effective learning rate and $\tilde{g}_t$ is the update term as defined in Eq. 21.

**Proposition B.2.** *Optimizing Eq. 21 using Adam results in gradient updates to not preserve the relative timescale information.*

Let's focus our analysis on the effective learning rate $\frac{\alpha}{\sqrt{\hat{v}_t}+\epsilon}$ and once again, without the loss of generality, let $\kappa \in \{\kappa_{1,2}, \kappa_{2,3}, ..., \kappa_{K,K+1}\}$:

$$
\begin{aligned}
\frac{\alpha}{\sqrt{\hat{v}_t}+\epsilon} &= \frac{\alpha}{\sqrt{\frac{v_t}{(1-\beta_2^t)}}+\epsilon} && \text{Sub Eq. 29 into } \hat{v}_t \\[2ex]
&= \frac{\alpha}{\sqrt{\frac{\beta_2 \cdot v_{t-1}+(1-\beta_2)\cdot \tilde{g}_t^{\,2}}{(1-\beta_2^t)}}+\epsilon} && \text{Sub Eq. 27 into } \hat{v}_t \\[2ex]
&= \frac{\alpha}{\sqrt{\frac{\beta_2 \cdot v_{t-1}+(1-\beta_2)\cdot (\kappa \cdot g_t)^2}{(1-\beta_2^t)}}+\epsilon} && \text{Sub } \tilde{g}_t = \kappa \cdot g_t \text{ from Eq.21}
\end{aligned}
$$

$$(31)$$

We can observe that when the learning rate $\alpha = 1$, the effective learning rate, $\frac{1}{\sqrt{\hat{v}_t}+\epsilon}$, increases as the timescale ratio variable $\kappa$ decreases. This is due to the fact that as we move down the chain of consolidation variables, we will get $\kappa_{1,2} >> \kappa_{2,3} >> ..., \kappa_{K,K+1}$. This implies that the timescale will no longer be preserved, as the relative scale of the updates will now become inversely proportional with respect to $\kappa$ rather than proportional to $\kappa$. $\square$

### B.3. Conclusion

These analyses demonstrate that learning the Successor Features (SFs), $\psi_{u_k}$, using Stochastic Gradient Descent (SGD) rather than Adam is critical for preserving the timescale information intrinsic to these features. The differential impact of SGD and Adam on the behavior of the updates highlights the importance of choosing an appropriate optimization strategy in RL settings that require maintenance of structured timescale information.

## C. Pseudocode Implementation

In this section, we present the pseudocode of our model (SF + SC), where we apply the synaptic consolidation mechanism (Benna & Fusi, 2016) to Successor Features.

---

**Algorithm 1** Learning Successor Features with Synaptic Consolidation

---

1: Determine the number of consolidation variables $(\psi_{u_2}, \psi_{u_3}, \ldots)$
2: Initialize reward weight vector $\boldsymbol{w}$
3: Initialize SF $\psi_\theta$ network, SF $\overline{\psi_\theta}$ target network
4: Set $\psi_{\theta_{u_1}} \leftarrow \psi_\theta$
5: Copy $\theta_{u_1}$ to the networks of the consolidation variables (e.g., $\psi_{\theta_{u_2}} \leftarrow \theta_{u_1}, \psi_{\theta_{u_3}} \leftarrow \theta_{u_1}, \ldots$)
6: **for** $t := 1, \text{T}$ **do**
7: $\quad$ Receive observation $S_t$ from environment
8: $\quad$ $A_t \leftarrow \epsilon$-greedy using $Q(S_t, \cdot \mid \boldsymbol{w})$
9: $\quad$ Send $A_t$ to receive $S_{t+1}$ and $R_{t+1}$ from environment
10: $\quad$ $a' \in \underset{b}{\operatorname{argmax}} \, \overline{\psi_\theta}(S_{t+1}, b, \boldsymbol{w})^\top \boldsymbol{w}$
11: $\quad$ $\hat{y} = R_{t+1} + \gamma \overline{\psi_\theta}(S_{t+1}, a', \boldsymbol{w})^\top \boldsymbol{w}$
12: $\quad$ $\phi \leftarrow$ L2 normalized output from the encoder of SF $\psi$ network
13: $\quad$ $loss_{\psi_\theta} = (\psi_\theta(S_t, A_t, \boldsymbol{w})^\top \boldsymbol{w} - \hat{y})^2$
14: $\quad$ $loss_w = (\phi^\top \boldsymbol{w} - R_{t+1})^2$
15: $\quad$ Gradient descent on $\psi_\theta$ and $\boldsymbol{w}$
16: $\quad$ Set $\psi_{\theta_{u_1}} \leftarrow \psi_\theta$
17: $\quad$ Update the parameters of the consolidation parameters analytically using Eq.12, Eq. 13, Eq.14 (Stochastic Gradient Descent)
18: $\quad$ Set $\psi_\theta \leftarrow \psi_{\theta_{u_1}}$
19: **end for**

---

## D. Code Availability

The PyTorch implementation used for the MuJoCo experiments, together with training scripts, hyperparameter configurations, and instructions for reproducing the results, is publicly available.[5]

The repository includes:

- implementations of all methods evaluated in the MuJoCo experiments;

- continual non-stationarity benchmarks for MuJoCo;

- training and evaluation scripts;

- hyperparameter configurations used in the paper;

- instructions for reproducing the reported results.

The JAX implementation used for the 3D Four Rooms experiments is also publicly available.[6] This repository contains the 3D Four Rooms continual non-stationarity experiments together with the corresponding training, evaluation, and reproducibility code.

---

[5]https://github.com/raymondchua/multi-timescale-successor-features-mujoco
[6]https://github.com/raymondchua/multi-timescale-successor-features-fourrooms

# E. Environments

In this section, we present the two environments used throughout this manuscript: the Slippery 3D Four Rooms environment and the MuJoCo control suite, along with the corresponding forms of continuous non-stationarity introduced in each setting. In the Slippery 3D Four Rooms environment, slippery probabilities were varied according to a periodic noisy sine function. In the MuJoCo control suite, we considered a broader range of continuous changes to the embodiment dynamics by perturbing the embodiment mass using periodic noisy sine functions, non-periodic variants, and Ornstein–Uhlenbeck processes.

## E.1. Slippery 3D Four Rooms Environment (Periodic)

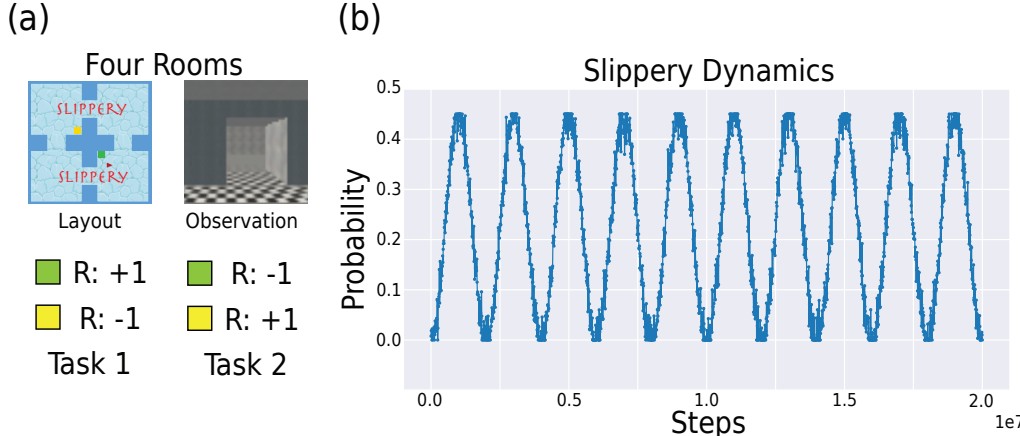

*Figure 10.* **(a)** The slippery variant of the 3D Four Rooms environment. The agent alternates between two tasks: in Task 1, reaching the green box produces +1 reward and the yellow box -1, in Task 2, the reward assignment is reversed. At each step, the agent's chosen action may be randomly replaced with a probability sampled from the noisy sine function shown in B. The agent receives only egocentric pixel observations. **(b)** A noisy sine wave that generates continuously varying slip probabilities, used to stochastically replace the agent's intended actions.

## E.2. Continuous Control in MuJoCo (Periodic)

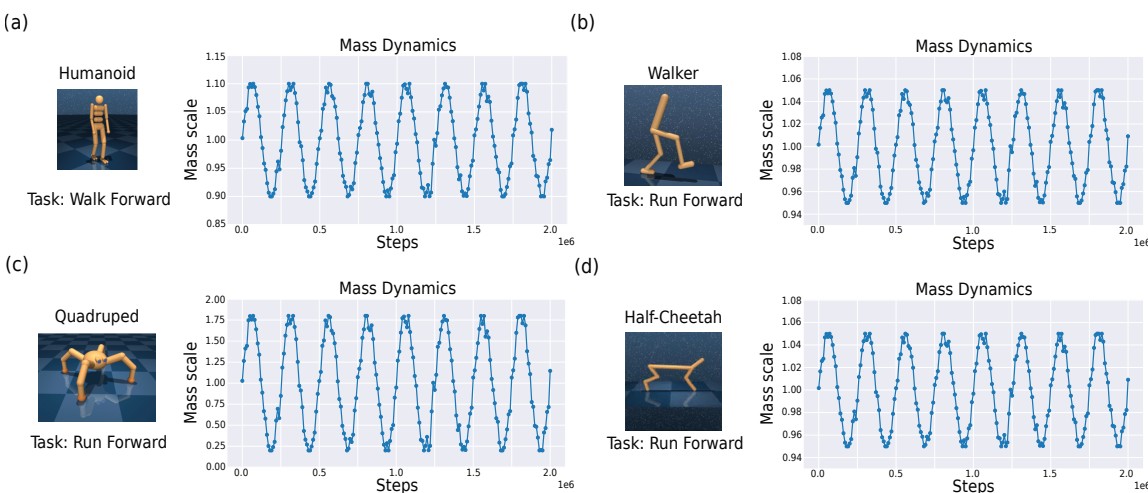

*Figure 11.* MuJoCo suite with continuous periodic mass changes during training and evaluation for (a) Humanoid, (b)Walker, (c) Quadruped, (d) Half-cheetah. A periodic noisy sine wave that generates continuously varying mass values, used to stochastically scale the agent's mass during training and evaluation.

### E.3. Slippery 3D Four Rooms Environment (Non-Periodic)

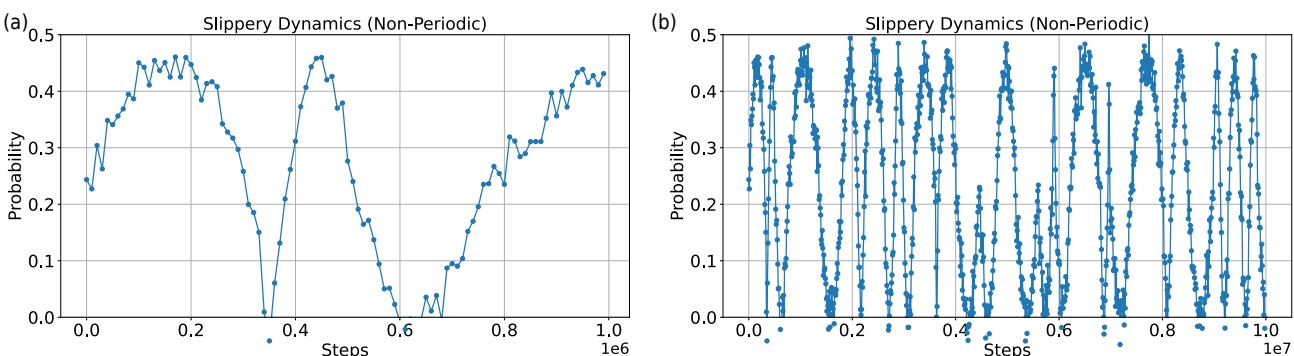

*Figure 12.* **(a)** Partial view over the first 1 million environment steps and **(b)** complete view over the full 10 million environment training steps of the noisy non-periodic sine wave used to generate continuously varying slip probabilities that stochastically replace the agent's intended actions.

### E.4. Continuous Control in MuJoCo (Non-Periodic)

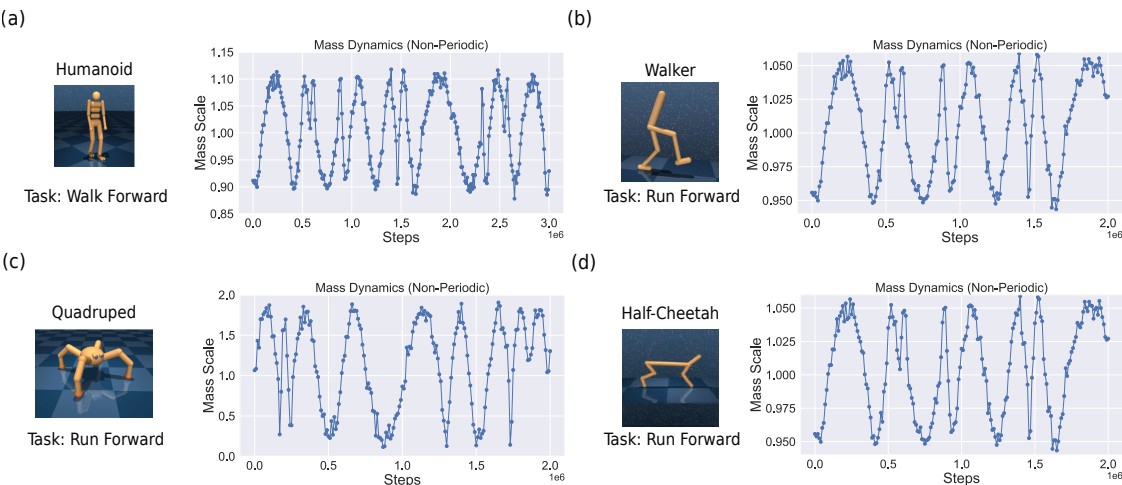

*Figure 13.* MuJoCo suite with continuous non-periodic mass changes during training and evaluation for (a) Humanoid, (b)Walker, (c) Quadruped, (d) Half-cheetah. A non-periodic noisy sine wave that generates continuously varying mass values, used to stochastically scale the agent's mass during training and evaluation.

### E.5. Slippery 3D Four Rooms Environment (Ornstein–Uhlenbeck processes)

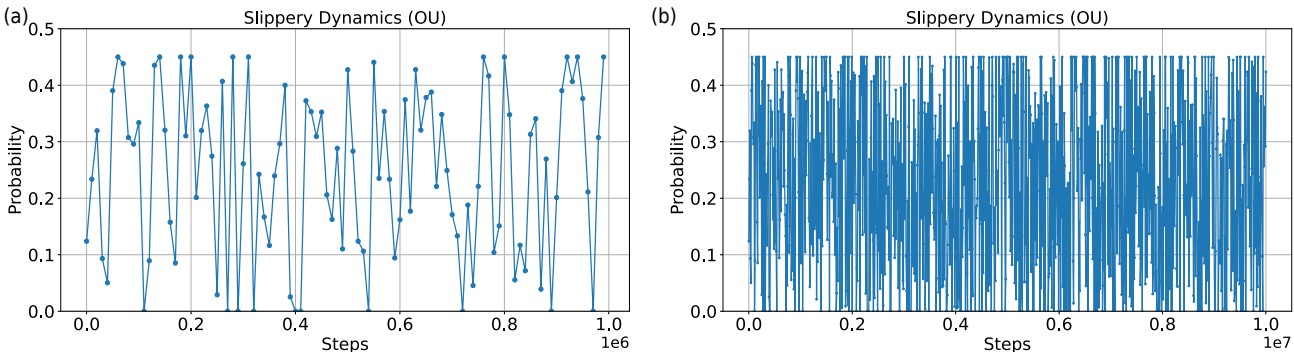

*Figure 14.* **(a)** Partial view over the first 1 million environment steps and **(b)** complete view over the full 10 million training steps of the Ornstein–Uhlenbeck (OU) process used to generate varying slip probabilities that stochastically replace the agent's intended actions.

### E.6. Continuous Control in MuJoCo (Ornstein–Uhlenbeck processes)

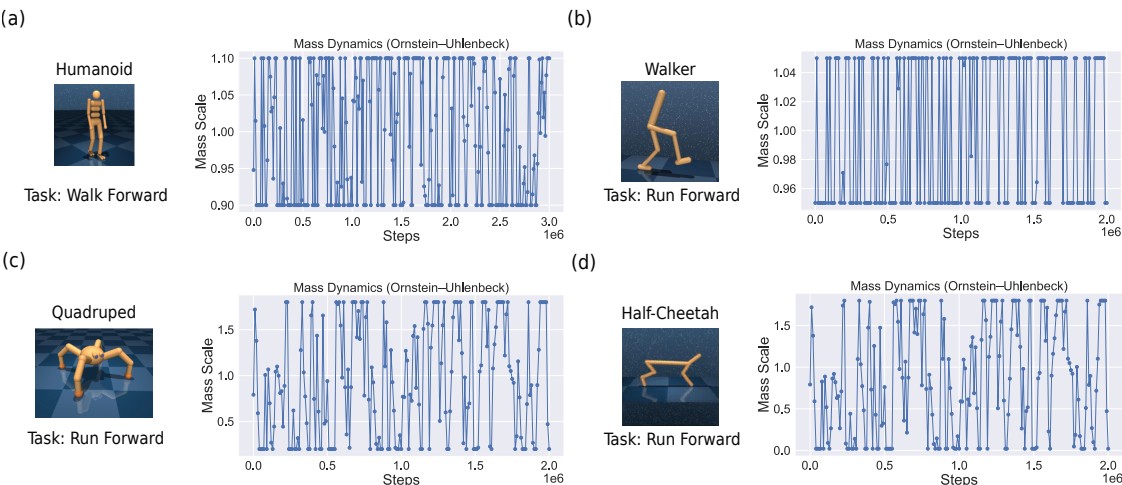

*Figure 15.* MuJoCo suite with continuous mass changes induced using Ornstein–Uhlenbeck(OU) processes during training and evaluation for (a) Humanoid, (b)Walker, (c) Quadruped, (d) Half-cheetah. An Ornstein–Uhlenbeck processes that generate continuously varying mass values, used to stochastically scale the agent's mass during training and evaluation.

# F. Experimental details

In this section, we provide more details about the environments used in our experiments.

## F.1. 3D Slippery Four Rooms Environment

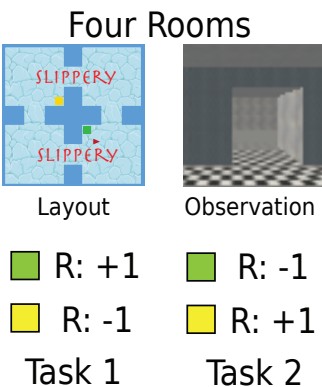

*Figure 16.* Slippery Four Rooms environment

We extended this environment used in the Simple Successor Features (Chua et al., 2024) which was built upon the original 3D Miniworld environment (Chevalier-Boisvert et al., 2023). In the slippery variant of the Four Rooms environment[7], we mimic wet or icy conditions in all four rooms, rather than just two rooms (top right and bottom left). The key difference is, unlike in the prior setup which uses a constant pre-determined slippery probability value, we sample the slippery probabilities using a noisy sine function to ensure continuous changes during training and evaluation. We kept the same two tasks structure, where the rewards alternate when the task switches. The agent receives egocentric pixel observations at every step and the actions are moving forward, backwards, turning left and turning right.

The specific parameters defining the 3D Slippery Four Rooms Environment are detailed in Table 1.

*Table 1.* 3D Slippery Four Rooms Environment Specific Parameters

| PARAMETER | VALUE |
| --- | --- |
| OBSERVATION TYPE | EGOCENTRIC |
| FRAME STACKING | NO |
| RGB OR GREYSCALING | RGB |
| NUM TRAINING FRAMES PER TASK | 5 MILLION FRAMES |
| NUM EXPOSURE | 2 |
| NUM TASK PER EXPOSURE | 2 |
| BATCH SIZE | 32 |
| $\epsilon$ DECAY | 1 MILLION FRAMES |
| ACTION REPEAT | NO |
| ACTION DIMENSION | 4 |
| OBSERVATION SIZE | $84 \times 84$ |
| MAX FRAMES PER EPISODE | 4000 |
| TASK LEARNING RATE | 0.001 |
| SLIPPERY PROBABILITY RANGE | 0 TO 0.45 |

## F.2. MuJoCo

In this work, we consider only state observations. For the embodiments, we chose both Walker, Half-Cheetah, Quadruped and Humanoid. We broadly follow the same setup as Yarats et al. (2021) and Chua et al. (2024), and include their models as baselines, which we denote as "TD3" and "SF" respectively.

---

[7]https://github.com/raymondchua/miniworld_four_rooms

The codebase was adapted from the Simple Successor Features repository[8]. The specific parameters we used in the MuJoCo environment for training broadly follow the ones defined in Chua et al. (2024), with some exceptions, such as the training steps and the learning rate for the reward weight vector. The specific parameters for MuJoCo are detailed in Table 2.

*Table 2.* MuJoCo Environment Specific Parameters

| PARAMETER | VALUE |
|---|---|
| FRAME STACKING | YES |
| OBSERVATION | STATE |
| NUM TRAINING STEPS PER TASK | 2 MILLION STEPS |
| NUM EXPOSURE | 1 |
| ACTION REPEAT | 1 |
| BATCH SIZE | 1024 |
| FEATURE DIM | 128 |
| HIDDEN DIM | 1024 |
| MAX STEPS PER EPISODE | 10000 |
| SF DIM | 10 |
| TASK LEARNING RATE | HALF-CHEETAH: 1E-3, WALKER: 1E-8, QUADRUPED: 1E-9, HUMANOID:1E-8 |
| TASK UPDATE FREQUENCY | 10 |

---

[8]https://github.com/raymondchua/simple_successor_features

# G. Plasticity-Stability Analysis in non-stationary conditions induced by a periodic noisy sinusoidal function

In this section, we present the experimental results for the various MuJoCo embodiments undergoing naturalistic, continuous changes sampled from a periodic noisy sinusoidal function. Results for the Slippery Four Rooms Environment are presented in Figure 3 in the main manuscript. See Appendix E, specifically Sections E.1 and E.2, for illustrations of the resulting dynamics.

## G.1. MuJoCo suite Results

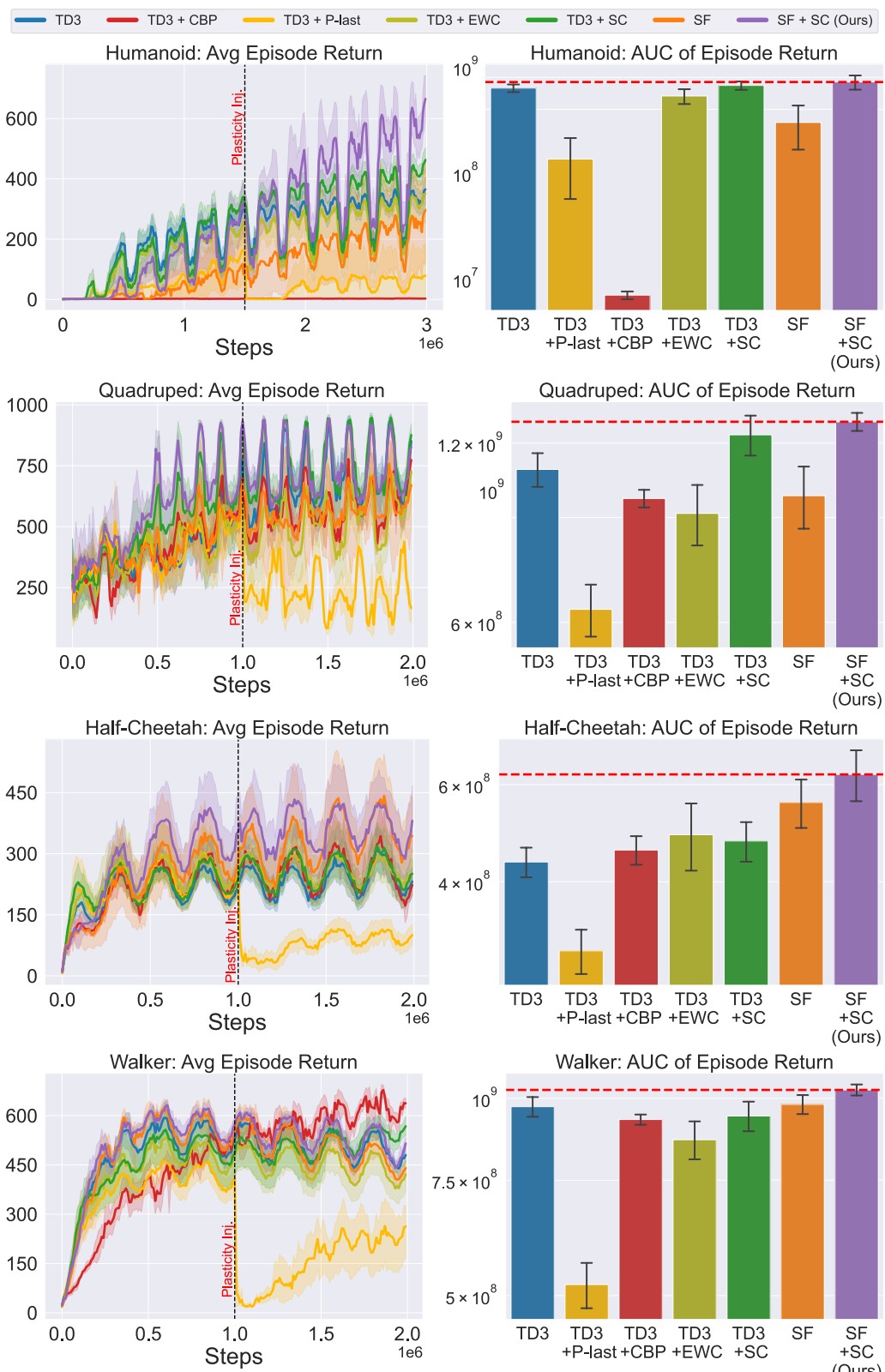

*Figure 17.* Plasticity–stability analysis in the MuJoCo suite under continuous mass changes induced by a periodic noisy sine function during training and evaluation. We compare a baseline TD3 agent with three variants: (i) Continual Backprop (CBP), which selectively resets least-active weights; (ii) plasticity injection by resetting the weights in the last layer (P-last); and (iii) synaptic consolidation (SC). Across embodiments, CBP generally struggled to outperform TD3, with complete collapse in Humanoid, while SC improved stability.

## H. Plasticity-Stability Analysis in non-stationary conditions induced by a non-periodic noisy sinusoidal function

In this section, we present experimental results for the 3D Miniworld environment and various MuJoCo embodiments undergoing naturalistic, continuous changes generated from a non-periodic noisy sinusoidal function. See Appendix E, specifically Sections E.3 and E.4, for illustrations of the resulting dynamics.

### H.1. Slippery Four Rooms Environment

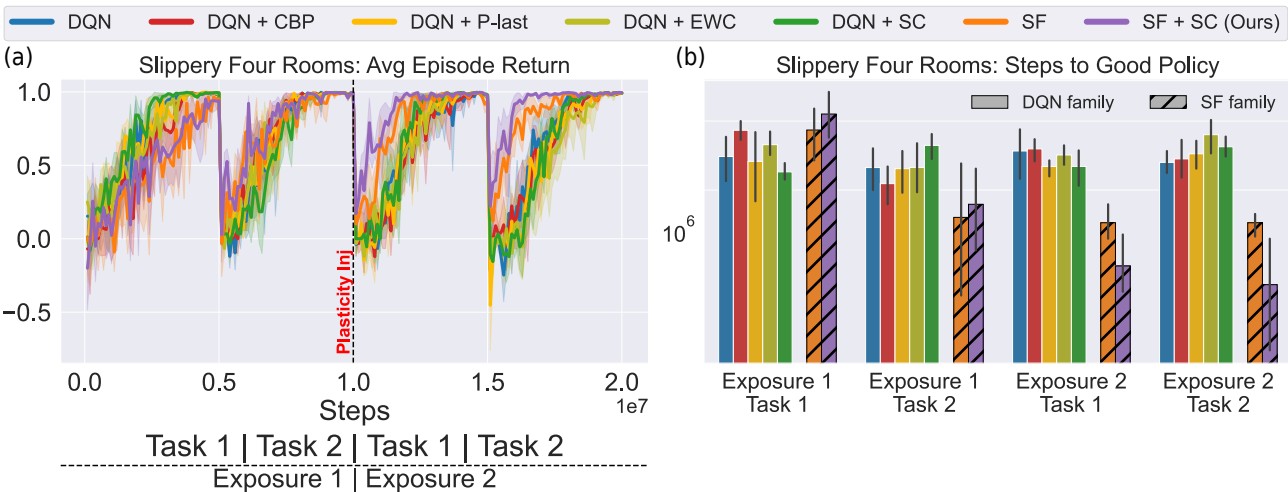

*Figure 18.* Plasticity–stability analysis in the Slippery Four Rooms environment using non-periodic noisy sinusoidal function. The agent undergoes two exposures; after each learning phase, the reward mapping is reversed. **(a)** Average return per episode. **(b)** Learning efficiency (steps to reach a good policy; lower is better). For the plasticity-injection agent, plasticity was injected once at 10 million environment steps (end of Exposure 1). In both panels, the agent with synaptic consolidation (SC) applied to the SFs (SF + SC, purple) consistently achieved better learning efficiency when re-encountering the same set of tasks sequentially in exposure 2.

### H.2. MuJoCo suite Results

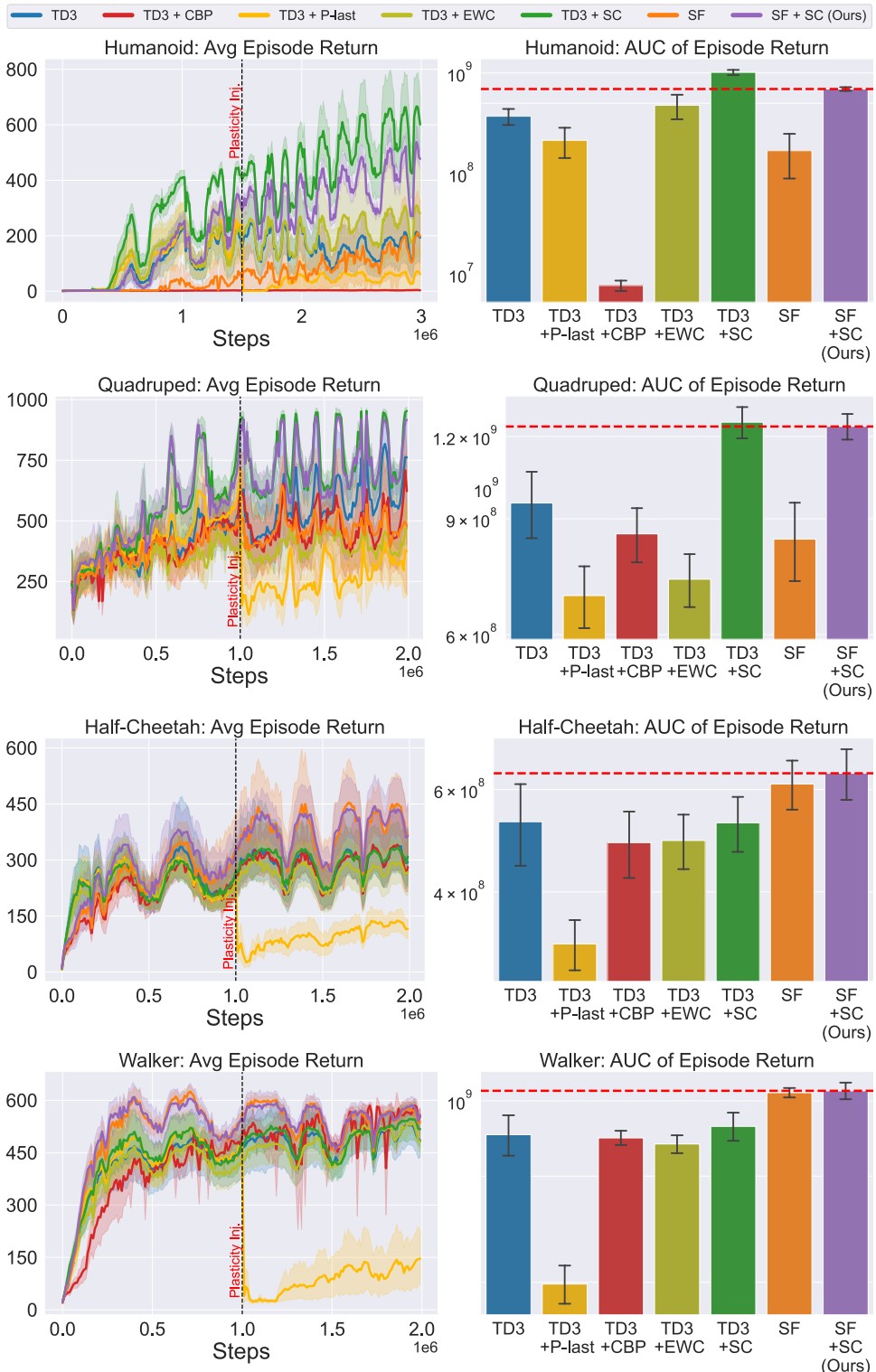

*Figure 19.* Plasticity–stability analysis in the MuJoCo suite under continuous mass changes induced by a non-periodic noisy sine function during training and evaluation. We compare a baseline TD3 agent with three variants: (i) Continual Backprop (CBP), which selectively resets least-active weights; (ii) plasticity injection by resetting the weights in the last layer (P-last); and (iii) synaptic consolidation (SC). Across embodiments, CBP generally struggled to outperform TD3, with complete collapse in Humanoid, while SC improved stability. Applying SC to SFs leads to improved learning performance. However, for complex embodiments such as Quadruped and Humanoid, these gains are reduced. This is unsurprising, as SFs are predictive representations, and the underlying mass dynamics become less predictable in non-periodic settings.

# I. Plasticity-Stability Analysis in non-stationary conditions induced by Ornstein–Uhlenbeck processes

In this section, we present experimental results for the 3D Miniworld environment and the various MuJoCo embodiments undergoing naturalistic, continuous changes sampled from the Ornstein–Uhlenbeck processes. See Appendix E, specifically Sections E.5 and E.6, for illustrations of the resulting dynamics.

## I.1. Slippery Four Rooms Environment

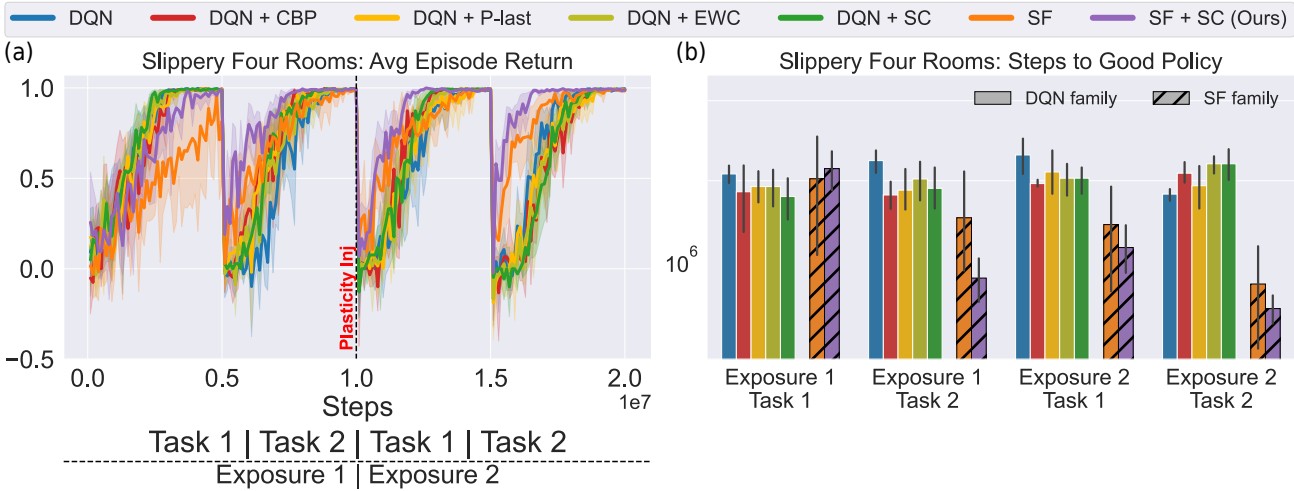

*Figure 20.* Plasticity–stability analysis in the Slippery Four Rooms environment using OU processes. The agent undergoes two exposures; after each learning phase, the reward mapping is reversed. **(a)** Average return per episode. **(b)** Learning efficiency (steps to reach a good policy; lower is better). For the plasticity-injection agent, plasticity was injected once at 10 million environment steps (end of Exposure 1). In both panels, the agent with synaptic consolidation (SC) applied to the SFs (SF + SC, purple) consistently achieved better learning efficiency when re-encountering the same set of tasks sequentially in exposure 2.

## I.2. MuJoCo suite Results

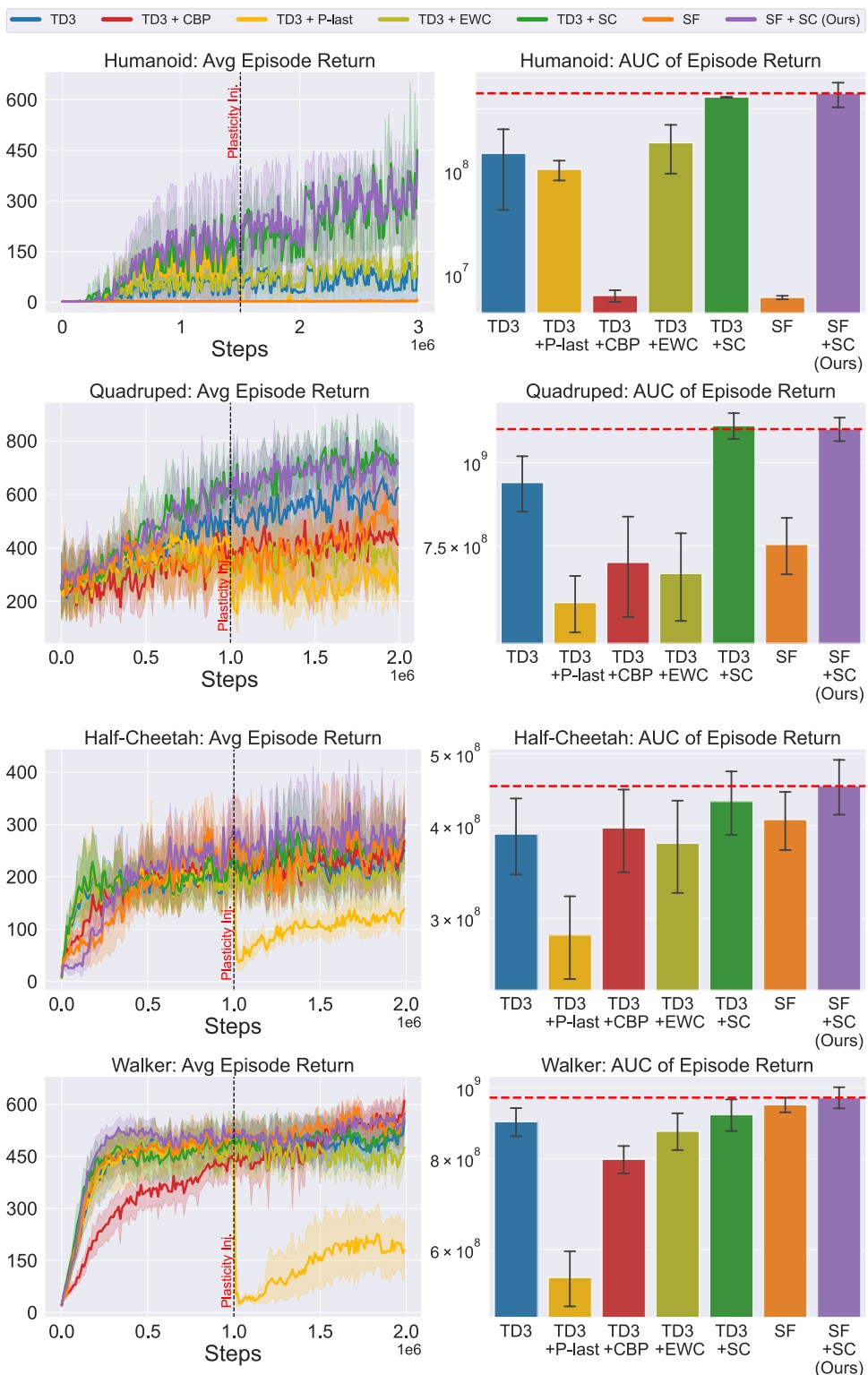

*Figure 21.* Plasticity–stability analysis in the MuJoCo suite under continuous mass changes induced by Ornstein–Uhlenbeck processes during training and evaluation. We compare a baseline TD3 agent with three variants: (i) Continual Backprop (CBP), which selectively resets least-active weights; (ii) plasticity injection by resetting the weights in the last layer (P-last); and (iii) synaptic consolidation (SC). Across embodiments, CBP generally struggled to outperform TD3, with complete collapse in Humanoid, while SC improved stability. Applying SC to SFs (SF+SC) leads to similar performance as applying SC to Q-values (TD3+SC). This is unsurprising, as SFs are predictive representations, and the underlying mass dynamics become less predictable in OU settings.

## J. Schematic of Synaptic Consolidation for Q-values vs. Successor Features

In this section, we present a schematic comparison of applying synaptic consolidation to the parameters of Q-values versus Successor Features.

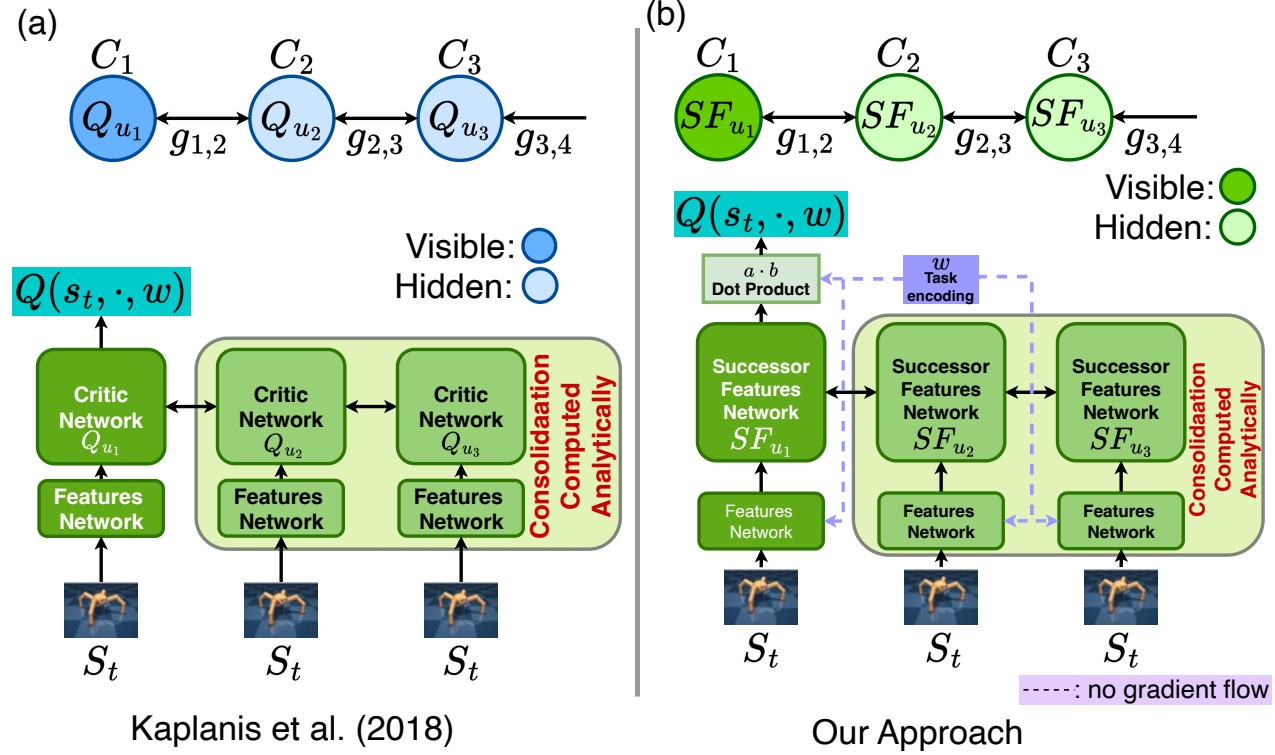

*Figure 22.* Schematic of synaptic consolidation applied to Q-values and to Successor Features (SFs). **(a):** (Kaplanis et al., 2018) showed that adapting the synaptic consolidation mechanism of (Benna & Fusi, 2016) to Q-values improves robustness in continual RL. **(b):** Here, we extend this approach to predictive, generalizable representations using Simple Successor Features (Chua et al., 2024). Consolidated variables (e.g., $Q_{u_2}, Q_{u_3}, \ldots$ or $SF_{u_2}, SF_{u_3}, \ldots$) are computed analytically and therefore lie outside the computational graph used to update $Q_{u_1}$ or $SF_{u_1}$ by backpropagation.

# K. Q-values vs. SFs, Elastic Weight Consolidation vs Synaptic Consolidation Comparison

In this section, we compare the effects of applying synaptic consolidation (SC) (Benna & Fusi, 2016) and Elastic Weight Consolidation (EWC) (Kirkpatrick et al., 2017) to either the Q-value parameters or the Successor Feature (SF) parameters. Across all environments, SC consistently outperforms EWC, regardless of whether the consolidation mechanism is applied to the Q-values or the SFs. These results suggest that multi-timescale consolidation dynamics provide a more effective mechanism for maintaining stability under continuous non-stationarity than importance-based regularization alone.

### K.1. Slippery Four Rooms Environment

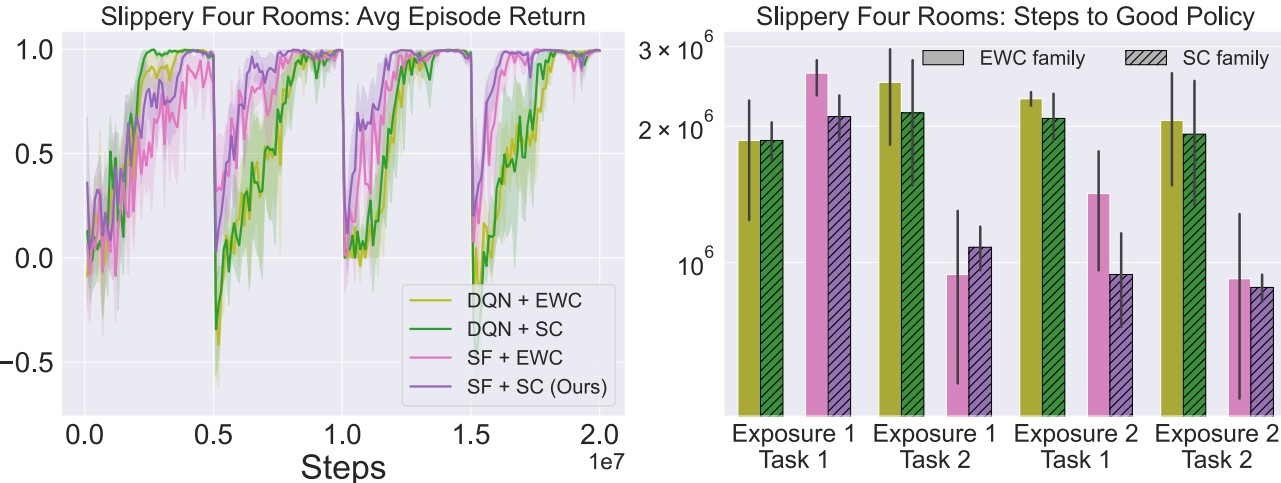

*Figure 23.* Comparison of Q-values and Successor Features (SFs), with synaptic consolidation (SC) or elastic weight consolidation (EWC), in the 3D Slippery Four Rooms environment during training and evaluation. Applying SC to Q-values (green) and SFs (purple) offers higher learning efficiency than their EWC counterparts, requiring fewer steps to learn a good policy. This demonstrates that SC is more effective than EWC, particularly when applied to SFs, during the agent's second exposure to the tasks.

### K.2. MuJoCo suite with continuous mass changes

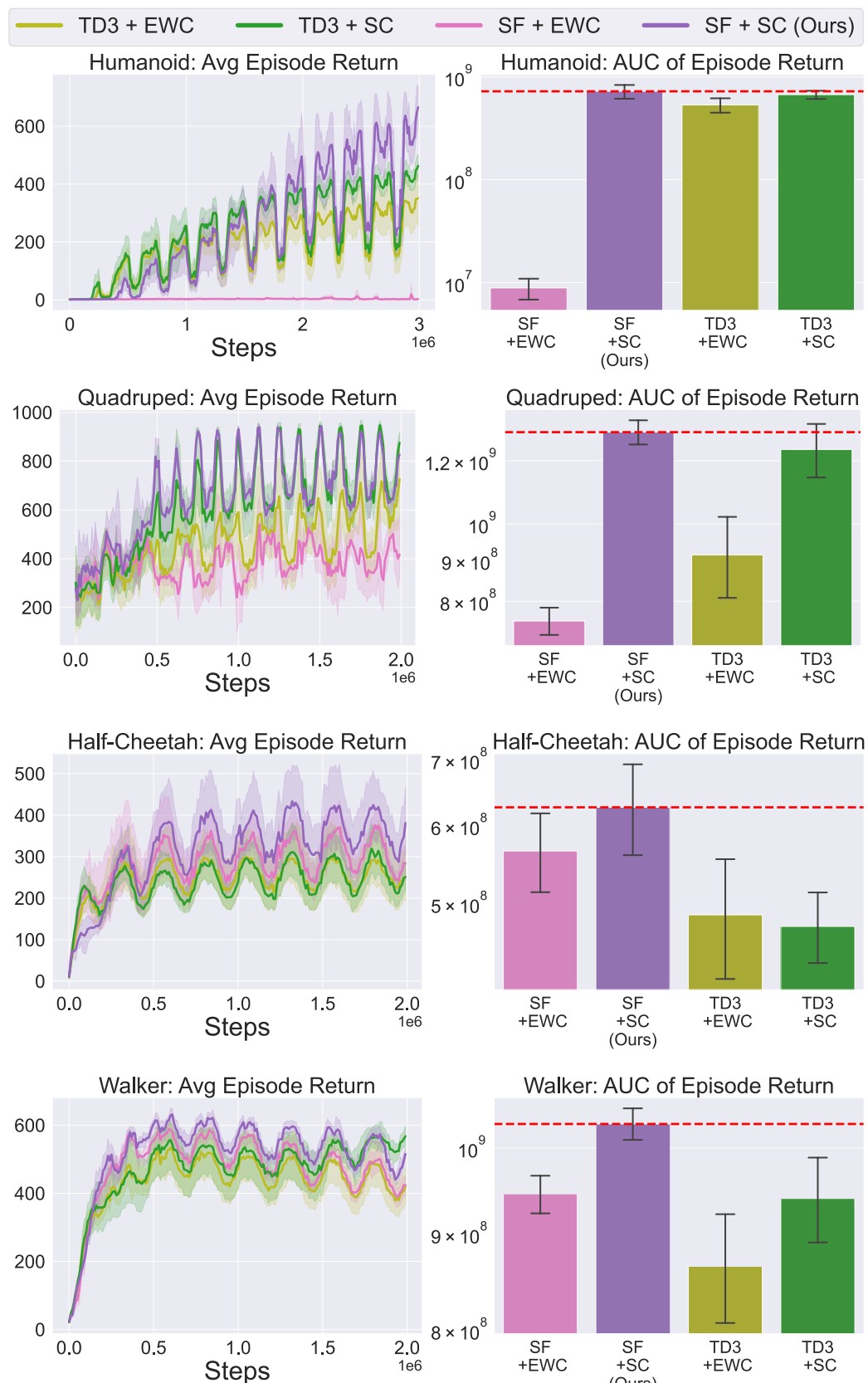

*Figure 24.* Comparison of Q-values and Successor Features (SFs), with and without synaptic consolidation, on the MuJoCo suite under continuous mass changes during training and evaluation. Interestingly, unlike Q-values, applying synaptic consolidation to SFs (purple) yields consistently higher learning efficiency.

# L. Q-values vs. SFs, With and Without Synaptic Consolidation Comparison

In this section, we present the results of applying synaptic consolidation to the parameters of Q-values versus the parameters of Successor Features.

### L.1. Slippery Four Rooms Environment

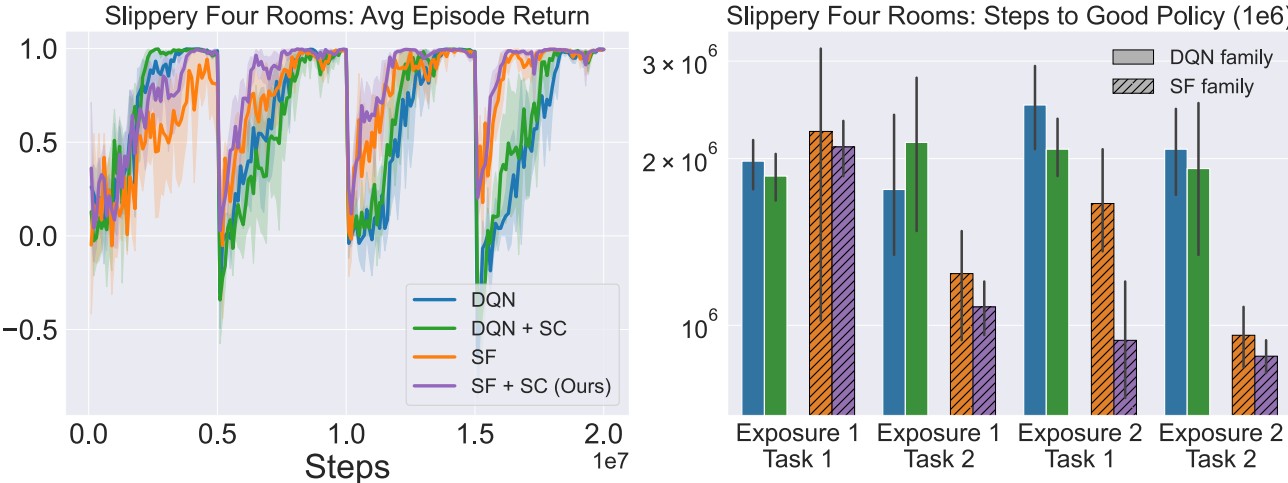

*Figure 25.* Comparison of consolidating the parameters of Q-values and SFs using Synaptic Consolidation (SC) using the 3D Slippery Four Rooms environment. **(left):** Average episode return plot. **(right):** Number of training steps needed to reach a pre-determined good policy. Lesser steps the better. Applying SC to the SFs (purple) yields better learning performance overall.

### L.2. MuJoCo suite with periodic mass changes

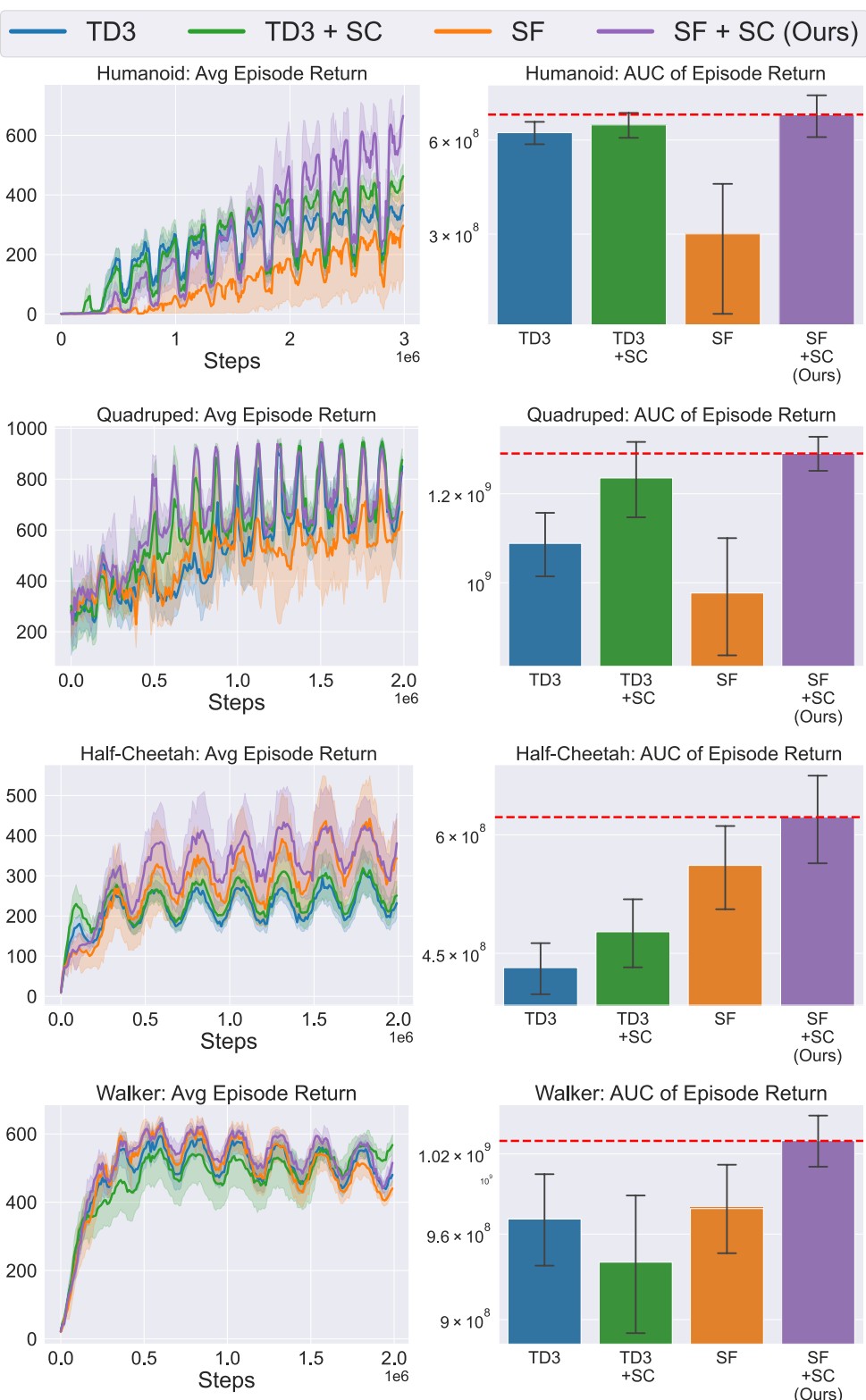

*Figure 26.* Comparison of consolidating the parameters of Q-values and SFs using Synaptic Consolidation using the MuJoCo suite. Interestingly, when compared to TD3 (blue), SFs (orange) learn well in Half-Cheetah and Walker but not Quadruped and Humanoid. This is probably due to higher complexity in Quadruped and Humanoid as they have larger state and action spaces. Overall, applying SC to the SFs (purple) yields better learning performance, highlighting their effectiveness when combined together.

**L.3. MuJoCo suite with non-periodic mass changes**

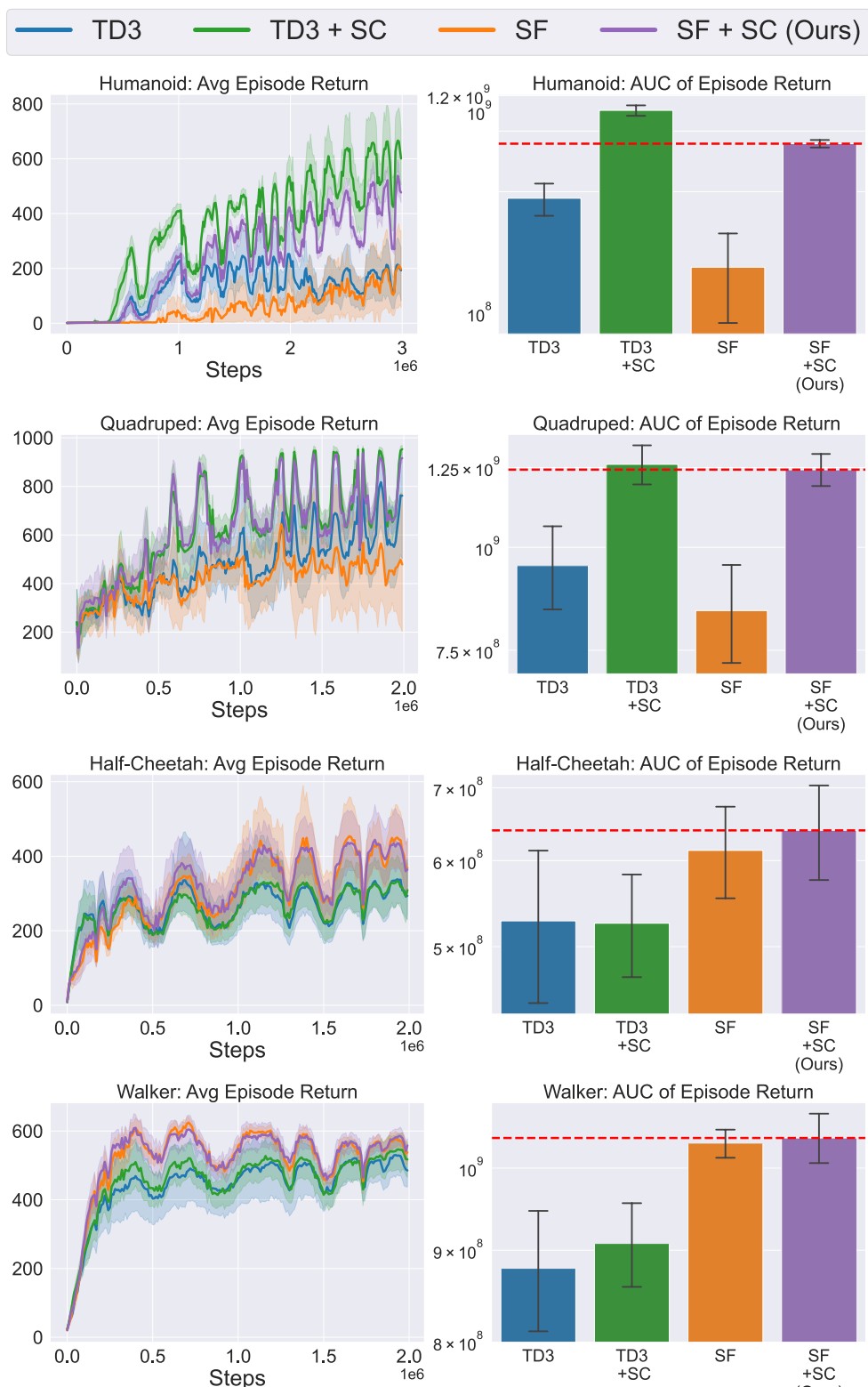

*Figure 27.* Comparison of consolidating the parameters of Q-values and SFs using Synaptic Consolidation using the MuJoCo suite when embodiments undergo non-periodic mass changes. Interestingly, when compared to TD3 (blue), SFs (orange) learn well in Half-Cheetah and Walker but not Quadruped and Humanoid. This is probably due to higher complexity in Quadruped and Humanoid as they have larger state and action spaces. Unlike in the periodic mass changes setting, here, applying SC to the SFs (purple) only yields better learning performance in simpler embodiments such as Half-Cheetah and Walker, but struggle in the more complex embodiments such as Quadruped and Humanoid when compared to applying SC to the Q-values (green).

**L.4. MuJoCo suite with Ornstein-Uhlenbeck mass changes**

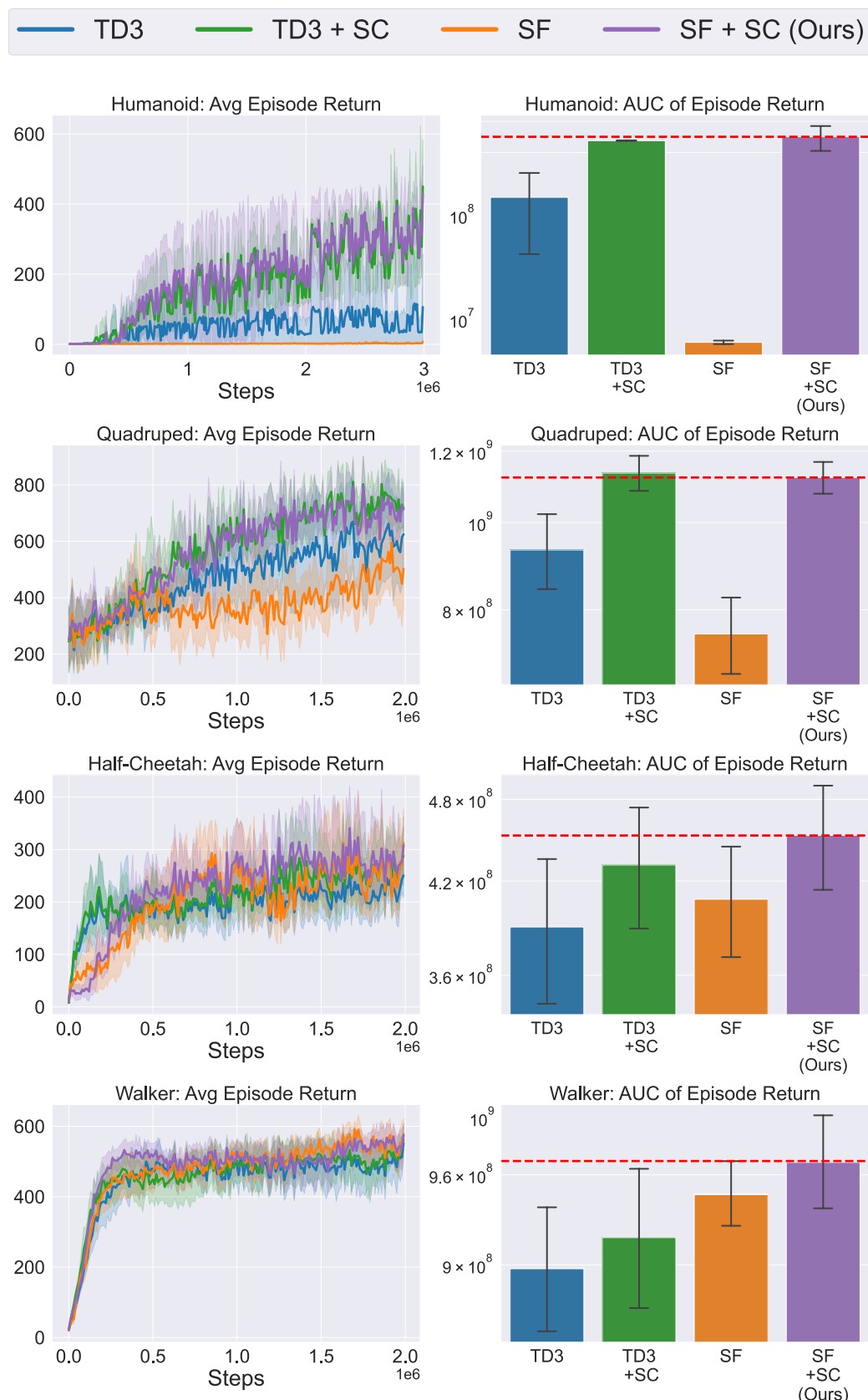

*Figure 28.* Comparison of consolidating the parameters of Q-values and SFs using Synaptic Consolidation using the MuJoCo suite when embodiments under Ornstein-Uhlenbeck mass Changes. In this setting, applying SC to the SFs (purple) only yields better learning performance compared to applying SC to Q-values.

# M. Analysis of Fast and Slow Timescale Variables

In this section, we present analyses aimed at understanding the contributions of the different timescale variables. To do so, we control for the number of consolidation variables. Increasing the number of consolidation variables allows the Successor Features to be preserved over longer timescales.

## M.1. Slippery Four Rooms Environment Results

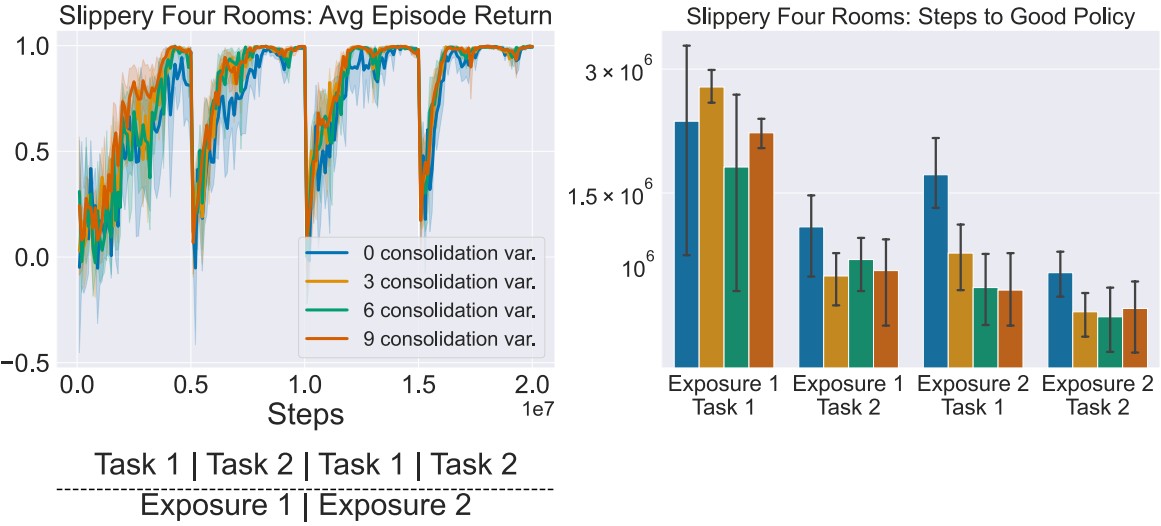

*Figure 29.* Analysis of fast and slow timescale variables in the 3D Slippery Four Rooms environment during training and evaluation. Using synaptic consolidation clearly leads to better learning efficiency, but there is no clear advantage between six and nine consolidation variables.

## M.2. MuJoCo suite Results

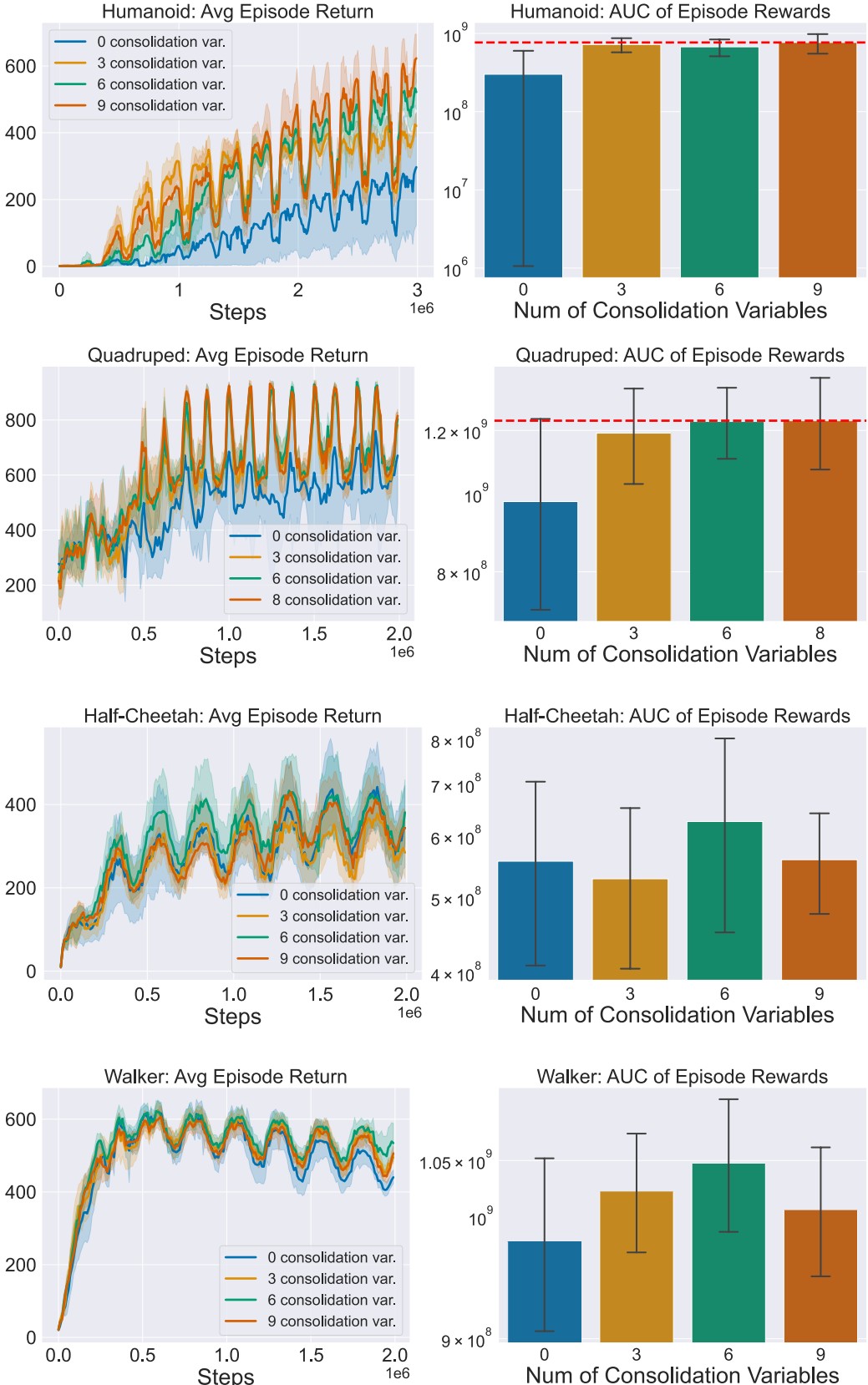

*Figure 30.* Analysis of fast and slow timescale variables on the MuJoCo suite under continuous mass changes during training and evaluation. Using more consolidation variables (six, eight or nine) yields consistently higher learning efficiency, highlighting the importance of slower-timescale variables.

# N. Architecture for recalling consolidated Successor Features using cross-attention

We adapt the canonical cross-attention mechanism by letting the reward weight vector $w$ serves as the query, while the SF consolidation variables (excluding the most plastic $SF_{u_1}$) serve as keys and values. To construct more discriminative representations, we subtracted each variable from its faster neighbor (eg. $SF_{u_k} = SF_{u_k} - SF_{u_{k-1}}$). The keys and values were layer-normalized before projected through learnable weights ($W_{\text{Keys}}, W_{\text{Values}}$), while the query vector $w$ is projected through $W_{\text{Query}}$. Attention scores were computed via the query-keys multiplication, followed by softmax activation, which was then multiplied by the Values to produce a weighted sum, thus integrating information across timescales (see Appendix N for more details). Since the deeper SF consolidation variables ($SF_{u_2}, SF_{u_3}, \ldots$) were computed analytically and not part of the computational graph, we applied a reparameterization trick (inspired by Kingma & Welling (2013)) to enable joint training of the cross-attention mechanism with the most plastic variable ($SF_{u_1}$) using the Q-SF-TD loss (Eq. 6).

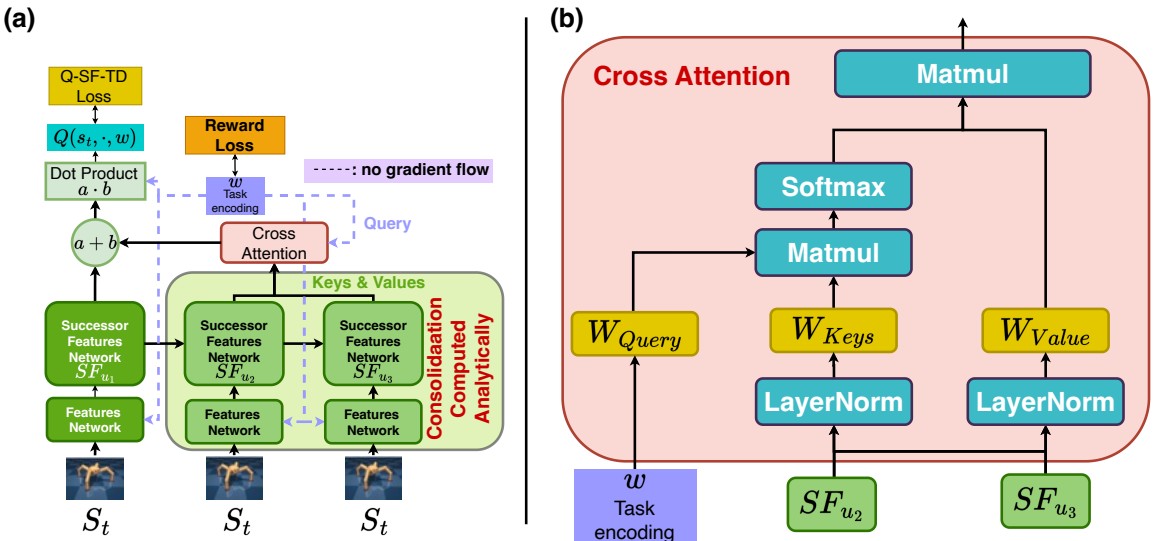

*Figure 31.* Using cross-attention to recall information from the SF consolidation modules. (**a:** A high-level schematic on how the cross-attention mechanism is used. (**b:** The computations for the cross-attention mechanism. We used the reward weight vector $w$ as the query, the SFs consolidation variables except the most plastic one as keys and values ($SF_{u_2}, SF_{u_3}, \ldots, SF_{u_K}$). Because these SFs consolidation variables are computed analytically, they are not part of the computational graph. Therefore, we apply a reparameterization trick to add the output of the cross-attention mechanism to the most plastic SF ($SF_{u_1}$) so that the learnable weights ($W_{Query}, W_{Keys}, W_{Values}$) in the cross-attention mechanism are learned via the Q-SF-TD loss (Eq. 6).

## O. Cross-Attention Analysis of Fast and Slow Timescale Variables

In this section, we perform a set of analyses using the cross-attention architecture (Figure 31) to better understand the functional roles of the individual timescale variables. By allowing interactions across Successor Features consolidated over different timescales, the architecture enables us to examine how rapidly adapting versus slowly varying predictive representations contribute to the stability–plasticity tradeoff under naturalistic non-stationarity.

### O.1. Slippery Four Rooms Environment Results

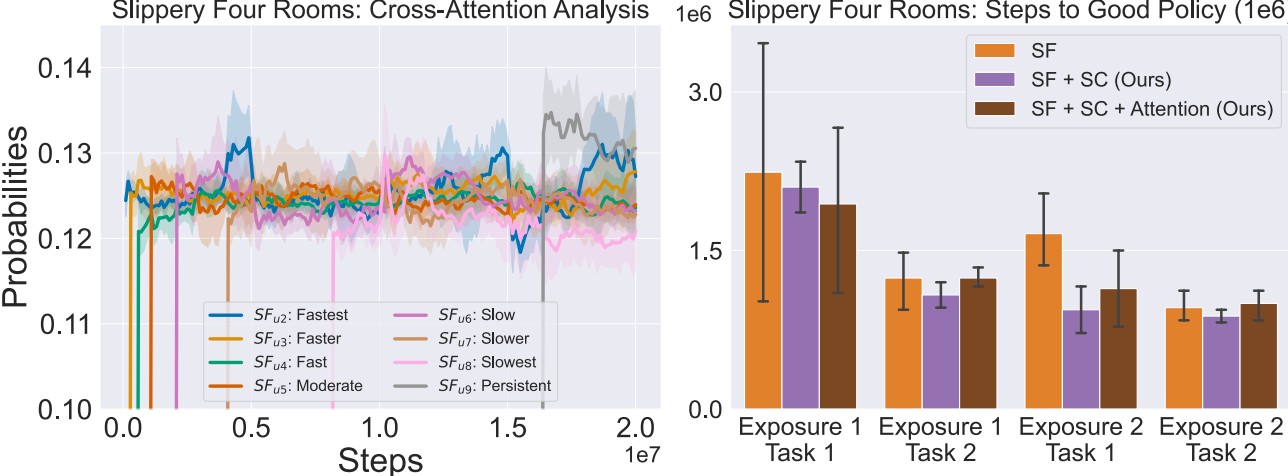

*Figure 32.* Analysis of all consolidated variables using Cross-Attention during training in the 3D Slippery Four Rooms environment. The cross-attention probabilities indicate that fast and slow timescale variables were attended to similarly, suggesting nearly equal contribution. This may be due to the sparse reward structure in the 3D Slippery Four Rooms environment, which affects how discriminate the SFs are given that the SFs are learned via the reward signal using Q-SF-TD loss (Eq. 6)

## O.2. MuJoCo suite Results

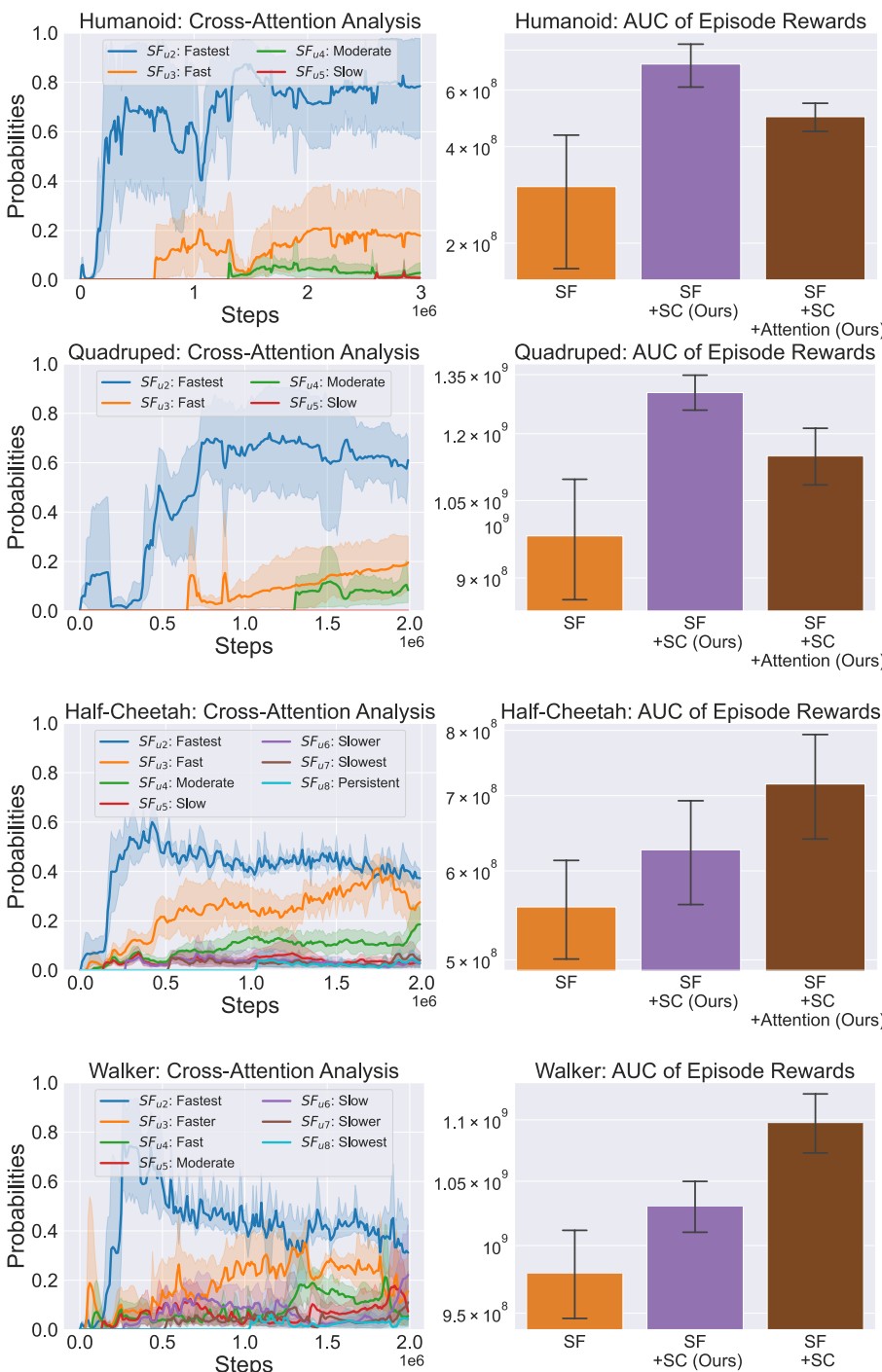

*Figure 33.* Analysis of all consolidated variables using cross-attention in the MuJoCo suite under continuous mass changes. Memory recall was performed solely through the cross-attention mechanism, rather than by waiting for information to propagate from slower to faster timescale variables. Unsurprisingly, faster timescale variables were attended to more than slower ones. Notably, Half-Cheetah and Walker benefited from memory recall via cross-attention, whereas Quadruped and Humanoid require more steps to learn. This may be due to the higher complexity of Quadruped and Humanoid, as their larger state and action spaces reduce the learning efficiency of the cross-attention mechanism.

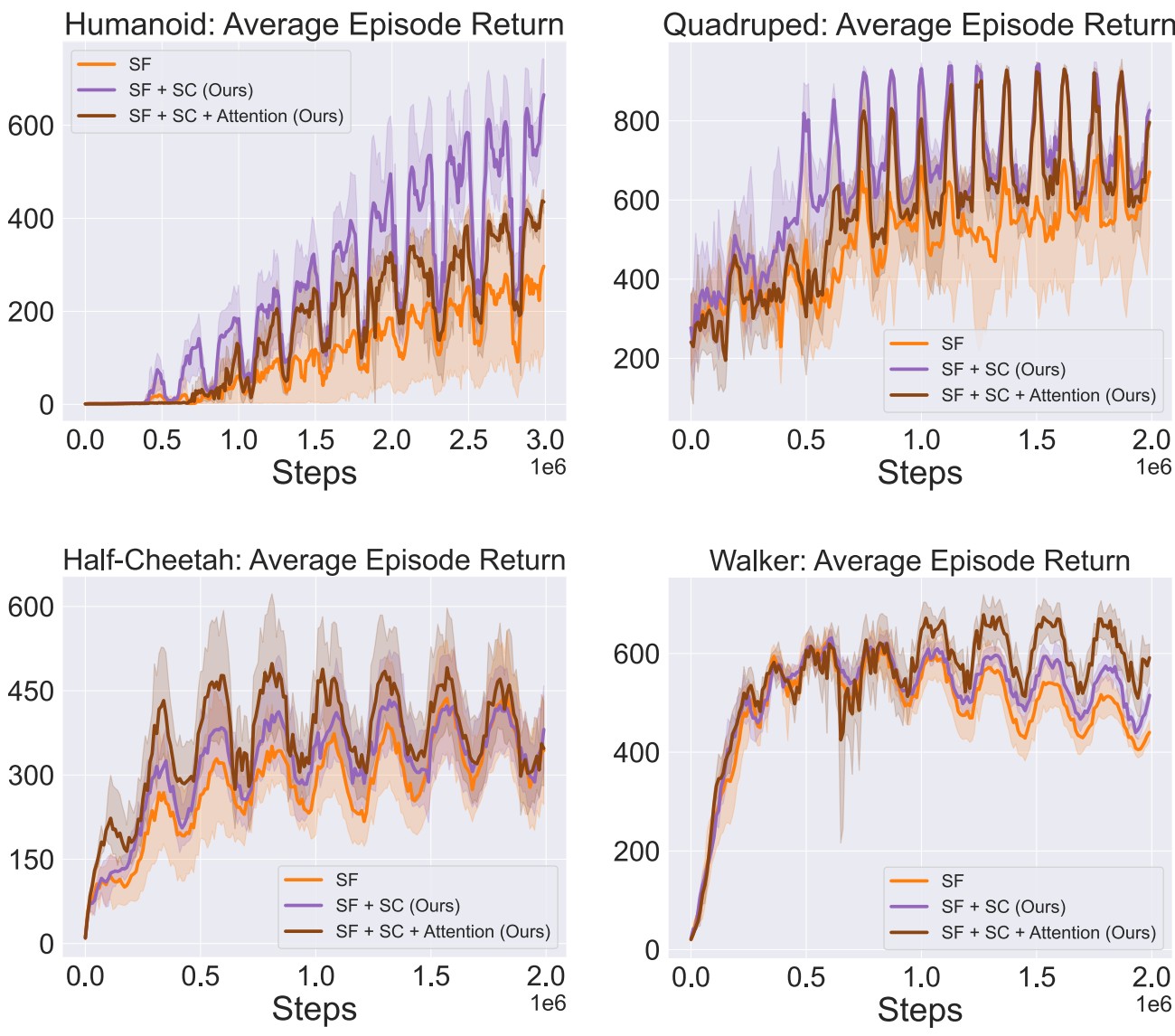

*Figure 34.* Learning curves in the MuJoCo suite under continuous mass changes with cross-attention over consolidated variables. Faster timescale variables were generally attended to more strongly than slower ones as shown in Figure 33. Half-Cheetah and Walker benefited from cross-attention–based memory recall, whereas Quadruped and Humanoid require more steps to learn, likely due to their higher complexity and larger state–action spaces which reduces the learning efficiency of the cross-attention mechanism.

# P. Agents

In this section, we describe our agent as well as the ones we used for comparisons.

## P.1. Successor Features with Synaptic Consolidation

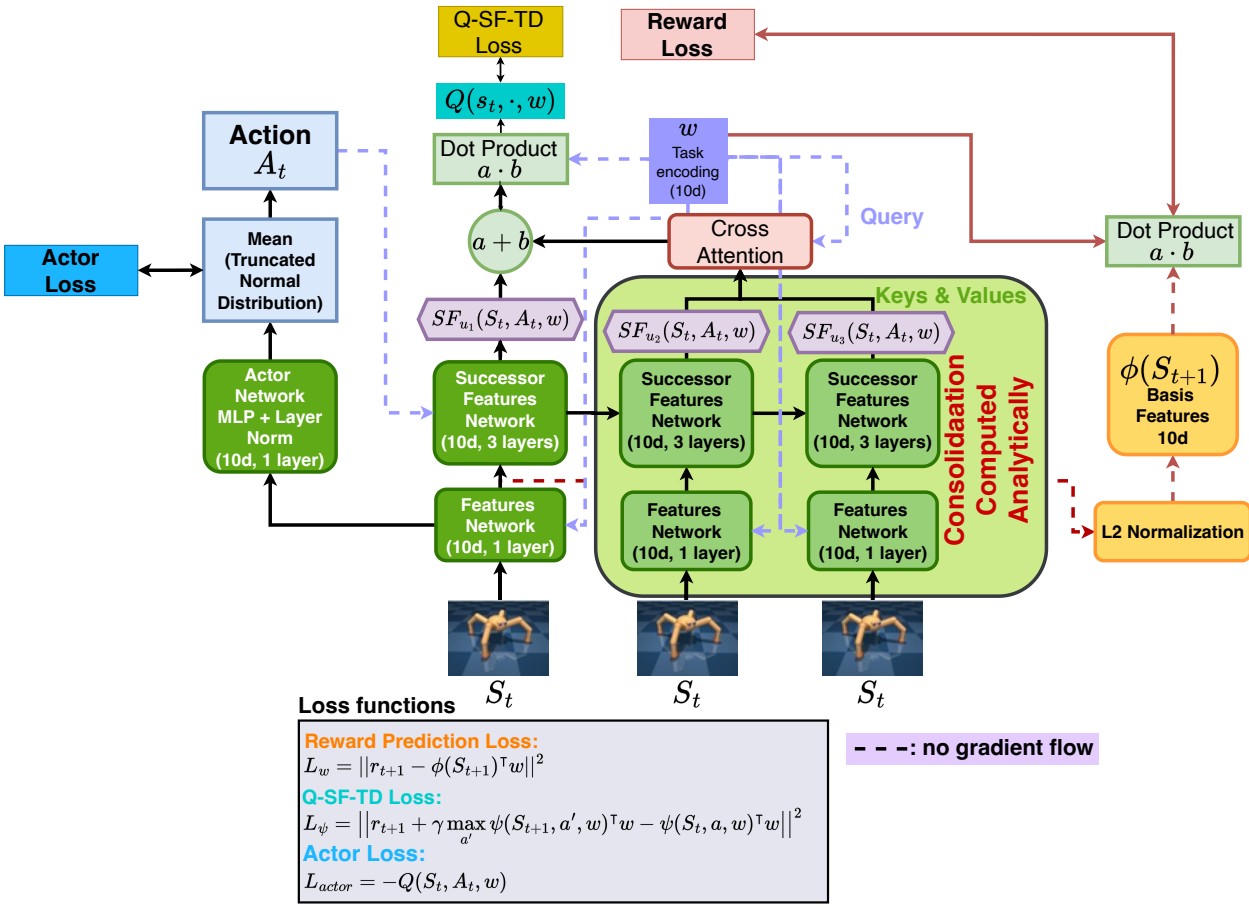

*Figure 35.* Simple SFs with synaptic consolidation architecture. Simple SFs were adapted from (Chua et al., 2024), with TD3 (Lillicrap et al., 2015) as base model. The synaptic consolidation variables are updated analytically (see section 4 for more details on the consolidation variables).

We swept the task learning rate of the reward weight vector across the values of $\{10^{-5}, 10^{-6}, \ldots, 10^{-10}\}$ when optimizing the reward prediction loss (Eq. 5) for the MuJoCo suite. In the naturalistic, continual non-stationary setting that we are studying, we find that a lower learning rate for learning the reward weight vector $w$ generally helps.

The DQN variant largely follows the same architecture, but without the actor network. The only major difference is that hidden dimensions are set to 256. The encoder architecture is the same as DQN encoder.

*Table 3.* Simple SF with Synaptic Consolidation Hyperparameters

| PARAMETER | VALUE |
|---|---|
| OPTIMIZER | ADAM (KINGMA & BA, 2014) |
| DISCOUNT($\gamma$) | 0.99 |
| REPLAY BUFFER SIZE | 1 MILLION |
| DOUBLE Q | YES (FUJIMOTO ET AL., 2018) |
| TARGET NETWORK: UPDATE PERIOD | 1000 |
| TARGET SMOOTHING COEFFICIENT | 0.01 |
| MIN REPLAY SIZE FOR SAMPLING | 5000 |
| LEARNING RATE | 0.0001 |
| NUMBER OF CONSOLIDATION VARIABLES | 9 |
| FLOW STRENGTH ($g_{1,2}$) | 0.125 |
| CAPACITY FOR THE FIRST VARIABLE ($C_1$) | 2 |
| LEARNING RATE FOR CONSOLIDATION | 1 |
| CAPACITY SCALING FACTOR ($s$) | 20 |
| BASIS FEATURES $\phi$ | L2-NORMALIZE (OUTPUT OF ENCODER) |
| FEATURE $\phi$ DIMENSION | 10 |
| FEATURES-TASK NETWORK HIDDEN UNITS | $\{1024, 10\}$ |
| FEATURES-TASK NETWORK NORMALIZATION | LAYER-NORM |
| FEATURES-TASK NETWORK NON-LINEARITY | TANH |
| SF $\psi$ DIMENSION | 10 |
| SF $\psi$ NETWORK HIDDEN UNITS | $\{1024, 1024, \text{SF } \psi \text{ DIM}\}$ |
| SF $\psi$ NETWORK NON-LINEARITY | RELU |

*Table 4.* Task $w$ encoding Hyperparameters

| PARAMETER | VALUE |
|---|---|
| TASK $w$ DIMENSION | 10 |
| TASK $w$ LEARNING RATE | ENVIRONMENT-DEPENDENT (SEE TABLE 1 & 2) |
| TASK $w$ OPTIMIZER | ADAM (KINGMA & BA, 2014) |

## P.2. Elastic Weight Consolidation

For the Elastic Weight Consolidation (EWC) agents (Kirkpatrick et al., 2017), we adapt the online variant (Schwarz et al., 2018), such that instead the fisher information is computed at every $k$ steps instead of waiting till the end of the task, thus removing the need for tasks boundaries information. The EWC loss is defined as:

$$L(\theta) = L_{TD}(\theta) + \sum_i \frac{\lambda}{2} F_i (\theta_i - \theta_i^*)^2 \qquad (32)$$

where $i$ is the number of parameters, $\lambda \in \mathbb{R}$ is the regularization factor, $L_{TD}$ can be either the DQN loss or Q-ST-TD loss (Eq. 6), if we are learning SFs. The fisher information is computed by squaring the gradients of the parameters. In our experiments, we set the fisher computation interval to every 10k steps, which is the number of steps per episode in MuJoCo. We also swept the regularization factor $\lambda \in \{12, 25, 75, 125, 175\}$, following the same setup as (Schwarz et al., 2018). In our experiments, we find $\lambda = 25$ works best.

## P.3. Plasticity Injection Model

For the plasticity injection models, we consider plasticity injection on the last layer (P-last) for both the Slippery Four Rooms environment and the MuJoCo suite. We follow the same setup as (Nikishin et al., 2023), whereby during the plasticity injection step $k$, we retain freeze the parameters of last layer in the artificial neural network $\theta$ and introduce a new set of parameters $\theta'$ which is sampled from random initialization. The set of new of parameters is then further copied, such that we will have $\theta' = \theta'_1 = \theta'_2$ and the output of the network is computed using:

$$h_\theta(x) + h_{\theta'_1}(x) - h_{\theta'_2} \qquad (33)$$

where $h$ is the function of the artificial neural network and $x$ is the input. After the plasticity injection step $k + 1$ and beyond, only $\theta_1'$ is allowed to be updated while $\theta', \theta_2'$ are kept frozen, leading $(\theta' - \theta_2')$ to be the bias term.

We experimented with injecting plasticity at 25%, 50% and 75% of the training steps, but observed little effect in our setting. We also found that this method was ineffective when adapted to TD3 and evaluated in MuJoCo. Therefore, we included Continual Backprop (Dohare et al., 2024) as additional baseline for both the Slippery 3D Four Rooms environment and MuJoCo.

## P.4. Continual Backprop

The plasticity injection method described above can hurt learning performance when injecting plasticity into the critic network of actor-critic architecture. Therefore, we consider another baseline model, known as continual backprop (CBP), that is more effective in enhancing plasticity (Dohare et al., 2024). Rather than re-initializing parameters at random, CBP selectively injects plasticity into parameters who were least useful for the current task. This least useful metric is known as the contribution utility, which measures both the hidden unit's activity and its outgoing connection strength. For each hidden unit $i$ in layer $l$ at time $t$, the contribution utility is defined as:

$$u_l[i] = \eta \times u_l[i] + (1 - \eta) \times |h_{l,i,t}| \times \sum_{k=1}^{n_{l+2}} |w_{l,i,k,t}| \tag{34}$$

where $h_{l,i,t}$ is the output of the $i$th hidden unit in layer $l$ at time $t$, $w_{l,i,k,t}$ is the weight connecting the $i$th unit in layer $l$ to the $k$th unit in layer $l + 1$ at time $t$ and $n_{l+1}$ is the number of units in the next layer $l + 1$. This contribution utility can be thought of as a running average of instantaneous contributions, with a decay rate $\eta$.

At each step, CBP identifies eligible units for re-initialization based on two criteria, first is the low contribution utility which indicates that the unit has not been useful in recent training phase, and second is the lifespan, which ensures that units are only re-initialized after allowing to have a sufficient steps of learning. By periodically resetting under-utilized units, CBP ensures that the network maintains plasticity throughout learning. In our experiments, we broadly follow the parameters defined in (Dohare et al., 2024). We swept the replacement rate across the values of $\{10^{-3}, 10^{-4}, 10^{-5}\}$ but observed little effect in our setting, so we kept it at $10^{-4}$. We also swept the maturity threshold across the values of $\{100, 1000, 10000\}$ and also observed little effect, so we kept it as 1000.

In our experiments, unsurprisingly, we found CBP to be much more effective than the plasticity injection model (P-last) introduced in section P.3. Particularly in Quadruped, Half-Cheetah and Walker (Figure 17), we see that CBP outperforms P-last. However, across MuJoCo tasks, CBP generally failed to outperform its base model, TD3, suggesting that in our natural and continually evolving environments, stability rather than plasticity, is the critical bottleneck, since injecting plasticity did not improve performance.

# Q. Implementation Details

For our experimental setup, we used Python 3 (Van Rossum & Drake, 2009) as the primary programming language. The agents and the training framework were developed using Jax (Bradbury et al., 2018; Godwin* et al., 2020) for the Slippery Four Rooms environment and PyTorch (Paszke et al., 2019) for the MuJoCo embodiments (Todorov et al., 2012). We used the DeepMind Control Suite (DMC) (Tunyasuvunakool et al., 2020) to manipulate the embodiments. Flax (Heek et al., 2024) was employed for implementing the neural network components. For data visualization, we used Seaborn (Waskom, 2021), Matplotlib (Hunter, 2007) on Jupyter Notebooks (Kluyver et al., 2016) to generate the plots. The configuration and management of our experiments were performed with Hydra (Yadan, 2019) and Weights & Biases (Biewald, 2020) respectively. All experiments were conducted using Nvidia A100 GPUs and completed within one to two days max. The code used in the study will be released in the near future, following an internal review process.

# R. Computational Complexity

In this section, we report the computational complexity of all methods, measured in frames per second (FPS) during training. Across both the Slippery Four Rooms and the MuJoCo Humanoid experiments, our model (SF + SC) exhibits the lowest throughput. This slowdown arises from the consolidation dynamics, which require analytical, sequential updates between consolidation variables, in addition to learning the SFs and the reward weight vector $w$. Because these updates are not handled by backpropagation and cannot be parallelised, they introduce a constant-factor computational overhead inherent to the mechanism. Moreover, this overhead increases with the number of consolidation variables, where using more variables leads to proportionally lower FPS.

## R.1. Slippery Four Rooms Environment

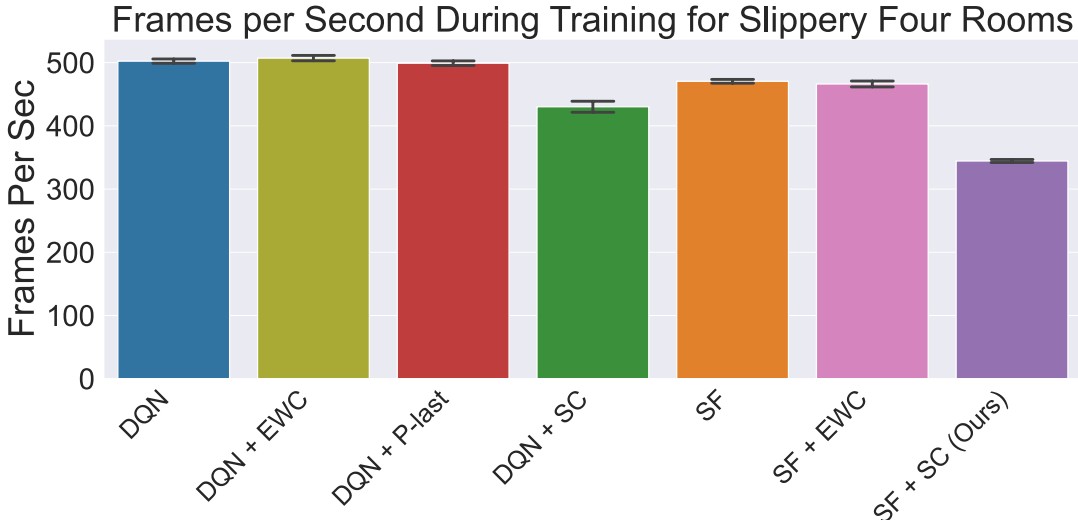

*Figure 36.* Comparison of training throughput (FPS) for all models in the Slippery Four Rooms environment. Higher FPS reflects more efficient computation.

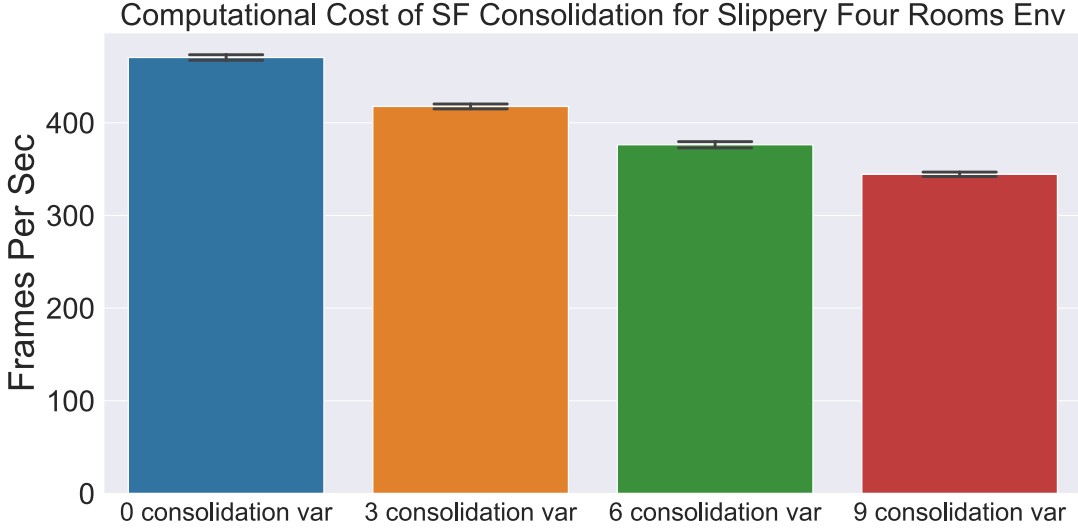

*Figure 37.* Comparison of training throughput (FPS) for different number of consolidation variables for the SFs within the slippery four rooms environment. Higher FPS reflects more efficient computation.

**R.2. MuJoCo - Humanoid**

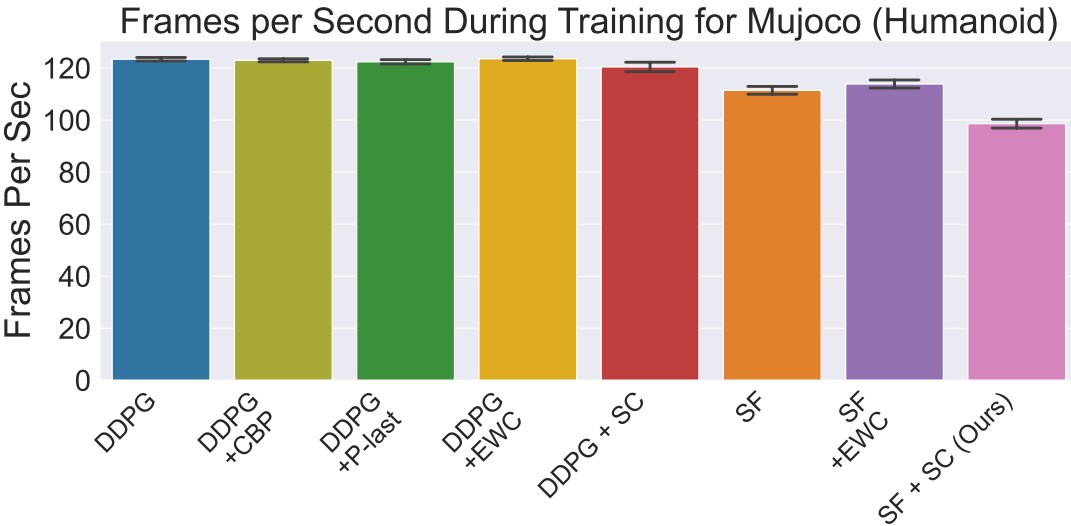

*Figure 38.* Comparison of training throughput (FPS) for all models in the humanoid embodiment within the MuJoCo environment. Higher FPS reflects more efficient computation.

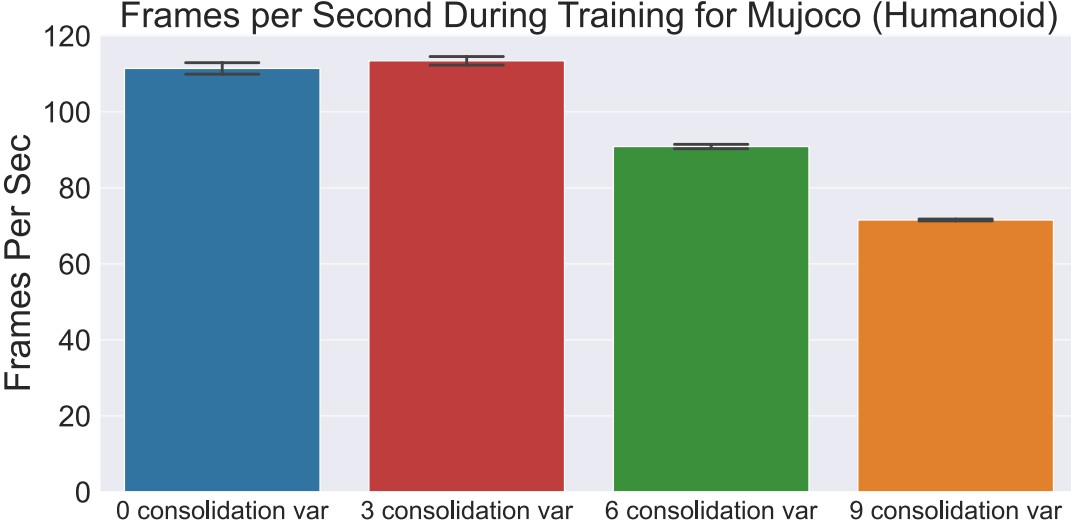

*Figure 39.* Comparison of training throughput (FPS) for different number of consolidation variables for the SFs within the humanoid embodiment within the MuJoCo environment. Higher FPS reflects more efficient computation.

# S. Comparison with Trust-Region Methods (PPO)

In this section, we include Proximal Policy Optimization (PPO) (Schulman et al., 2017) as an additional baseline to investigate whether trust-region optimization alone is sufficient to maintain stability under continuous non-stationarity. PPO is particularly relevant in this setting because its clipped objective is designed to stabilize policy updates by constraining large changes in the policy distribution across optimization steps. Given that PPO relies on parallel sample collection, we also report the number of environment samples used during training to account for differences in data usage across methods. Despite these stabilization mechanisms, PPO struggles to maintain strong performance under gradual changes in the environment dynamics, suggesting that trust-region-based optimization alone may be insufficient for continual learning under persistent non-stationarity.

## S.1. MuJoCo suite Results

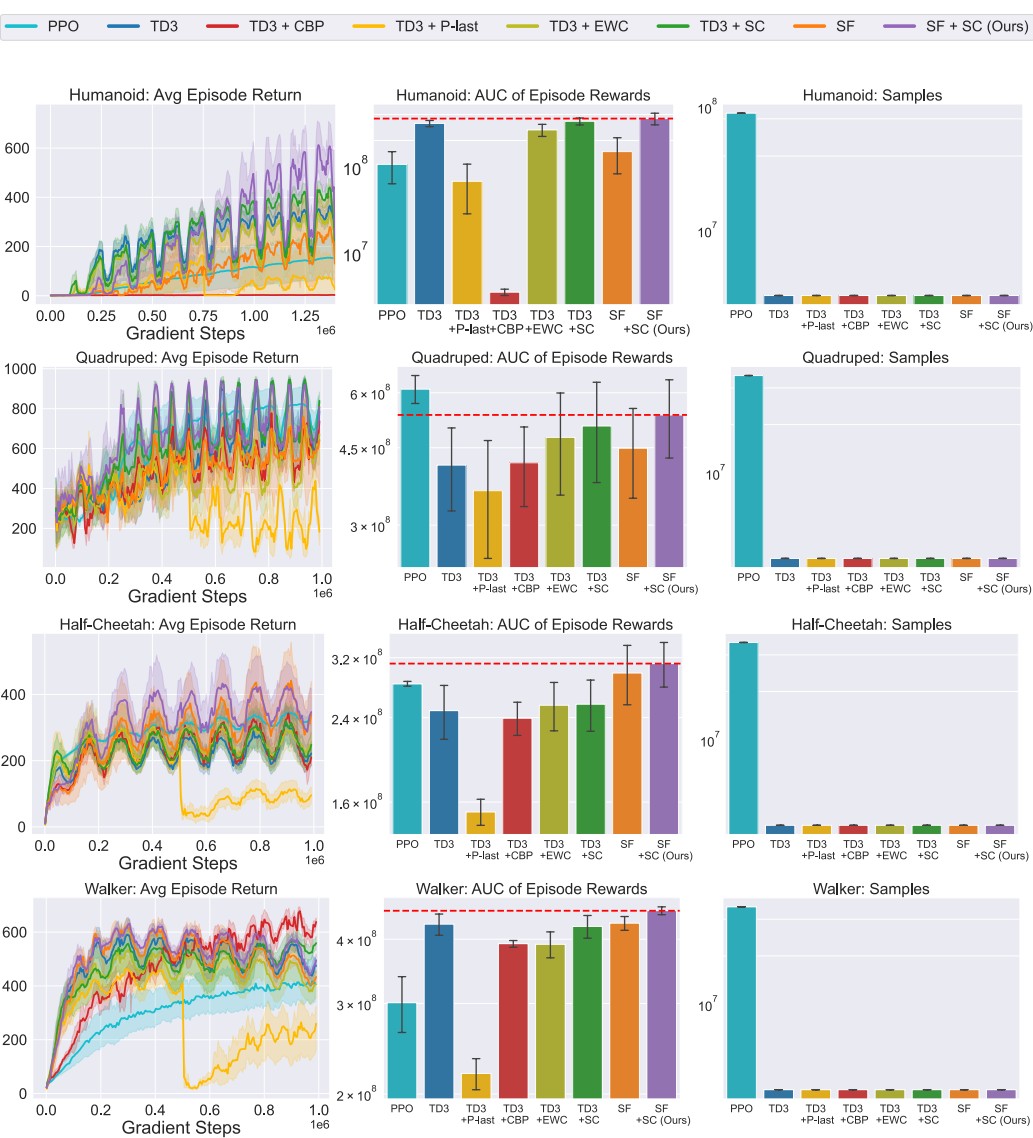

*Figure 40.* Comparison of PPO (teal) with TD3, SF, and their variants with plasticity preservation or stability enhancement mechanisms under continuous mass changes. (Left) Average episode return over training. (Middle) Area under the curve (AUC) of the return, summarizing overall performance. (Right) Total number of environment samples used during training. While PPO leverages parallelized data collection and uses more samples, it achieves lower return and AUC compared to SF + SC (purple). This suggests that trust-region updates alone are insufficient to maintain stable and efficient learning under continuous non-stationarity, compared to explicit multi-timescale consolidation mechanisms.

# T. Quantifying Continuous Non-Stationarity

In this section, we present results under varying levels of perturbation in the 3D Four Rooms environment and MuJoCo, quantified across three regimes: mild, moderate, and severe. For the mild regime, perturbations were capped at 25% of the maximum allowable value. Here, the maximum corresponds either to a 0.45 probability that the selected action is replaced by a randomly sampled alternative action in the 3D Four Rooms environment, or to the largest perturbation permitted by the MuJoCo physics engine before the simulation becomes unstable. For the moderate and severe regimes, perturbations were capped at 50% and 100% of this maximum, respectively. All perturbations were generated using the *periodic* continuous-change setting (Sections E.1 and E.2 in Appendix E).

We observe that, in the 3D Four Rooms environment, applying synaptic consolidation to the Successor Feature (SF) parameters consistently yields more robust learning performance across all perturbation regimes. In contrast, for MuJoCo environments under mild perturbations (25%), applying synaptic consolidation to the Q-value parameters generally results in stronger performance. However, as the magnitude of perturbation increases in the moderate and severe regimes, consolidating the SF parameters becomes increasingly effective.

## T.1. Slippery Probabilities Perturbations (3D Four Rooms)

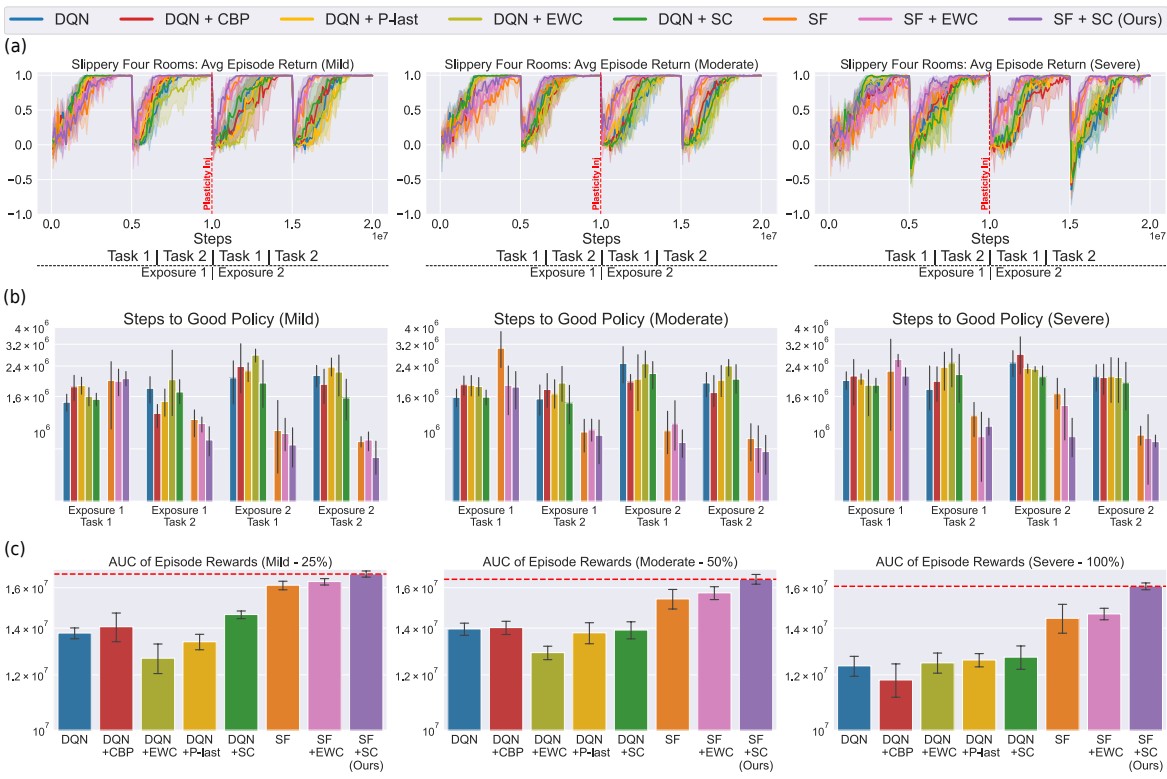

*Figure 41.* Quantification of slippery dynamics in the 3D Four Rooms environment. (a) Average episode return, (b) minimum number of environment steps required to learn a successful policy, and (c) area under the curve (AUC) of the episode returns. We consider three levels of slippery probability variation: mild (25%), moderate (50%), and severe (100%), where the maximum corresponds to a 0.45 probability that the selected action is replaced by a randomly sampled alternative action. The results suggest that methods primarily designed to promote plasticity (CBP, P-last) are less effective under continuous non-stationarity than approaches incorporating synaptic consolidation (SC). While Successor Features (SFs) consistently improve performance across all settings, additional gains are only achieved when SC is applied to SFs.

## T.2. Mass Perturbations (MuJoCo)

### T.2.1. HUMANOID

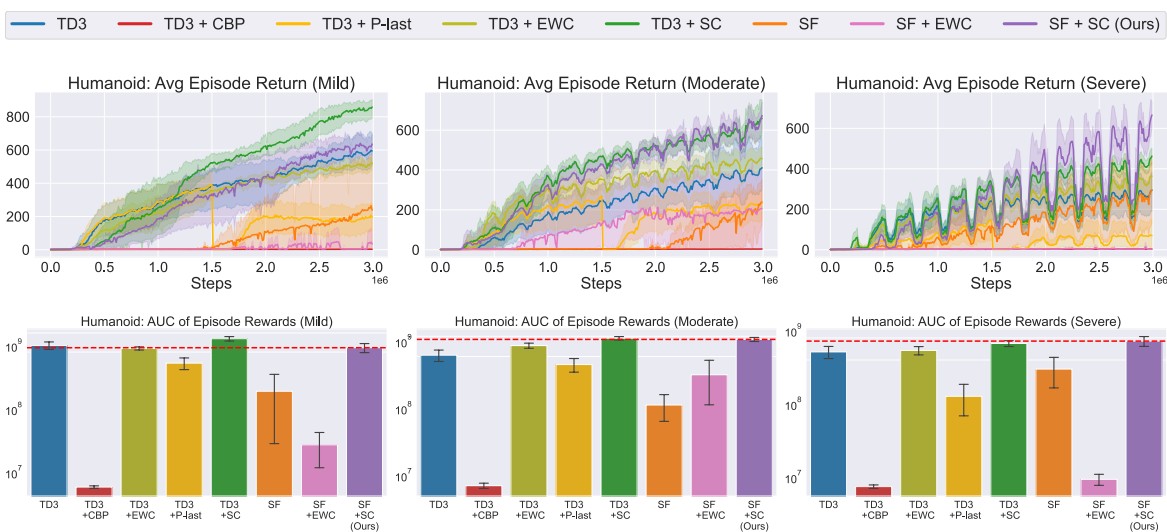

*Figure 42.* Quantification of mass changes for the Humanoid embodiment. We consider three levels of mass dynamics variation: mild (25%), moderate (50%), and severe (100%), corresponding to the maximum change allowed before the physical simulation becomes unstable. Across these settings, plasticity-preserving methods (CBP, P-last) are less effective than approaches incorporating synaptic consolidation (SC). Under moderate—and especially severe—conditions, where the agent experiences substantial changes in dynamics, applying SC to Successor Features (SF + SC) yields the best performance. In contrast, under mild changes, the benefits of SC are limited, as the environment remains close to stationary.

### T.2.2. QUADRUPED

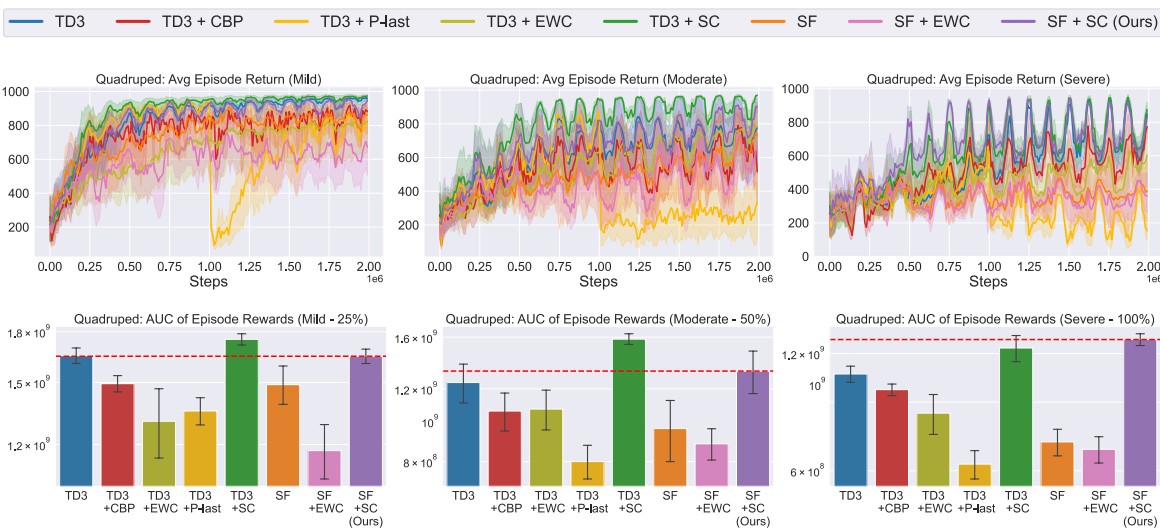

*Figure 43.* Quantification of mass changes for the Quadruped embodiment. We consider three levels of mass dynamics variation: mild (25%), moderate (50%), and severe (100%), corresponding to the maximum change allowed before the physical simulation becomes unstable. Across these settings, plasticity-preserving methods (CBP, P-last) are less effective than approaches incorporating synaptic consolidation (SC). Under severe conditions, where the agent experiences substantial changes in dynamics, applying SC to Successor Features (SF + SC) yields the best performance. In contrast, under mild and moderate changes, applying SC to Q-values (TD3 + SC) yields better performance, as the environment remains closer to stationary.

### T.2.3. HALF-CHEETAH

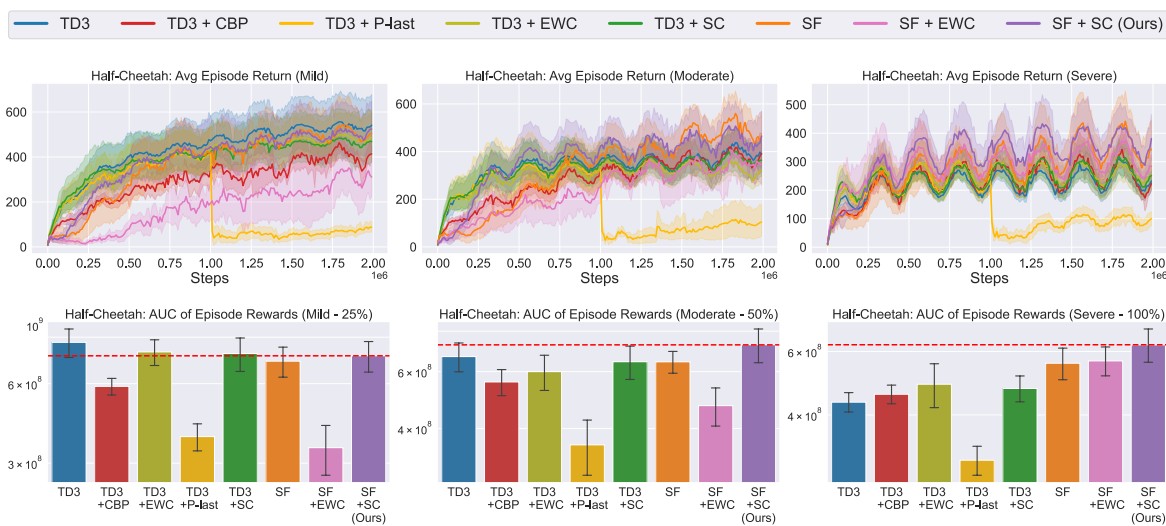

*Figure 44.* Quantification of mass changes for the Half-Cheetah embodiment. We consider three levels of mass dynamics variation: mild (25%), moderate (50%), and severe (100%), corresponding to the maximum change allowed before the physical simulation becomes unstable. Across these settings, plasticity-preserving methods (CBP, P-last) are less effective than approaches incorporating synaptic consolidation (SC). Under moderate—and especially severe—conditions, where the agent experiences substantial changes in dynamics, applying SC to Successor Features (SF + SC) yields the best performance. In contrast, under mild changes, the benefits of SC are limited, as the environment remains close to stationary.

### T.2.4. WALKER

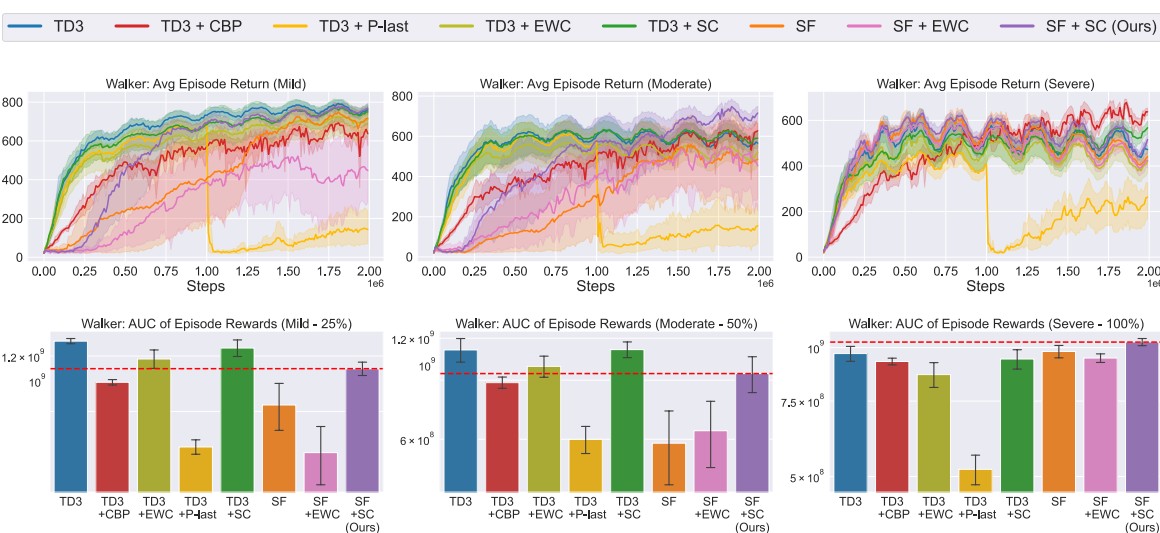

*Figure 45.* Quantification of mass changes for the Walker embodiment. We consider three levels of mass dynamics variation: mild (25%), moderate (50%), and severe (100%), corresponding to the maximum change allowed before the physical simulation becomes unstable. Across these settings, plasticity-preserving methods (CBP, P-last) are less effective than approaches incorporating synaptic consolidation (SC). Under severe conditions, where the agent experiences substantial changes in dynamics, applying SC to Successor Features (SF + SC) yields the best performance. In contrast, under mild and moderate changes, applying SC to Q-values (TD3 + SC) yields better performance, as the environment remains closer to stationary.

