# Balancing plasticity and stability with Fast and Slow Successor Features

## Abstract

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

We consider the RL setting as a Markov Decision Process, defined by a tuple $(S, A, p, r, \gamma)$, where $\mathcal{S}$ is the set of states, $\mathcal{A}$ is the set of actions, $r : S \to \mathbb{R}$ is the reward function, $p : \mathcal{S} \times \mathcal{A} \to [0, 1]$ is the transition probability function and the discount factor $\gamma \in [0, 1)$ which determines the importance of immediate and future rewards (Sutton & Barto, 2018).

At each time step $t$, the agent observes state $S_t \in \mathcal{S}$ and takes an action $A_t \in \mathcal{A}$ sampled from a policy $\pi : \mathcal{S} \times \mathcal{A} \to [0, 1]$, resulting to the transition of next state $S_{t+1}$ with probability $p(S_{t+1} \mid S_t, A_t)$ and the reward $R_{t+1}$.

### 3.2. Successor Features

SFs are defined via a decomposition of the state-action value function (i.e. the expected return), $Q$, into the reward function and a representation of expected features occupancy for each state $S_t$ and action $A_t$ of time step $t$:

$$Q(S_t, A_t, \boldsymbol{w}) = \psi(S_t, A_t, \boldsymbol{w})^\top \boldsymbol{w} \tag{1}$$

where $\psi \in \mathbb{R}^n$ are the SFs that capture expected feature occupancy and $\boldsymbol{w} \in \mathbb{R}^n$ is a vector of the task encoding, which can be considered a representation of the reward function (Borsa et al., 2018).

Canonically, the SFs for a state-action pair $(s, a)$ under a policy $\pi$ are defined as:

$$\psi^\pi(s, a) \equiv \mathrm{E}^\pi \left[ \sum_{i=t}^{\infty} \gamma^{i-t} \phi_{i+1} \mid S_t = s, A_t = a \right] \tag{2}$$

where $\phi \in \mathbb{R}^n$ is a set of basis features (Barreto et al., 2017).

However, as shown by Borsa et al. (2018), we can treat the task encoding vector $\boldsymbol{w}$ as a way to encode policy $\pi$, resulting in *Universal SFs*, $\psi(s, a, \boldsymbol{w})$. The task encoding vector $\boldsymbol{w}$ can also be related directly to the rewards themselves via the underlying basis features ($\phi$):

$$R_{t+1} = \phi(S_{t+1})^\top \boldsymbol{w} \tag{3}$$

Among the various SF learning techniques, our approach builds on Simple SFs (Chua et al., 2024), which can be learned during task engagement, and do not require additional auxiliary losses or pre-training. Both the SFs $\psi$ and the task encoding vector $\boldsymbol{w}$ are learned by optimizing the following two losses:

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

3. Third, will SC remain effective under continual nonperiodic or stochastic drifts?

### 5.1. Stability as the Bottleneck: Synaptic Consolidation outperforms EWC

To study the question of plasticity versus stability, we first evaluated the performance in the slippery Four Rooms environment using DQN, along with its plasticity injection variants (P-last) and stability-preserving variants (EWC and SC). The results in Figure 3 show that stability-preserving models (DQN + EWC and DQN + SC) consistently outperformed the plasticity-injection model (DQN + P-last), with synaptic consolidation (DQN + SC) achieving higher learning efficiency.

We next evaluated performance in the MuJoCo suite using DDPG, together with its two plasticity-injection variants (P-

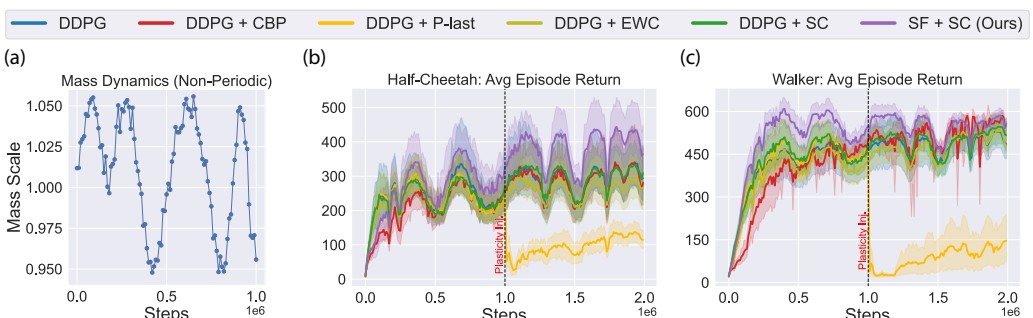
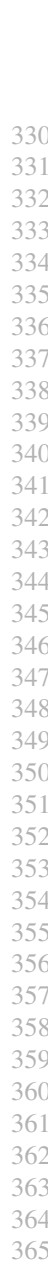

*Figure 6.* Simulating continual mass changes with a non-periodic sine function to module the embodiment's mass over time. **(a):** Example of the non-periodic sine function used. Average episode return for **(b)** Half-Cheetah and **(c)** Walker when the embodiment undergoes non-periodic mass changes. Overall, synaptic consolidation applied to SFs (SF+SC) remains effective even when mass changes become less predictable compared to periodic sine setting. Results for other embodiments are included in Appendix G.1.

last and CBP) and stability-preserving variants (EWC and SC). The results in Figure 4 show that stability-preserving models (DDPG + EWC and DDPG + SC) consistently outperformed the plasticity-injection model (DDPG + P-last and DDPG + CBP). Once again, synaptic consolidation (DDPG + SC) achieved higher learning efficiency, particularly in the more complex embodiments, Humanoid (Figure 1) and Quadruped (Figure 4a and b).

Both sets of results indicate that *stability is the primary bottleneck* as agents lacking stability fail to learn effectively in the naturalistic, continual non-stationary environments. The results further showed that the synaptic consolidation (SC) mechanism is more effective than EWC.

Next, we asked, what exactly should be stabilized: the Q-value parameters themselves, or the parameters of the underlying representations such as SFs?

### 5.2. What should be stabilized: Q-values or Successor Features?

As many stability preserving methods focus on stabilizing parameters of Q-value functions, we investigate if SFs could be a better target for consolidation. To address this, we evaluated a SFs variant combined with synaptic consolidation (SF + SC) in both slippery Four Rooms and the MuJoCo suite. The results for both slippery Four Rooms (Figure 3) and the MuJoCo suite (Figures 1 and 4) showed that SF + SC consistently improves performance compared to Q-value based consolidation. We also compared with EWC (Appendix J), which revealed that while SC is more effective than EWC overall, only the combination of SFs with SC yields consistently strong performance.

### 5.3. Robustness to Non-Periodic and Stochastic Drift

To test robustness beyond periodic drift, we replace the noisy sine modulation with either a non-periodic sine func-

tion (Appendix 11) or Ornstein–Uhlenbeck (OU) process (Appendix 12). Results from mass changes induced by a non-periodic sine function (Figure 6) show that applying SC to SFs continues to improve learning performance. Results from other embodiments (Appendix G) as well as mass dynamics induced by OU (Appendix H) affirm that applying SC on SFs remains effective.

## 6. Analyzing Multi-Timescale Contributions

To gain insights into why combining SFs and SC yields an effective model, we perform ablation studies by varying the number of consolidation variables, and we complement this with a cross-attention (Dosovitskiy et al., 2020) analysis to analyze the relative contributions of individual variables.

### 6.1. Do fast or slow timescale variables matter more for learning?

In this analysis, we varied over the number of consolidation variables (3, 6, or 9), with fewer variables yielding greater plasticity, and more variables yielding greater stability. The aim was to assess whether the inclusion of slower timescale variables improve policy learning.

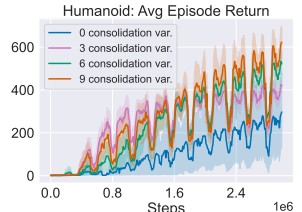

Figure 8 illustrates the results of the Humanoid embodiment (see Appendix L for more results). We found

*Figure 8.* Analysis of timescales in Humanoid. More consolidation variables improve learning efficiency, highlighting the benefit of slower timescales.

that using six or more timescale variables leads to better learning performance. Zero variables correspond to the Simple SF agent. For the slippery four rooms environment (Fig-

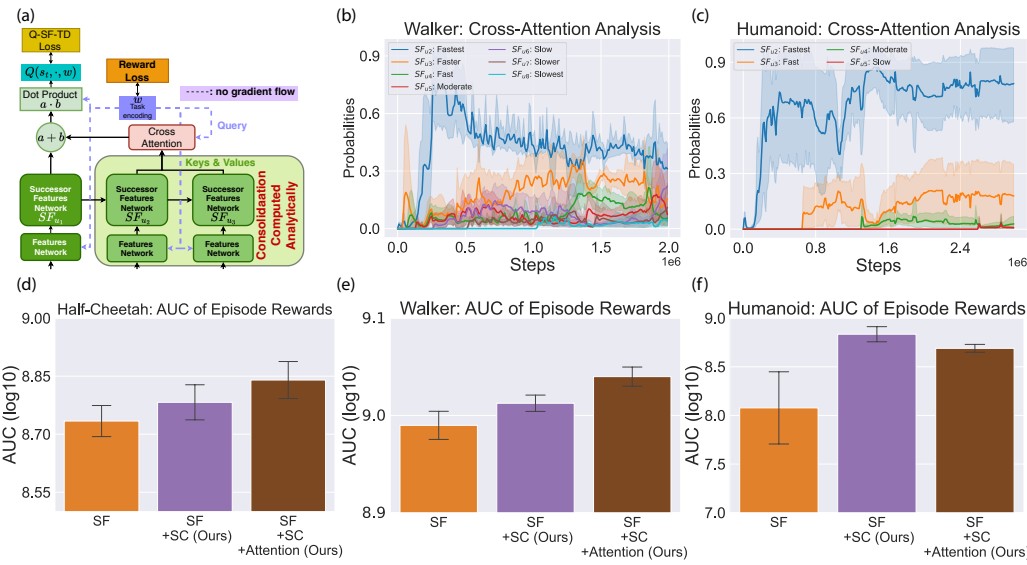

*Figure 7.* Cross-Attention analysis of individual consolidations, replacing memory recall via backflow. **(a):** Implementation design. **(b-c):** Attention probabilities over consolidation variables, where higher probability indicates greater contribution to policy learning. **(d-f):** Area under the curve (AUC) of average episode returns: simpler embodiments (Half-Cheetah, Walker) benefited, while more complex ones (Humanoid) performed worse. Full results are provided in Appendix N.

ure 23 in Appendix 6.1), we observed a similar trend: using six or nine consolidation variables leads to better performance. Together, these results demonstrated that preserving stability is crucial in naturalistic, continual settings.

### 6.2. What does cross-attention reveal about the contributions of consolidation variables?

While varying over the number of consolidation variables provides a coarse measure of their utility, it does not reveal which specific variables contribute most. At the same time, using a cross-attention mechanism also allow us to bypass the need for information to propagate gradually via the flow strength ($g_{k,k+1}$), instead providing an instant readout from all the consolidation variables (see Figure 7a).

We adapt the canonical cross-attention mechanism by letting the task-encoding vector $w$ serves as the query, while the SF consolidation variables (excluding the most plastic $SF_{u_1}$) serve as keys and values. To construct more discriminative representations, we subtracted each variable from its faster neighbor (eg. $SF_{u_k} = SF_{u_k} - SF_{u_{k-1}}$). The keys and values were layer-normalized before projected through learnable weights ($W_{\text{Keys}}, W_{\text{Values}}$), while the query vector $w$ is projected through $W_{\text{Query}}$. Attention scores were computed via the query-keys multiplication, followed by softmax activation, which was then multiplied by the Values to produce a weighted sum, thus integrating information across timescales (see Appendix M for more details). Since the deeper SF consolidation variables ($SF_{u_2}, SF_{u_3}, \ldots$) were computed analytically and not part of the computational

graph, we applied a reparameterization trick (inspired by Kingma & Welling (2013)) to enable joint training of the cross-attention mechanism with the most plastic variable ($SF_{u_1}$) using the Q-SF-TD loss (Eq. 5).

The softmax probabilities (Figure 7b and c) showed that the faster timescale variables, particularly $SF_{u_2}$ and $SF_{u_3}$, received the highest attention. However, the slower variables also contributed, more prominently in Walker (Figure 7b), as well as in Half-Cheetah and Quadruped (Figure 27 in Appendix N). Furthermore, we also observed that memory recall via attention mechanism led to improved performance in Half-Cheetah and Walker, but not in Humanoid or Quadruped (Figure 27), likely due to the additional complexity of learning the attention mechanism in more complex embodiments. Together, these findings suggest that fast timescales drive most learning, while slower ones provide complementary stability that aids performance.

## 7. Discussion

In this work, we introduced naturalistic, continual non-stationarity environments to study stability-plasticity trade-off in deep RL, and showed that stability is the primary bottleneck. By adapting a neuro-inspired synaptic consolidation mechanism with SFs, we developed a multi-timescale system that balances stability and plasticity through predictive representations. Our approach shows that combining synaptic consolidation with fast and slow SFs is key to robust learning under naturalistic, continual non-stationarity.