# OpenReview forum: "Balancing Plasticity and Stability with Fast and Slow Successor Features"
_ICML.cc/2026/Conference — ICML 2026 regular_

### Official Review · Reviewer_Mvg2 · 2026-03-06

**Soundness:** 2
**Presentation:** 2
**Significance:** 3
**Originality:** 3
**Overall Recommendation:** 2
**Confidence:** 3

**Summary:**

This paper focuses on continual non-stationarity that evolves gradually through continuous drift, which differs from prior studies that primarily consider abrupt changes with clear task boundaries. In practice, the authors induce non-stationarity through continuous stochastic drift processes, such as the Ornstein–Uhlenbeck (OU) process. They present a motivating experiment showing that stability-preserving methods, such as Synaptic Consolidation (SC), outperform purely plastic approaches. Based on this observation, the authors argue that performance degradation is primarily driven by instability rather than insufficient plasticity. To address this issue, the paper combines Successor Features (SFs) with SC across multiple timescales to promote more stable learning. Experimental results on the 3D MiniWorld and MuJoCo environments demonstrate the effectiveness of the proposed method.

**Compliance With Llm Reviewing Policy:**

Affirmed.

**Final Justification:**

This paper combines Successor Features (SFs) with Synaptic Consolidation (SC) across multiple timescales to promote more stable learning in continual non-stationary reinforcement learning. My major concerns are as follows:

(1) The authors do not provide an in-depth analysis, such as from the perspective of MDP/task structure under non-stationary setups, to clarify the motivation for integrating SFs with SC. As a result, the method appears to be a simple combination of existing techniques.

(2) More importantly, I believe it is not sound to employ the formulation of SFs $Q(s,a,w)=\psi(s,a,w)^Tw$ (Eq. 1) to handle non-stationary dynamics. The current formulation of SFs in this paper is suitable for addressing tasks that share the same state transition dynamics but differ in reward functions, which contradicts the non-stationary setup considered here, where the state transition dynamics change. In this paper, the task differences lie in the dynamics, and the task encoding vector w should therefore reflect changes in dynamics. However, the paper sets the reward weight equal to the task encoding vector, and the decomposed $\psi(s,a,w)$ still mixes changes in reward functions and dynamics. This is also not aligned with the original motivation of SFs, which aims to decouple dynamics and rewards and enable efficient adaptation to tasks with shared structure.

During the rebuttal and discussion phase, I believe the above concerns remain unresolved. Considering these issues, I maintain my score of 2 (reject).

**Key Questions For Authors:**

1. How is the task encoding vector $w$ obtained in practice? Is it initialized randomly and then learned during training?

2. In prior studies, how is non-stationarity typically introduced? What are the main differences between those setups and the one considered in this paper? In addition, how is non-stationarity generally introduced in the features?

3. In Appendix Eq. (29), why does the effective learning rate not include the term $\hat{m_t}$?

**Limitations:**

No, they do not discuss limitations.

As mentioned in the last point of the weaknesses, I think this paper may not be able to address non-stationary environments with changing state transition dynamics, since the task encoding vector corresponds to reward functions.

**Strengths And Weaknesses:**

**Strengths:**

1. From the perspective of MDP structure, the idea of integrating Successor Features (SFs) with Synaptic Consolidation (SC) is interesting, as it may help discern changes with different task structures.

2. The authors provide a motivating example to illustrate that, compared to plasticity, stability is more important under continuous non-stationarity. They also present a proof showing that updating SFs with Stochastic Gradient Descent (SGD) can preserve timescale information, whereas Adaptive Moment Estimation (Adam) cannot.

3. The experiments in the paper are comprehensive. In addition to the overall evaluation on MiniWorld and MuJoCo, the authors also examine robustness to non-periodic changes and analyze the contributions of the consolidation variables.

**Weaknesses:**

1. My major concern lies in the insufficient analysis of the motivation for integrating SFs with the SC method in a continuous non-stationary environment. Simply stating that “the role of representations is largely unstudied” and then introducing SFs is not sufficiently convincing. More intuition and in-depth analysis are needed to support the claim that “SFs reduce interference and enable stable learning in continual learning scenarios with gradual changes.” Without such justification, the proposed method appears to be merely replacing the Q-network with SFs on top of the prior Benna-Fusi [1] framework.

2. The current formulation of the non-stationary setup may be inaccurate, which makes it difficult to clearly understand the distinctions between prior studies and the setup considered in this paper. In Section 3.1, the preliminaries are still presented under the standard reinforcement learning framework. However, since the non-stationarity in this paper is induced by changes in the action space or the agent’s body mass, the transition probability function of the MDP should also vary over time. Therefore, it seems inaccurate to formalize the non-stationary environment using a single fixed MDP. I suggest that the authors consider adopting a more appropriate formalization of non-stationary reinforcement learning, such as [2].

3. I have some doubts about the soundness of employing SFs to address environments with gradually changing state transition dynamics. SFs are generally used to facilitate transfer across tasks that share the same dynamics but differ in reward functions [3]. In Equations (4)-(6) of this paper, the original reward weight in SFs [3] equals to the task encoding vector $w$. However, in USFs [4], these two quantities are generally not identical. Moreover, as mentioned in the second weakness, the dynamics of the MDP change over time in the setup considered in this paper. In such a case, the task representation (i.e., $w$) should reflect the changes in transition dynamics rather than unchanged reward functions. Please correct me if my understanding is incorrect.

[1] Kaplanis C, Shanahan M, Clopath C. Continual reinforcement learning with complex synapses[C]//International Conference on Machine Learning. PMLR, 2018: 2497-2506.

[2] Poiani R, Tirinzoni A, Restelli M. Meta-reinforcement learning by tracking task non-stationarity[J]. arXiv preprint arXiv:2105.08834, 2021.

[3] Barreto A, Dabney W, Munos R, et al. Successor features for transfer in reinforcement learning[J]. Advances in neural information processing systems, 2017, 30.

[4] Borsa D, Barreto A, Quan J, et al. Universal successor features approximators[J]. arXiv preprint arXiv:1812.07626, 2018.

---

> ### Author Rebuttal · Authors · 2026-03-31
>
> Thank you for your feedback.
>
> # 1. Role of representations is largely unstudied
>
> Our statement is based on an in-depth review of the consolidation literature. As discussed in the introduction (paragraph 4, lines 46–67), prior work on synaptic consolidation in reinforcement learning has primarily focused on stabilizing value functions [1,3] or policies [2].
>
> In contrast, the role of consolidation at the level of learned representations has received comparatively limited attention. As also noted by reviewers uAzh and RoCq, this area is under explored as most existing approaches focus on Q-values or policy parameters, rather than on the underlying feature representations.
>
> # 2. Justifications for SFs reduce interference
>
> Q-values are a relatively entangled quantity. They jointly reflect environment dynamics, policy, and reward. Under gradual non-stationarity, these factors may all shift over time, meaning that the target represented by the Q-network can change substantially even when some underlying structure is shared across tasks. As a result, directly consolidating Q-value parameters can limit adaptation.
>
> By contrast, SFs provide predictive representation and separates from the reward information, which gives it a more principled basis for stability. Consolidation can act on the reusable predictive structure, while downstream adaptation can still occur through the reward-dependent components.
>
> # 3. Insufficient analysis of the motivation for integrating SFs with the SC
> Building on our earlier discussion on why SFs can reduce interference, our motivation here is that the effectiveness of synaptic consolidation depends on the representation being stabilized. In particular, if the underlying representation is highly entangled (as in Q-values), consolidation may preserve outdated information.  Our results (Appendix K) support SFs to be a target.
>
> # 4. Extension from single fixed MDP
> In our work, the non-stationarity arises from continuous changes in environment parameters (e.g., action perturbations or variations in the agent’s body mass), which directly affect the transition dynamics. To better reflect this, we will revise the formulation to explicitly model the environment as a time-varying (non-stationary) MDP, where the transition function depends on an additional parameter:
>
> $P(s′ | s, a; \omega_t​),$
>
> where $\omega_t \in \Omega$  represents a time-dependent latent variable (such as the mass) governing the environment dynamics.
>
> We will revise Section 3.1 to make this formulation explicit.
>
> # 5. Clarifications relating to task representation
> We follow the formulation in Universal Successor Features  [5], where the task representation w is learned through a reward prediction loss and simultaneously used as a task-encoding vector. As noted in [5] (page 4, last paragraph) and in [4] (page 4, first paragraph), treating w as both the reward predictor and task embedding corresponds to a valid special case of the general USF framework, which leads to Eq. 1 in our manuscript.
>
> # 6. How is the task encoding vector obtained?
> The task-encoding vector is randomly initialised and learned using the reward prediction loss (Eq. 4). This follows the same approach used in previous SFs work [4, 5, 6].
>
> # 7. How is non-stationarity typically introduced in prior studies?
> In prior studies, non-stationarity is most commonly introduced in a discrete manner. A standard approach is to sample task parameters (e.g., reward functions or environment properties) from a distribution at specific intervals, leading to abrupt changes between tasks [7, 8]. Another common setup involves switching between distinct tasks or environments, such as different games or domains [9, 10].
>
> # 8. Effective learning rate
> $\hat{m}_t$ is not considered part of the effective learning rate, as it represents the first moment estimate (i.e., the gradient term, as shown in Eqs. 20 and 29). This makes Eq. 29 analogous to a standard gradient descent update, where the quantity $\frac{\alpha}{\sqrt{\hat{v}_t} + \epsilon}$ acts as the effective learning rate for the SF $\psi(s,a)$
>
> [1] Kaplanis et al., "Continual reinforcement learning with complex synapses."
>
> [2] Kaplanis et al., "Policy consolidation for continual reinforcement learning."
>
> [3] Kirkpatrick, James, et al. "Overcoming catastrophic forgetting in neural networks."
>
> [4] Chua, Raymond, et al. "Learning successor features the simple way."
>
> [5] Borsa, Diana, et al. "Universal successor features approximators."
>
> [6] Barreto, André, et al. "Successor features for transfer in reinforcement learning."
>
> [7] Dohare, Shibhansh, et al. "Loss of plasticity in deep continual learning."
>
> [8] Anand, Nishanth, and Doina Precup. "Prediction and control in continual reinforcement learning."
>
> [9] Abbas, Zaheer, et al. "Loss of plasticity in continual deep reinforcement learning."
>
> [10] Powers, Sam, et al. "Cora: Benchmarks, baselines, and metrics as a platform for continual reinforcement learning agents."

---

> > ### Author Rebuttal · Reviewer_Mvg2 · 2026-04-04
> >
> > Thank you for your rebuttal. I still find the explanation regarding the motivation and justification for using Successor Features (SFs) to reduce interference insufficient and not fully convincing. SFs are proposed to decouple state transition dynamics from reward functions, enabling efficient transfer across tasks that share the same transition dynamics but differ in reward functions. This provides a principled framework for efficient transfer, but it may not be related to reducing interference or improving stability.
> >
> > I remain concerned about the soundness of the paper. Regarding the third weakness, although treating w as both the reward predictor and the task embedding corresponds to a valid special case of the general USF framework, this formulation relies on the assumption that tasks differ in reward functions while sharing the same transition dynamics. In this paper, however, the non-stationarity is introduced by changes in the transition dynamics. This appears to violate the underlying assumptions of the SF/USF framework, and the proposed method may not be suitable for addressing non-stationary dynamics settings.
> >
> > Considering these points, I will maintain my original score.

---

> > > ### Author Response · Authors · 2026-04-07
> > >
> > > # Soundness of the paper
> > >
> > > Thank you for your follow-up and for highlighting this important concern.
> > >
> > > ***In our work, we do not claim that Successor Features (SFs) themselves are a mechanism for improving stability.***
> > >
> > > As you pointed out, in the canonical formulation, such as the SF/USF framework [5], "SFs are proposed to decouple state transition dynamics from reward functions, enabling efficient transfer across tasks that share the same transition dynamics but differ in reward functions."
> > >
> > > Our claims regarding interference are specific to this setting and are supported by prior work [4]. In particular, in the 3D Four Rooms environment ***(Figures 3 and 4 in [4])***, where the layout and transition dynamics remain fixed and only the reward function changes, the SF agent (orange) consistently outperforms the DQN agent (blue). This behavior arises because the value function is decomposed into a shared predictive representation (SFs) and task-specific weights (task-encoding vector), reducing the need to overwrite previously learned, entangled information in the value function, and thereby mitigating interference.
> > >
> > > Mathematically, this decomposition can be written as:
> > >
> > > $Q(s,a, w) = \psi(s,a,w)^T w$
> > >
> > > where $\psi(s, a, w)$ denotes the task-dependent SFs (as shown in [4]), capturing expected future feature occupancies under the current transition dynamics, and $w$ is the task-encoding vector.
> > >
> > > In the canonical setting where transition dynamics are fixed and only the reward function changes, this decomposition enables efficient adaptation by updating only the task-encoding vector $w$, while reusing $\psi(s, a, w)$, thereby reducing interference (as shown in [4]).
> > >
> > > ***However, we agree that this property does not directly extend to settings with continuously changing transition dynamics, which is the primary focus of our work.*** In this setting, $\psi(s,a,w)$ must also adapt to reflect evolving dynamics (i.e. slippery actions in Four Rooms and mass in Mujoco). ***Importantly, since these changes are gradual and previously encountered dynamics may be revisited, preserving learned representations (SFs) becomes beneficial.***
> > >
> > > This motivates the use of synaptic consolidation (SC) on SFs: by stabilizing $\psi(s,a, w)$ across time, the agent can ***retain useful predictive representations (SFs) and more efficiently re-adapt when similar transition dynamics reoccur.*** This is supported by our empirical results in Figures 21-22 in Appendix K, which show that SFs alone do not consistently achieve strong performance under changing transition dynamics, whereas applying SC leads to more stable and efficient learning.
> > >
> > > For example, in the 3D Four Rooms environment with slippery action dynamics (Figure 3 in our main text), one can see that when the SF agent did not consistently learn better. Notably, in Figure 3b, during the transition from Exposure 1 Task 2 to Exposure 2 Task 1, the SF agent (orange) requires substantially more environment steps to recover performance. In contrast, when synaptic consolidation is applied (SF + SC, purple), the agent adapts more quickly and consistently improves across task exposures.
> > >
> > > We observe similar trends in the MuJoCo experiments (Figure 22 in Appendix K). While SFs (orange) outperform DDPG (blue) in simpler embodiments such as Half-Cheetah and Walker, they do not consistently outperform DDPG in more complex embodiments such as Quadruped and Humanoid. Crucially, consistent improvements across all embodiments are only achieved when synaptic consolidation is applied (SF + SC, purple).
> > >
> > > ***These results demonstrate that stability under continuous changing dynamics is not provided by SFs alone, but by the synaptic consolidation mechanism.***
> > >
> > > Taken together, our results suggest a complementary relationship:
> > > 1. While SFs can help reduce interference in settings with same transition dynamics (reward changes) as shown in [4],
> > > 2. SC is necessary to preserve stability under continuous changing dynamics as shown in our paper.
> > >
> > > In summary, our work shows that SFs alone are insufficient to address continuous non-stationarity in transition dynamics. Instead, when used as a representation in conjunction with synaptic consolidation, they enable robust learning under continuous dynamics changes.
> > >
> > > [4]  Chua, Raymond, et al. "Learning successor features the simple way." NeurIPS (2024)
> > >
> > > [5] Borsa, Diana, et al. "Universal successor features approximators." (2018).

---

### Official Review · Reviewer_RoCq · 2026-03-07

**Soundness:** 3
**Presentation:** 2
**Significance:** 3
**Originality:** 3
**Overall Recommendation:** 5
**Confidence:** 4

**Summary:**

This work explores a novel problem in continual RL–stability-plasticity tradeoff–where the agent is continually learning in a dynamically changing environment. They empirically show that the method that solely favours plasticity fails to solve the tasks. They propose a novel method inspired by two neuroscientific theories: synaptic consolidation and successor representation, to balance between stability and plasticity of neural networks. Empirically, they show that the proposed method outperforms the vanilla RL methods in these nonstationary environments.

**Compliance With Llm Reviewing Policy:**

Affirmed.

**Final Justification:**

My concern is fully addressed by the rebuttal. **I increased the score to 5.**

**Key Questions For Authors:**

- What computation overhead does this method add? Is it much slower than the normal RL with SF training?
- How much do flow strengths affect the performance? It’s probably good to show ablation studies if you have them.
- How are Figure 3 (Main) and Figure 14 (Appendix) different? Does Figure 14 have the same slippery dynamics?
- Why do you subtract the variable from its faster neighbours instead of using the raw variable for cross-attention analysis?
- How do you use the reparameterization trick? I do not see any stochastic nodes here.

**Limitations:**

I did not see any limitations mentioned. I suggest adding these points or any other points that the authors think are suitable.
- SC+SF only works with SGD, but not ADAM.
- Computational overhead during training
- Some extreme non-stationarity, like in multi-agent learning, is not explored

**Strengths And Weaknesses:**

The paper is very interesting, and I like it. I will raise the score to 5 if the authors follow some of the suggestions below.

## Strengths
### 1. Novel method and evaluation.
The authors propose combining neuroscience-inspired synaptic consolidation into the successor features to tackle non-stationarity (dynamically changing environments). The proposed method and results are insightful to both machine learning and computational neuroscience.

### 2. Solid technical execution.
The paper is full of analysis and some interesting analytical techniques like cross-attention. The tasks are modified from standard, widely accepted RL benchmarks. The results are convincing. Theoretical analysis is insightful and could be another novel contribution.

### 3. Rich contextualisation.
The authors really respect the prior work and clearly state what is novel and what is add on (i.e., using SF and Q function).

### 4. Clarity.
The main part of the paper is well-written and a pleasure to read. The only problem is in the appendix, explained below.

## Weaknesses
### 1. Framing:
It is unclear how the authors show a diagnosis of plasticity and stability tradeoff, which is one of the main novel contributions. Specifically, plasticity injection may not be necessary because the agent might not have lost it yet. To encourage the loss of plasticity, the agent may need to be trained with more than ten tasks. I recommend that the authors remove the claim about plasticity and focus on the stability of the learning and the non-stationarity of the environments. This reframing would greatly reduce the paper’s weakness.

**Another potential solution:** If the authors would like to stand on the original framing, you could add a Plasticity test in the Appendix where an agent is trained for $10-20$ times longer with more diverse tasks (physical parameters) to see if the proposed SF + SC eventually suffers from loss of plasticity or if the multi-timescale consolidation actually protects against it.

### 2. No quantification of non-stationarity:
I suggest showing how much mass, or physical variables, are perturbed and how much they affect learning dynamics. The authors could divide non-stationarity levels into three: mild, moderate, and severe.


### 3. Some strong baselines are missing:
The paper mainly applies the method to DDPG and DQN. They are outdated, and PPO could perform much better. Especially, there is a mathematical and empirical proof that multi-agent variants of PPO can deal with non-stationarity due to the use of a trust region (See Sun et al. https://dl.acm.org/doi/10.5555/3545946.3598613). My intuition is that trust-region acts as an implicit form of stability. I am not asking for a new experiment with adding SF+SC into PPO at this review stage. I suggest that comparing the proposed method against vanilla PPO on these tasks could significantly reduce the weakness.

### 4. Clarity
The appendix seems to have a lot of interesting results, but the authors choose to leave them empty without saying anything. For example, Figure 14 has not been mentioned elsewhere. I have no idea why Figure 14 (Appendix) and Figure 3 (Main) are different. There is no experimental setting mentioned here. Moreover, all the methods seem to have the same results. I would die on the hill to see all parts of the appendix filled with interpretation, analysis, and discussion.

### 5. Discussion is very short, looks unprofessional, and sounds like a conclusion.
Ideally, the discussion should analyse and criticise the paper. Specifically, the limitations and unexpected results should be elaborated.
I suggest some points that could be added in the **limitations**.

### 6. Minor:
- Y-axis name is missing in Fig. 3, please add them.
- It’s better to explain the intuition behind the SC update rule, making it easier to grasp the concept for readers who are not familiar with SC.

---

> ### Author Rebuttal · Authors · 2026-03-31
>
> Thank you for your positive feedback.  Please let us know if there is further clarification we can provide.
>
> # 1. Framing
> ***We would like to clarify that we do not claim that our approach mitigates the loss of plasticity.*** We agree that, in our current setting, agents may not be sufficiently pushed into a regime where loss of plasticity becomes the dominant failure mode. As such, our results should not be interpreted as plasticity is preserved.
>
> The synaptic consolidation mechanism used in our approach incorporates a multi-timescale structure in which fast timescale SFs rapidly adapt to new information, while slower timescales provide stability through consolidation. This can be interpreted as an architectural mechanism that supports plasticity at short timescales while maintaining stability, although we do not directly measure whether plasticity is preserved.
>
> We will revise the abstract, intro and discussion, to more clearly distinguish between plasticity (the capacity to adapt) and adaptation (observed performance improvement), and to clarify that our results provide evidence of improved adaptation and robustness, rather than a direct diagnosis of plasticity preservation.
>
> # 2. Quantification of non-stationarity
> We are currently working on the experiments and will report back soon.
>
> # 3. DDPG outdated
> We chose DDPG because our implementation already incorporates the core TD3 components, which are twin critics and target-policy smoothing. This follows the same design used in prior MuJoCo studies on DDPG and successor features [1, 2, 4]. The only TD3 feature we omit is the delayed policy update, as in our experiments, the agent learns stably without it. Importantly, using this DDPG variant ensures direct comparability with past SF and continual RL work that adopts identical architectures [1, 3, 4].
>
> # 4.  DQN outdated
> We chose this model because the backbone of the best performing plasticity mitigation methods, such as plasticity injection and continual backprop, rely on this as the backbone and to ensure fair comparison.
>
> # 5. Comparison with PPO
> Currently running some experiments.
>
> # 6. Appendix
> Thank you. For each section in the appendix, we will include a short write-up on what the section is about and the aim of the section.
>
> # 7. Discussion
> Thank you and we will update the figure.
>
> # 8. Computational overhead
> Indeed, using the Synaptic Consolidation does incur some additional computational complexity. We present an analysis based on the frames per second during training in Appendix Q. Unsurprisingly, using more consolidation variables has a negative effect on the overall computational complexity.
>
> # 9. Flow Strength
> We did not do an exhaustive sweep on these hyper-parameters (flow strength and capacity) as it is intractable. The more fine-grained the difference of the effective learning rate is between the variables, the more variables are required to capture the overall timescale of the training duration. We decide based on the trade-off complexity between the number of consolidation variables and the learning performance.
>
> # 10. Figures 3 and 14
> Figure 14 in the Appendix is redundant and we will remove it. Figure 14 is a subset of Figure 3, from the same results.
>
> # 11. Cross Attention Analysis and the re-parameterization trick
> When incorporating the cross-attention mechanism, we introduced several architectural modifications to ensure proper gradient computation. Crucially, to preserve the integrity of the multi-timescale SFs, we prevent gradients from directly updating the SF networks at each timescale. Allowing such updates would override the analytically computed synaptic consolidation dynamics (Eqs. 8 & 9).
>
> To enable the attention mechanism to be trained through the SF-TD update, we construct inputs to attention using differences between consecutive timescales. Specifically, each consolidation variable is transformed by subtracting its faster-timescale neighbor. This ensures that each variable captures only the incremental, timescale-specific information, making the representations more discriminative.
>
> These transformed consolidation signals are then combined with $SF_u1$ before computing the TD loss. This design ensures that gradients flow only through $SF_u1$, while still allowing the model to leverage information from slower timescales.
>
> Inspired by the reparameterization trick used in [5]. In our case, the consolidation variables play a role analogous to auxiliary noise variables (Eq 4 in [5]), while SF_u1 acts as the primary learnable component.
>
> [1] Chua, Raymond, et al. "Learning successor features the simple way."
>
> [2] Fujimoto et al., "Addressing function approximation error in actor-critic methods."
>
> [3]  Yarats, Denis, et al. "Mastering visual continuous control: Improved data-augmented reinforcement learning." a
>
> [4] Touati, Ahmed et al., "Does zero-shot reinforcement learning exist?."
>
> [5] Kingma, Diederik P., and Max Welling. "Auto-encoding variational bayes."

---

> > ### Author Rebuttal · Reviewer_RoCq · 2026-04-02
> >
> > I thank the author for the responses. However, due to multiple concerns that take a significant amount of time to resolve, the paper may need substantial revision to resolve all of them. Therefore, I maintain my score.

---

> > > ### Author Response · Authors · 2026-04-07
> > >
> > > # 1. PPO Comparison
> > > We agree that trust-region methods such as PPO can provide an implicit form of stability and that including PPO as a baseline helps strengthen the empirical evaluation.
> > >
> > > Therefore, we evaluated PPO alongside DDPG, SF, and their variants with synaptic consolidation (DDPG + SC, SF + SC) on two MuJoCo embodiments: Humanoid and Half-Cheetah. These environments were chosen to cover different levels of complexity, with Humanoid representing a more challenging setting.
> > >
> > > Results of comparison with PPO: https://imgur.com/uwcfTs9
> > >
> > > Across both embodiments, we observe that ***PPO achieves lower average episode returns*** compared to the best-performing variants, particularly in the more complex Humanoid setting. Notably, this occurs ***despite PPO leveraging parallelized data collection and consuming more samples during training***.
> > >
> > > These results suggest that while trust-region updates can provide a degree of stabilization, they may not be sufficient to maintain performance in our setting.
> > >
> > > # 2. Quantification of non-stationarity
> > >
> > > Similar to above, we performed additional analyses using Humanoid and Half-Cheetah.
> > >
> > > As suggested, we introduced three levels of mass variation to characterize non-stationarity: mild (25%), moderate (50%), and severe (100%), defined relative to the maximum perturbation allowed before the physical simulation becomes unstable. ***We note that all results reported in the manuscript correspond to the severe (100%) setting.***
> > >
> > > Quantification analysis for Humanoid: https://imgur.com/ic5l8Q9
> > >
> > > Quantification analysis for Half-Cheetah: https://imgur.com/l4u9rUt
> > >
> > > Across both embodiments, we observe consistent trends: Plasticity-preserving methods (CBP, P-last) are less effective than approaches incorporating synaptic consolidation (SC). Under moderate and severe conditions, where the agent experiences substantial changes in transition dynamics, applying SC to SFs (SF + SC) yields the best performance. In contrast, under mild changes, the benefits of SC are limited, as the environment remains close to stationary.
> > >
> > > # 3. Discussion expansion
> > >
> > > We present a revised Discussion section:
> > >
> > > In this work, we introduced naturalistic continual non-stationarity benchmarks to study the stability–plasticity trade-off in deep RL. Across both navigation and MuJoCo settings, our results suggest that stability is often the primary bottleneck under continuous environmental drift, and that maintaining plasticity (e.g., reset-based methods) is insufficient. It is important to note that we do not directly measure plasticity and evaluate adaptation through performance under continuous non-stationarity.
> > >
> > > We find that combining SC with SFs yields a multi-timescale learning system, where slower components promote stability and faster components support adaptation. Cross-attention analysis further indicates that different timescales contribute differently depending on task complexity.
> > >
> > > However, several limitations remain. First, the method is restricted to SGD-based updates, as combining it with adaptive optimizers like Adam can lead to instability. Second, it introduces computational overhead due to non-parallelizable analytical updates, which grows with the number of timescales. More extreme settings such as multi-agent environments remain an interesting direction for future research.
> > >
> > > # 4. Manuscript require substantial revision
> > >
> > > ***We respectfully disagree that the concerns raised necessitate substantial revision of the manuscript.*** The core issue identified relates to how we frame plasticity versus stability, rather than to the experimental design or the validity of the results themselves.
> > >
> > > Concretely, we will make the following revisions:
> > > - add a one-line clarification in the discussion stating that we do not directly measure plasticity; instead, we evaluate adaptation through performance under continuous non-stationarity,
> > > - clarify in Section 3.3 that the multi-timescale mechanism emphasises rapid adaptation and stability, with fast timescale components enabling behaviour consistent with functional plasticity, defined as the observable ability of an agent to adapt its behaviour under changing dynamics,
> > > - replace Line 26 in the abstract to: “... and use them to examine how stability and adaptation affect performance under continuous environmental change.”
> > >
> > > Importantly, these are clarifications and do not affect the methodology, experiments, or conclusions of the work. ***The central narrative that stability is a key bottleneck under naturalistic non-stationarity, and that synaptic consolidation improves robustness remains unchanged.***
> > >
> > > We therefore believe that, based on the reviewers’ concerns, the revision required is minor in scope. If the reviewer has other concerns that prevent them from raising their score, then we note that those should have been raised earlier in the review/rebuttal process.

---

### Official Review · Reviewer_Z4JU · 2026-03-12

**Soundness:** 3
**Presentation:** 3
**Significance:** 2
**Originality:** 3
**Overall Recommendation:** 4
**Confidence:** 3

**Summary:**

This work studies continual reinforcement learning under gradually changing environments and proposes to combine Successor Features (SF) with multi-timescale synaptic consolidation. Instead of consolidating Q-values or policies as in previous work, the proposed method applies consolidation directly to the SF representations, aiming to stabilize predictive representations of the environment dynamics. The approach is evaluated on several environments including MuJoCo continuous control tasks and a 3D Four Rooms environment with gradual non-stationarity, showing improved performance compared with several stability- and plasticity-oriented baselines.

**Compliance With Llm Reviewing Policy:**

Affirmed.

**Final Justification:**

Thanks to the authors for the detailed reply. I will retain my initial positive scores as they reflect the current state of the submitted manuscript.

**Key Questions For Authors:**

1. Is it possible that some baselines (e.g., EWC) underperform because of limited model capacity? Under the same computational budget, would increasing the network size improve their performance and reduce the gap with the proposed method?

2. The current experiments mainly use sinusoidal drift. Since this form of drift has a certain structure, it would be interesting to see whether the method remains effective under other types of non-stationarity, such as stochastic drift (e.g., Ornstein–Uhlenbeck processes).

**Limitations:**

yes

**Strengths And Weaknesses:**

Strengths：

- The paper evaluates the proposed method across multiple environments and embodiments, and compares it against a range of baselines (e.g., EWC, CBP), providing a relatively systematic experimental study.
- The paper is generally well written, and includes extensive supplementary results and ablation studies, which help clarify the behavior of the proposed method.
- The focus on gradual non-stationarity instead of discrete task boundaries is well motivated and reflects more realistic scenarios for continual reinforcement learning.

Weaknesses：
- The cascade consolidation mechanism introduces additional variables and sequential updates. This may increase computational cost and potentially limit the scalability of the SF network size. It would be helpful to better understand how this overhead scales with larger models or more complex environments.
- The interpretation that slower consolidation variables store long-term knowledge is plausible, but currently seems more observational or explanatory. The paper lacks a clear experimental validation demonstrating how these slow variables specifically contribute to long-term retention or transfer.

---

> ### Author Rebuttal · Authors · 2026-03-31
>
> Thank you for your feedback and your positive comments that our manuscript is well written, and for providing a systemic set of experiments. We appreciate the opportunity to clarify and enhance our manuscript based on your observations. Please let us know if there is further clarification we can provide.
>
> # 1. Computational cost and Scalability
> Thank you for your comment and the opportunity to address this concern.
>
> We provide a detailed computational analysis in Appendix Section Q, where we report training throughput (frames per second) for both the 3D Four Rooms and MuJoCo environments (Figures 30 and 32). As expected, incorporating the Synaptic Consolidation (SC) mechanism introduces additional computational overhead, and this overhead increases with the number of consolidation variables (Figures 31 and 33). This reflects a trade-off between computational cost and the number of timescales used.
>
> To further examine scalability, we conducted an additional analysis during the rebuttal using the humanoid embodiment. Specifically, we increased the model capacity of DDPG and its variants (DDPG, DDPG + EWC, DDPG + CBP, DDPG + P-last) beyond that of the SF-based models (SF, SF + EWC, SF + SC).
>
> Link to figure: https://imgur.com/a/wl8gjp8
>
> In this figure, the x-axis shows the number of parameters (log scale), and the y-axis shows performance measured by area under the curve (AUC). Notably, the default DDPG baseline was already tuned and uses more parameters than the SF model. Despite further increasing their capacity, DDPG and its variants do not match the performance of our method (SF + SC), which achieves the best performance with significantly fewer parameters.
>
> These results suggest that improved performance does not simply arise from increased model size, but from the combination of predictive representations and multi-timescale consolidation. From a scalability perspective, this indicates that our approach is parameter-efficient, achieving strong performance without requiring large model capacity.
>
> # 2. Experimental validation on the contributions of slow variables
>
> Thank you for your comment and the opportunity to address this concern.
>
> From a theoretical perspective, the slow variables are governed by their effective learning rates, as defined by the consolidation dynamics (Eqs. 8 and 9). As the variable index $k$ increases (i.e., deeper in the consolidation chain), the flow terms $g_{k−1,k}​$ and $g_{k,k+1}$ decrease, while the capacity $C_k$​ increases. Together, these terms determine the effective learning rate of each variable. Consequently, variables at larger $k$ evolve more slowly and retain information over longer timescales.
>
> This leads to a key implication: when fewer variables are used (smaller $k$), the system is dominated by faster timescales, resulting in quicker overwriting of previously stored information. In contrast, incorporating more variables introduces slower timescales that preserve information for longer durations.
>
> We validate this experimentally by sweeping the number of variables ($k=3,6,9$) in both the 3D Four Rooms and MuJoCo environments. The results (Figure 8 in the main manuscript; Figures 23 and 24 in Appendix L) show that configurations with more variables (i.e., longer timescales) consistently achieve better performance than those with fewer variables.
> These results provide empirical support for the role of slow variables: increasing the number of consolidation variables improves stability by enabling information to be retained across longer timescales, which in turn leads to improved learning performance.
>
>
> # 3. Non-periodic and Ornstein–Uhlenbeck (OU) random walk settings
>
> Thank you for your comment, and we apologise if this was not clearly presented in the manuscript.
>
> In the main paper, we include results under both periodic (Figures 3 and 4) and non-periodic sinusoidal drift (Figure 6). For a more comprehensive evaluation of the non-periodic setting, we refer the reader to Figure 16 in Appendix G.
>
> We also evaluate our method under stochastic, non-periodic dynamics using Ornstein–Uhlenbeck (OU) random walk processes (Figure 17, Appendix H). Additional results for the Cheetah and Walker embodiments under OU drift are provided here: https://imgur.com/PHGOG4D.
>
> Across both non-periodic sinusoidal and OU settings, we consistently observe that applying Synaptic Consolidation (SC) to Successor Features (SF + SC) improves learning performance, particularly in the Cheetah and Walker embodiments. In more complex embodiments such as Quadruped and Humanoid, SF + SC achieves performance comparable to applying SC directly to Q-value parameters (DDPG + SC). Importantly, across all settings, SC-based methods remain more effective than alternatives such as Elastic Weight Consolidation (EWC) and Continual Backpropagation (CBP).

---

> > ### Author Rebuttal · Reviewer_Z4JU · 2026-04-01
> >
> > Thank you for the clarifications and additional experiments. I will maintain my positive score.

---

### Official Review · Reviewer_uAzh · 2026-03-13

**Soundness:** 2
**Presentation:** 2
**Significance:** 2
**Originality:** 2
**Overall Recommendation:** 3
**Confidence:** 4

**Summary:**

This paper addresses the stability-plasticity dilemma in online reinforcement learning. Unlike prior work that primarily focuses on mitigating plasticity loss, the authors argue that stability is the more critical bottleneck and propose leveraging successor features (SF) to preserve it. In addition, the paper incorporates synaptic consolidation techniques (e.g., EWC) to further enhance stability in the online RL setting. Empirical results demonstrate improved performance over existing baselines, suggesting that successor features can be an effective tool for maintaining stability.

**Compliance With Llm Reviewing Policy:**

Affirmed.

**Key Questions For Authors:**

1. Can you provide a more rigorous explanation (theoretical or empirical) for why successor features are effective at preserving stability?

2. How does the method perform on standard benchmarks (e.g., Atari, DMControl) beyond the environments presented?

3. Could you include a more detailed ablation isolating the effect of synaptic consolidation versus successor features?

**Limitations:**

Yes

**Strengths And Weaknesses:**

Strengths

1. Novel application of successor features to the stability-plasticity problem. The idea of repurposing successor features — a concept well-established in transfer learning and successor representation literature — to address stability loss in online RL is creative and, to my knowledge, relatively unexplored. The empirical gains over existing methods further support the promise of this direction.

Weaknesses

1. Insufficient theoretical or intuitive justification for why SFs improve stability. While the empirical results show that the proposed method works well, the paper lacks a convincing explanation of why successor features are particularly suited to alleviating stability loss. A deeper analysis — whether theoretical, through ablations, or via illustrative examples — would significantly strengthen the contribution.

2. Limited contribution of synaptic consolidation. Although synaptic consolidation is presented as a key component of the approach, its empirical impact appears marginal. In several cases, the AUC results show minimal differences compared to EWC alone, suggesting that the performance gains are primarily driven by the successor feature component rather than the consolidation mechanism. The paper would benefit from a more honest discussion of each component's relative contribution.

3. Narrow experimental evaluation. The experimental environments (e.g., slippery settings) seem inherently favorable to successor feature-based methods, as the underlying transition dynamics are well-suited to SF decomposition. It remains unclear whether the proposed approach generalizes to more widely adopted benchmarks such as Atari or DeepMind Control Suite. Evaluation on these standard domains would substantially improve confidence in the method's broader applicability.

---

> ### Author Rebuttal · Authors · 2026-03-31
>
> Thank you for your feedback and recognising the novelty of our contribution. We appreciate the opportunity to clarify and enhance our manuscript based on your observations. Please let us know if there is further clarification we can provide.
>
> # 1. Insufficient theoretical or intuitive justification for why SFs improve stability.
> We apologise for any confusion.
>
> ***In our work, we do not claim that Successor Features (SFs) themselves are a mechanism for improving stability.*** Instead, SFs provide a representation with strong generalisation properties, which makes them a suitable candidate for transfer and continual learning [1–4].
>
> Intuitively, SFs decouple environment dynamics from reward by learning predictive representations of expected future feature occupancies. This structure allows the agent to reuse representations across related tasks or gradually changing environments, which can reduce the need to relearn representations from scratch. However, this alone does not guarantee stability.
>
> In our setting, the observed improvements in stability arise primarily from the integration of SFs with the synaptic consolidation (SC) mechanism. SC explicitly enforces stability across multiple timescales, while SFs provide a structured representation that is more amenable to reuse under non-stationarity.
>
> Empirically, prior work has shown that SF-based methods can be effective in continual reinforcement learning settings with changing dynamics [5]. In our experiments, we similarly observe that SFs perform well under naturalistic continuous changes (e.g., Figure 3 in the main text, Figures 21–22 in the appendix). Importantly, the additional stability gains are achieved when SFs are combined with SC, rather than from SFs alone.
>
> # 2. Limited contribution of synaptic consolidation
>
> We respectfully disagree with the assertion that the contribution of synaptic consolidation (SC) is limited. In Appendix Section J, we provide a systematic comparison between Elastic Weight Consolidation (EWC) and SC, applied to both Q-value function parameters and Successor Features (SFs). The results are shown in Figures 19 (3D Four Rooms) and 20 (MuJoCo).
> Across both environments, SC consistently outperforms EWC. This holds even when applied to Q-value parameters (dark green: DQN + SC, DDPG + SC vs. light green: DQN + EWC, DDPG + EWC), indicating that SC provides a stronger stability mechanism than standard regularization-based approaches.
>
> More importantly, when applied to SFs, the performance gains are substantially more pronounced (purple: SF + SC vs. pink: SF + EWC), demonstrating that SC plays a significant role in stabilizing learned representations under continuous non-stationarity.
>
> # 3. Performance gains are primarily driven by the successor feature
>
> We agree that SFs provide a strong representation candidate for generalisation and continual reinforcement learning. ***However, our results indicate that the performance gains cannot be attributed to SFs alone.*** In Appendix Section K, we present a direct comparison between SFs without synaptic consolidation (SF, orange) and SFs with synaptic consolidation (SF + SC, purple).
>
> Across both the 3D Four Rooms environment (Figure 21) and the MuJoCo embodiments (Figure 22), SF + SC consistently outperforms SF alone. This improvement is observed across multiple settings and tasks, indicating that the synaptic consolidation mechanism plays a critical role in enhancing performance under continual non-stationarity.
>
> # 4. Narrow experimental evaluation
>
> Thank you for this important comment.
>
> We would like to clarify that our evaluation is not limited to slippery navigation environments. In addition to the 3D Four Rooms setting, ***we also evaluate our approach on the DeepMind Control Suite, using four distinct embodiments (Humanoid, Quadruped, Cheetah, and Walker)***, which vary significantly in state and action dimensionality and therefore span different levels of control complexity. These results are presented in Figures 1, 4, and 15 for periodic continuous changes, and Figures 6, 16 and 17 for non-periodic settings.
>
> Importantly, these experiments demonstrate that our approach (SF + SC) generalizes beyond value-based methods such as DQN to actor-critic methods such as DDPG and TD3, suggesting that the benefits are not tied to a specific algorithmic class or environment structure.
>
> [1] Barreto, André, et al. "Successor features for transfer in reinforcement learning." NeurIPS (2017).
>
> [2] Liu, Hao, and Pieter Abbeel. "Aps: Active pretraining with successor features." ICML, 2021.
>
> [3] Barreto, Andre, et al. "Transfer in deep reinforcement learning using successor features and generalised policy improvement."ICML, 2018.
>
> [4] Borsa, Diana, et al. "Universal successor features approximators." (2018).
>
> [5] Chua, Raymond, et al. "Learning successor features the simple way."NeurIPS (2024)
>
> [6] Fujimoto, Scott, et al. "Addressing function approximation error in actor-critic methods." ICML, 2018.

---

> > ### Author Rebuttal · Reviewer_uAzh · 2026-04-05
> >
> > After reading the rebuttal, most of my concerns remain unresolved. In particular, regarding Weaknesses 2 and 4, I am not convinced that simply combining SC and SF constitutes a significant contribution, as both components are well-established in existing literature. Furthermore, the experimental evaluation remains insufficiently broad to support the paper's claims.
> >
> > For these reasons, I will maintain my current score.

---

> > > ### Author Response · Authors · 2026-04-07
> > >
> > > # Concern 2. Limited contribution
> > >
> > > ***We would like to clarify that the contribution of this work lies not in the individual components, but in how they are combined and studied to address two specific and underexplored questions in continual RL.***
> > >
> > > First, prior work in continual RL has largely focused on discrete and abrupt task changes, whereas real-world environments evolve in a more continuous manner. Our work investigates how existing approaches—whether aimed at preserving plasticity or stability—perform under such naturalistic, continuously evolving dynamics.
> > >
> > > Second, while prior consolidation-based methods (e.g., EWC) operate on the parameters of the Q-value function, we study whether consolidating predictive representations (i.e., Successor Features) is more effective in this setting. This distinction is important, as SFs encode environment dynamics in a structured way, which may interact differently with consolidation mechanisms.
> > >
> > > Our results (Figures 1, 3, 4, 6, 21-22) consistently show that applying synaptic consolidation to SFs leads to improved stability and performance across both navigation and mujoco tasks support the importance of this design choice.
> > >
> > > ***Taken together, this work provides a systematic study of consolidation applied to SFs under naturalistic, continuously evolving dynamics—a setting that, to the best of our knowledge (and as also pointed out by yourself and Reviewer RoCq), has not been previously explored.***
> > >
> > > If the reviewer is aware of closely related studies, we would greatly appreciate the reference and happy to discuss them in the manuscript.
> > >
> > > # Concern 4. Narrow experimental evaluation
> > > ***We respectfully disagree that our evaluation is insufficient to support the paper’s claims.***
> > >
> > > Our work is guided by two central research questions:
> > > 1. Under naturalistic, continuous environmental changes, is performance degradation driven by loss of plasticity or loss of stability?
> > >
> > > 2. If stability is the bottleneck, is it more effective to consolidate the parameters of the Q-value function or those of predictive representations, such as SFs?
> > >
> > > Accordingly, we prioritised experiments that directly address these questions across a diverse set of settings. Our evaluation spans:
> > > - Observation: pixels (Four Rooms) and states (DeepMind Control Suite (DMC))
> > > - Action spaces: discrete (Four Rooms) and continuous (DMC)
> > > - Architectures: value-based (Double DQN) and actor–critic (DDPG/TD3, with PPO added during rebuttal)
> > >
> > > The results are reported across Figures 1, 3, 4, 6-8, 15–17, and 19–28 in the main paper and appendices.
> > >
> > > ---
> > >
> > > Regarding generalisation to benchmarks such as Atari and DMC:
> > >
> > > We already include extensive experiments on the DMC, covering embodiments (Humanoid, Quadruped, Half-Cheetah, Walker) that span a wide range of complexity in continuous control.
> > >
> > > We did not include Atari because it does not support the type of continuous non-stationarity studied in this work. *Put another way, there is no natural way to make Atari non-stationary that doesn’t involve sudden, unrealistic jumps in the environment.* In contrast, our environments are designed to model naturalistic changes in dynamics (e.g., mass variation), which are central to our research questions. As such, we prioritised environments that allow interpretable continuous drift, rather than discrete task variations typical of Atari benchmarks.
> > >
> > > ---
> > >
> > > Finally, prior existing studies on SFs and stability–plasticity trade-offs have typically been evaluated in more limited settings:
> > >
> > > - Some focus only on simple navigation or reaching tasks with state inputs [1]
> > > - Others focus exclusively on Atari, without evaluation in continuous control domains [2,8]
> > > - Several works consider only state-observations or predefined task encodings [3–4]
> > > - Others evaluate only a subset of control tasks without considering discrete-actions and pixel-based RL environments [6–7]
> > > ---
> > >
> > > In summary, we believe our experimental evaluation is sufficiently broad and, importantly, well-aligned with our core questions. We would be happy to consider additional benchmarks if the reviewer believes they would provide new insights beyond those already demonstrated.
> > >
> > > [1] Barreto, André, et al. "Successor features for transfer in reinforcement learning." NeurIPS (2017)
> > >
> > > [2] Liu, Hao, and Pieter Abbeel. "Aps: Active pretraining with successor features." ICML, 2021
> > >
> > > [3] Barreto, Andre, et al. "Transfer in deep reinforcement learning using successor features and generalised policy improvement." ICML, 2018
> > >
> > > [4] Borsa, Diana, et al. "Universal successor features approximators." (2018)
> > >
> > > [5] Chua, Raymond, et al. "Learning successor features the simple way." NeurIPS (2024)
> > >
> > > [6] Touati, Ahmed, Jérémy Rapin, and Yann Ollivier. "Does zero-shot reinforcement learning exist?." (2022)
> > >
> > > [7] Dohare, Shibhansh, et al. "Loss of plasticity in deep continual learning." Nature (2024)
> > >
> > > [8] Hansen, Steven, et al. "Fast task inference with variational intrinsic successor features." (2019)

---

### Decision · Program_Chairs · 2026-04-30

**Decision:**

Accept (regular)

**Comment:**

Reviewers agreed that combining successor features with synaptic consolidation (Benna & Fusi) presents an interesting approach to addressing the stability-plasticity tradeoff in continually changing environments. They also noted that the presentation and empirical analysis with the chosen environments are well executed. Although reviewers expressed concerns about generalizing to other types of environments or non-stationarities, such as situations where transition dynamics change too rapidly for successor features to be useful, the revised paper is worthy of publication at this conference.